# Surface-circulation change in the Southern Ocean across the Middle Eocene Climatic Optimum: inferences from dinoflagellate cysts and biomarker paleothermometry

Margot J. Cramwinckel[1], Lineke Woelders[1,*], Emiel P. Huurdeman[2], Francien Peterse[1], Stephen J. Gallagher[3], Jörg Pross[2], Catherine E. Burgess[4,#], Gert-Jan Reichart[1,5], Appy Sluijs[1], Peter K. Bijl[1]

[1]Department of Earth Sciences, Faculty of Geoscience, Utrecht University, Utrecht, The Netherlands
[2]Paleoenvironmental Dynamics Group, Institute of Earth Sciences, Heidelberg University, Heidelberg, Germany
[3]School of Earth Sciences, The University of Melbourne, Melbourne, Australia
[4]School of Earth and Ocean Sciences, Cardiff University, Cardiff, United Kingdom
[5]NIOZ Royal Netherlands Institute for Sea Research and Utrecht University, Den Burg, Texel, The Netherlands
*Now at Institute of Arctic and Alpine Research, University of Colorado, Boulder, US
#Now at Shell UK LTD, Aberdeen, UK

*Correspondence to*: Margot J. Cramwinckel (m.j.cramwinckel@uu.nl)

**Abstract**

Global climate cooled from the early Eocene hothouse (~52–50 Ma) to the latest Eocene (~34 Ma). At the same time, the tectonic evolution of the Southern Ocean was characterized by the opening and deepening of circum-Antarctic gateways,
which affected both surface- and deep-ocean circulation. The Tasmanian Gateway played a key role in regulating ocean throughflow between Australia and Antarctica. Southern Ocean surface currents through and around the Tasmanian Gateway have left recognizable tracers in the spatiotemporal distribution of plankton fossils, including organic-walled dinoflagellate cysts. This spatiotemporal distribution depends on both physico-chemical properties of the water masses as well as the path of surface ocean currents. The extent to which climate and tectonics have influenced the distribution and composition of
surface currents and thus fossil assemblages has, however, remained unclear. In particular, the contribution of climate change to oceanographic changes, superimposed on long-term and gradual changes induced by tectonics, is still poorly understood.

To disentangle the effects of tectonism and climate in the southwest Pacific Ocean, we target a climatic deviation from the long-term Eocene cooling trend, the Middle Eocene Climatic Optimum (MECO; ~40 Ma). This 500 thousand year long
phase of global warming was unrelated to regional tectonism, and thus provides a test case to investigate the ocean's physiochemical response to climate change alone. We reconstruct changes in surface-water circulation and temperature in and around the Tasmanian Gateway during the MECO through new palynological and organic geochemical records from the central Tasmanian Gateway (Ocean Drilling Program Site 1170), the Otway Basin (southeastern Australia) and the Hampden Beach section (New Zealand). Our results confirm that dinocyst communities track specific surface ocean currents, yet the
variability within the communities can be driven by superimposed temperature change. Together with published results from

the east of the Tasmanian Gateway, our new results suggest that as surface ocean temperatures rose, the East Australian Current likely extended further south during the peak of MECO warmth. Simultaneous with high sea-surface temperatures in the Tasmanian Gateway area, pollen assemblages indicate warm temperate rainforests with paratropical elements along the southeastern margin of Australia. Finally, based on new age constraints we suggest that a regional southeast Australian transgression might have been caused by sea-level rise during MECO.

## 1 Introduction

The Eocene epoch (~56–34 millions of years ago; Ma) was characterised by gradual ocean cooling from the early Eocene hothouse (~52–50 Ma) to the early Oligocene icehouse (33 Ma), accompanied by decreasing atmospheric $CO_2$ concentrations (Zachos et al., 2008; Inglis et al., 2015; Anagnostou et al., 2016; Cramwinckel et al., 2018). In the framework of Eocene climate evolution, the Southern Ocean (SO) and its circulation are of particular interest. Geochemical tracers (Thomas et al., 2003; Huck et al., 2017) and model simulations using specific Eocene boundary conditions (Huber and Caballero, 2011) indicate that the SO, and the Southwest Pacific (SWP) in particular (Sijp et al., 2014; Baatsen et al., 2018), was the main source of intermediate and deep water formation during the early Paleogene. This effectively relays SO surface conditions to the global deep ocean. Several sites from the SWP sector of the SO have yielded proxy-based sea-surface temperatures (SSTs) (Bijl et al., 2009; Hollis et al., 2009, 2012) that are 5–10°C higher than the temperatures derived from the current generation of fully coupled climate models (Huber and Caballero, 2011; Lunt et al., 2012; Cramwinckel et al., 2018). These high sea water temperatures are supported by biomarker-based continental air temperature estimates, and vegetation reconstructions on the surrounding continents that indicate paratropical conditions (Pross et al., 2012; Carpenter et al., 2012; Contreras et al., 2013, 2014), although land and ocean temperatures did not necessarily change synchronously in this region (Pancost et al., 2013). This mismatch between proxy- and model-based temperatures has remained a conundrum.

As a result of tectonic processes, the bathymetry and geography of the Southern Ocean experienced major reorganizations in the Eocene (Kennett et al., 1974; Cande and Stock, 2004) that strongly affected regional and global ocean circulation (Huber et al., 2004; Sijp et al., 2014) (**Figure 1**). In the earliest Eocene, the Australian and South American continents were much closer to Antarctica (e.g., Cande and Stock, 2004) and obstructed circum-Antarctic ocean circulation. Instead, sub-polar gyres dominated circulation patterns in the southern sectors of the Indian and Pacific Ocean, transporting relatively warm surface waters to the Antarctic coast (Huber et al., 2004; Sijp et al., 2011; Baatsen et al., 2018) (**Figure 1a**). Tectonic activity in the Eocene led to the opening and subsequent deepening of the Tasmanian Gateway (Stickley et al., 2004b; Bijl et al., 2013b) and Drake Passage (Scher and Martin, 2004; Lagabrielle et al., 2009). Furthermore, a transition from northwesterly to accelerated northerly displacement of the Australian continent (Cande and Stock, 2004; Hill and Exon, 2004; Williams et al., 2019) and post-rift collapse of the outer continental shelf on both the Australian and Antarctic margins (Totterdell et al., 2000; Close et al., 2009) occurred. Subduction initiation affected vertical motion of submerged parts of northwestern

Zealandia including the Lord Howe Rise in the Tasman Sea (Sutherland et al., 2017, 2018). This complex tectonic evolution should have affected ocean circulation, and, in turn, heat transport and regional climate.

Along with the indirect inferences from modelling and heat distribution based on SST reconstructions, biogeographic patterns of surface-water plankton may be used as a tool to reconstruct surface-ocean circulation. In the Paleogene SO, high levels of endemism characterise a diverse range of fossil groups, including molluscs (Zinsmeister, 1979), radiolarians and diatoms (Harwood, 1991; Lazarus et al., 2008; Pascher et al., 2015), calcareous nannoplankton and planktonic foraminifera (Nelson and Cooke, 2001; Villa et al., 2008), and organic-walled dinoflagellate cysts (dinocysts) (Wrenn and Beckman, 1982; Wrenn and Hart, 1988; Bijl et al., 2011, 2013a). The endemic dinocyst assemblage from the Southern Ocean is traditionally referred to as "Transantarctic Flora" (Wrenn and Beckman, 1982). Here, following more recent extensive biogeographic mapping (Huber et al., 2004; Warnaar et al., 2009; Bijl et al., 2011, 2013b), we use these "Antarctic endemic dinocysts" to track Antarctica-derived surface currents, while cosmopolitan assemblages track currents sourced from the low latitudes. Throughout the Eocene, the Australian margin of the Australo-Antarctic Gulf (AAG) as well as New Zealand east of the Tasman Sea were characterised by high percentages of cosmopolitan dinocysts, implying an influence of the low-latitude-sourced Proto-Leeuwin Current (PLC) and the East Australian Current (EAC), respectively (**Figure 1**). In contrast, coeval assemblages on the eastern side of the Tasmanian Gateway were Antarctic-endemic, showing influence of the Antarctica-derived northward-flowing Tasman Current (TC) (Huber et al., 2004; Bijl et al., 2011, 2013b). From about ~50 Ma onwards, endemic dinocyst assemblages were established on both the Antarctic margin in the Australo-Antarctic Gulf and the eastern boundaries of the Tasmanian Gateway and Drake Passage (Bijl et al., 2011, 2013b). This indicates surficial westward flow through the Tasmanian Gateway of a proto-Antarctic Counter Current (proto-ACC), which is supported by simulations using an intermediate-complexity coupled model (Sijp et al., 2016). Pronounced widening and deepening of the gateway did not start until the late Eocene (Stickley et al., 2004b), although some subsidence already took place during the middle Eocene (Röhl et al., 2004).

These biogeographical patterns broadly confirm the Paleogene ocean circulation patterns as simulated by numerical climate models (Huber et al., 2004). Thus, on tectonic timescales (i.e., tens of Myrs), plankton biogeographical patterns predominantly follow changes in surface-ocean circulation (Bijl et al., 2011). During periods with a relatively stable ocean-current configuration, such as the middle Eocene, SO dinocyst assemblage variability was instead driven by (orbital-scale; Warnaar et al., 2009) climatic factors such as SST (Bijl et al., 2011). Superimposed changes in SWP dinocyst assemblages also occur during transient climate change such as the Paleocene-Eocene Thermal Maximum (PETM, ~56 Ma, (Sluijs et al., 2011)) and the Middle Eocene Climatic Optimum (MECO, ~40 Ma, (Bijl et al., 2010)). During the PETM, global warming of ~5 °C occurred within millennia, associated with the injection of a large mass of reduced carbon into the ocean-atmosphere system, which resulted in the appearance of tropical dinocyst taxa at the East Tasman Plateau (Sluijs et al., 2011). In contrast, although with similar magnitude of warming, the MECO was a 500 thousand year (kyr) period of

transient warming of the global deep ocean (Bohaty et al., 2009; Bohaty and Zachos, 2003) and surface ocean (Boscolo-Galazzo et al., 2014; Cramwinckel et al., 2018). Regionally, the MECO was associated with changes in oceanic productivity and oxygenation, reflected by changes in planktic and benthic assemblages (e.g., Spofforth et al., 2010; Boscolo-Galazzo et al., 2015; Cramwinckel et al., 2019). However, the mechanism that caused MECO warming remains enigmatic. Deep-ocean
carbonate dissolution (Bohaty et al., 2009), indications for $pCO_2$ rise (Bijl et al., 2010; Steinthorsdottir et al., 2019) and a diminished weathering feedback (van der Ploeg et al., 2018) during the MECO imply that climate change was forced by an accumulation of carbon in the exogenic carbon pool. The lack of a negative trend in stable carbon isotope ratios ($\delta^{13}$C) over the MECO suggests this carbon to be volcanic, rather than organic, in origin (Bohaty et al., 2009). One of the proposed MECO carbon-cycle scenarios suggests a global sea-level rise in order to shift the locus of carbonate deposition from the
deep ocean to the continental shelves (Sluijs et al., 2013). Although speculative isotopic evidence for a MECO-associated change in glacioeustasy exists (Dawber et al., 2011), constraints on global sea level change during the MECO are lacking. At the East Tasman Plateau, the MECO is characterised by an incursion of low-latitude dinocyst taxa that temporarily replaced the largely endemic Antarctic community (Bijl et al., 2010). The origin of these cosmopolitan dinocysts remains an unresolved question. Potentially, cosmopolitan dinoflagellates outcompeted the Antarctic-endemic taxa in the warming TC,
similar to during the PETM. Alternatively, a southward extension of the EAC from the north or leakage of the PLC from the west through the Tasmanian Gateway supplied cosmopolitan assemblages to the region east of Tasmania, possibly even associated with sea level rise.

To disentangle the effects of tectonism and climate change in the southwest Pacific Ocean, we here assess the biotic and
oceanographic response in that region to MECO warming. The MECO allows us to assess oceanographic response to climate change, independent of tectonic change. We reconstruct surface-ocean circulation and temperature by generating new dinocyst and organic geochemical records from Ocean Drilling Program (ODP) Site 1170 on the South Tasman Rise in the central Tasmanian Gateway. We place these records into their broader regional context by comparing them to newly generated middle Eocene palynological records, including pollen from terrestrial plants, from the Otway Basin (SE
Australia) and the Hampden Beach section (New Zealand) (**Figure 2a**).

## 2 Material

### 2.1 South Tasman Rise (ODP Site 1170) and East Tasman Plateau (ODP Site 1172)

Ocean Drilling Program Site 1170 is located at a water depth of ~2704 m, 400 km south of Tasmania at 47.1507° S and 146.0498° E (Exon et al., 2001) (**Figure 2a**). It was drilled on the western side of the South Tasman Rise (STR), a
continental block to the south of present-day Tasmania. The site is located in a 2–3 km deep and 50 km wide graben within the Ninene Basin (**Figure 2b**). A ~300 m thick package of shallow marine silty claystones of middle Eocene age overlies an erosional unconformity. Northwest-southeast rifting between Australia and Antarctica accelerated after 51 Ma, resulting in

prominent NW-SE structural trends in seabed seismic topography associated with seafloor spreading between Tasmania-STR on the one side and Antarctica on the other (Exon et al., 2004; Bijl et al., 2013b; Williams et al., 2019) (**Figure 2a**). This coincided with renewed subsidence of both conjugate continental margins (Totterdell et al., 2000) and the STR (Hill and Exon, 2004). Marked lateral thinning of middle Eocene deposits at Site 1170 is apparent in the seismic profile,

suggesting synsedimentary growth faulting caused local subsidence (**Figure 2c**). Middle Eocene sediments are present in Hole 1170D as a thick sequence from ~500 metres below sea floor (mbsf) to the total depth at 780 mbsf (Exon et al., 2001). The precise age of the middle Eocene strata at Site 1170 has thus far not been well constrained (Stickley et al., 2004a). Nevertheless, the thickness of the middle Eocene sequence implies high sedimentation rates (Exon et al., 2001), together with the seismic evidence suggesting that the surrounding graben was a depocenter that formed as rifting developed. Middle

Eocene sediments are overlain by latest Eocene-earliest Oligocene glauconite-rich clayey siltstones (Exon et al., 2001; Sluijs et al., 2003; Stickley et al., 2004a). Here, we target the middle Eocene claystones from the interval ~500–780 mbsf for dinocyst biogeography and organic geochemistry, to gain a central Tasmanian Gateway perspective on regional effects of the MECO.

Ocean Drilling Program Site 1172 is located at a water depth of ~2620 m on thinned continental crust on the western side of the East Tasman Plateau (ETP), ~170 km southeast of Tasmania at 43.9598° S and 149.9283° E (Exon et al., 2001) (**Figure 2a**). While the ETP has a similar tectonic history to the STR, Site 1172 was not affected by growth faulting and subsidence like Site 1170 during the middle Eocene (Hill and Moore, 2001). Palynological and organic geochemical results for the middle Eocene of the East Tasman Plateau are presented in Bijl et al. (2009, 2010, 2011, 2013a), and are compared to our

results from the South Tasman Rise in this study.

## 2.2      Latrobe-1 borehole, Otway Basin (Australo-Antarctic Gulf, Southeast Australia)

Sediment cores from the Otway Basin, on the Australian margin of the AAG (**Figure 2a**), were analysed as a location under influence of the PLC during the MECO. The Otway Basin contains a regionally thick sequence of shallow marine Paleogene

deposits (Gallagher et al., 1999; Gallagher and Holdgate, 2000). These deposits developed due to Paleocene-Eocene post-rift extension on the edge of the continental margin, causing subsidence of extensive troughs that served as depocentres of terrigenous sediment in deltaic and shallow marine environments (Krassay et al., 2004; Stacey et al., 2013; Frieling et al., 2018a). In southeast Australia, the middle Eocene to early Oligocene Nirranda Group uncomformably overlies the early Eocene Dilwyn Formation (Wangerrip Group) (Abele, 1994; Krassay et al., 2004; Tickell et al., 1993). This unconformity

can be traced throughout southeast Australia (Holdgate et al., 2003). The overlying Wilson Bluff transgressive deposits have an age between 44 and 40 Ma (Holdgate et al., 2003; McGowran et al., 2004). In the Portland Trough and Port Campbell Embayment of the Otway Basin, the basal part of the Nirranda Group consists mainly of the Burrungule and Sturgess Point members. Outside of these main depocentres and on the ridges in between, the basal part of the Nirranda Group is

represented by the Narrawaturk Formation. Planktonic foraminiferal biostratigraphy indicates a Bartonian age for the Sturgess Point Member (Abele, 1994; Gallagher and Holdgate, 2000).

The Latrobe-1 borehole (38.693009° S, 143.149995° E) was drilled in 1963–1964 near the Port Campbell Embayment, reaching a total depth of 620 metres. It spans Cretaceous to Eocene sediments, with initial biostratigraphic age constraints (Archer, 1977; Taylor, 1964; Tickell et al., 1993) and well log data (White, 1963) placing the middle Eocene Narrawaturk Fm at a depth of 60–76 metres below surface (mbs), overlying the early Eocene Dilwyn Fm (76–289 mbs). The Dilwyn Fm in the Latrobe-1 core consists largely of light- to dark brown sandstones with some contributions of mud- and siltstone, while the Narrawaturk Fm is a dark brown muddy sandstone (Frieling et al., 2018a). Based on the occurrence of the stratigraphic marker dinocysts *Achilleodinium biformoides* and *Dracodinium rhomboideum*, and in accordance with the regional dinocyst zonation (Bijl et al., 2013a) sediments around a depth of 67.35 metres below surface (mbs) in the Narrawaturk Fm (Nirranda Group) of the Latrobe-1 borehole have an age near the MECO (Frieling et al., 2018a). Here, we target 4 samples from the Latrobe-1 core Narrawaturk Fm (interval ~60-90 mbs) for palynology and organic geochemistry.

## 2.3    Hampden Beach section (South Island, New Zealand)

The Hampden Beach section at Hampden Beach, New Zealand (**Figure 2a**) (45.30° S, 170.83° E), was analysed to identify influences of TC and/or EAC at southern New Zealand in the middle Eocene prior to the MECO (Hines et al., 2017). This 256.5 m thick section spans the Paleocene to late Eocene and has a well-resolved foraminiferal biostratigraphy (Morgans, 2009). Middle Eocene sediments of the Hampden Beach section consist of calcareous clay-rich siltstone to very fine sandstone. Benthic foraminiferal assemblages suggest a depositional environment near the shelf-slope transition. An interval of 4 m was previously selected for high-resolution investigation (Burgess et al., 2008). This interval spans 70 kyr around 41.7 Ma, based on biostratigraphy and orbital interpretation of lithological cycles. Sea-surface temperature (SST) reconstructions based on Mg/Ca and $\delta^{18}O$ of excellently preserved foraminifera and $TEX_{86}$ indicate values of 23–25 °C (Burgess et al., 2008), which is consistent with regional Eocene SST reconstructions (Hollis et al., 2012; Inglis et al., 2015). We have analysed the same 4 m interval for dinocyst biogeography.

## 3    Methods

## 3.1    Palynology

### 3.1.1    Processing and analysis

A total of 43 samples from ODP Site 1170 (Hole 1170D), 8 samples from the Latrobe-1 core, and 39 samples from the Hampden Beach section were processed for palynology following standard procedures. A known amount of *Lycopodium*

*clavatum* spores was added for quantification of the dinocyst content in specimens per gram. Sediment samples were crushed and oven dried (60 °C), followed by treatment with 30% HCl and ~40% HF to dissolve carbonate and silicate minerals, respectively. After each acid step, samples were washed with water, centrifuged or settled for 24 h, and decanted. The residue was sieved over nylon mesh sieves of 250 μm and 10 μm (Site 1170) or 15 μm (Otway Basin, Hampden Beach section) and subjected to an ultrasonic bath to break up agglutinated particles of the residue. A drop of the homogenised residue was mounted on a glass microscope slide with glycerine jelly and sealed. All slides are stored in the collection of the Laboratory of Palaeobotany and Palynology, Utrecht University. Palynomorphs were counted up to a minimum of 200 identified dinocysts for ODP Site 1170, typically to the taxonomic level of genus or species. Because the dinocyst yield was relatively low for the other localities, palynomorphs were counted up to a minimum of 90 (Hampden Beach section) or 50 (Otway Basin) identified dinocysts. Terrestrial palynomorphs were counted in broad categories of gymnosperm pollen, angiosperm pollen and spores for Site 1170 and the Hampden Beach section. As the Otway Basin samples yielded diverse and abundant sporomorph assemblages, a minimum of 300 sporomorphs was counted per sample. Dinocyst taxonomy as cited in Williams et al. (2017) was generally followed, with the exception of the wetzelielloids. For this group, we follow the suggestion made in Bijl et al. (2016) to use the taxonomy of Fensome and Williams (2004) instead of (Williams et al., 2015, 2017a). Sporomorph taxonomy follows Stover and Partridge (1973), Macphail et al. (1994), and Raine et al. (2011).

### 3.1.2 Dinocyst biostratigraphy and palaeogeographic affinity

Regional dinocyst biostratigraphy for the middle Eocene is based on Bijl et al. (2013a) (ages presented in table 2 of that work). Dinocyst-based environmental interpretation follows Sluijs et al. (2005), Sluijs and Brinkhuis (2009), and Frieling and Sluijs (2018). For biogeographic analysis, dinocyst taxa were binned into Antarctic endemics, cosmopolitan taxa, and mid-/low-latitude taxa (**Supplementary Data**). Shifts in relative abundance between these groups signal changes in surface ocean currents. Surface ocean currents deriving from the water surrounding Antarctica are dominated by Antarctic endemics, whereas low-latitude derived current such as the EAC and PLC transport more cosmopolitan and mid-/low-latitude taxa. We primarily follow the biogeographical groupings of Bijl et al. (2011) and (2013b), based on occurrence and stratigraphic range of species at different latitudes. Cosmopolitan dinocysts are those taxa that have been recorded globally, at low (tropics), middle (subtropical and temperate) and high (polar) latitudes. The Antarctic endemic group consists of species endemic to either the Southern Ocean (including the Transantarctic Flora (TF) cf. Wrenn and Beckman (1982)) or both the Southern Ocean and northern high latitudes (bipolar taxa). To the mid/low-latitude group we add those taxa that are considered thermophilic (all wetzellioids and goniodomids) based on recent empirical information on ecological affinities of Paleogene dinocysts (Frieling and Sluijs, 2018). We note that this addition only constitutes a minor change in biogeographic grouping for this study (**Supplementary Data**).

Taxa with unknown biogeographic affinities were excluded from biogeographical analysis. For instance, a large fraction of *Deflandrea* specimens that lost their outer bodies could not be identified to the species level. As some *Deflandrea* species are endemic to the SO, while others are cosmopolitan, we have excluded these specimens (and other taxa with unknown affinity) from biogeographic analysis. We note that a different choice was made for the published middle Eocene dinocyst assemblages from Site 1172, where the only *Deflandrea* species recorded was *D. antarctica*; consequently, *Deflandrea* inner bodies were counted as *D. antarctica* (Bijl et al., 2011). Endemic and cosmopolitan dinocysts during the MECO at Sites 1170 and 1172 largely consist of two species belonging to the genus *Enneadocysta*, i.e., the cosmopolitan species *Enneadocysta multicornuta* and the Southern Ocean endemic *Enneadocysta dictyostila*. These species are morphologically similar, but differ by their tabulation patterns and the morphology of the distal ends of the processes (Fensome et al., 2006) (**Supplementary Figure 1**). The species morphology has been crosschecked with the original Site 1172 material and dinocyst counts to validate consistency in species determination. The above biogeographical affinity of dinocysts, in particular the relative abundance of endemic *vs.* non-endemic dinocyst taxa, is used here to distinguish the relative influence of the Antarctic-derived TC *vs.* the lower-latitude-derived EAC and PLC.

## 3.2     Organic geochemistry

To quantify SST changes, 52 samples from ODP Hole 1170D and one sample from the Latrobe-1 core were processed for $TEX_{86}$ palaeothermometry based on isoprenoid glycerol dibiphytanyl glycerol tetraether (GDGT) membrane lipids of marine archaea (Schouten et al., 2002). The GDGTs were extracted from freeze-dried, powdered samples (~8–10 g dry weight) with dichloromethane (DCM):methanol (MeOH) (9:1, v:v) using a Dionex accelerated solvent extractor (ASE) 350, at a temperature of 100°C and a pressure of $7.6 \times 106$ Pa. Lipid extracts were subsequently separated by $Al_2O_3$ column chromatography into 4 fractions, using hexane:dichloromethane (DCM) (9:1, v/v), ethyl acetate (100%), DCM:MeOH (95:5, v/v) and DCM:MeOH (1:1, v/v). For quantification purposes, 9.9 ng of a $C_{46}$ GDGT internal standard (*m/z* 744) was added to the DCM:MeOH (95:5, v/v) fraction after this. This fraction, containing the GDGTs, was subsequently dissolved in hexane:isopropanol (99:1, v/v) to a concentration of ~3 mg/mL, passed through a 0.45 μm polytetrafluoroethylene (PTFE) filter and analysed using ultra-high performance liquid chromatography-mass spectrometry (UHPLC-MS) following (Hopmans et al., 2016). We note that the published $TEX_{86}$ records from Site 1172 and the Hampden Beach section were generated using high performance liquid chromatography-mass spectrometry (HPLC-MS) after (Schouten et al., 2007), but differences in $TEX_{86}$ values between the two methods have been shown to be negligible (Hopmans et al., 2016). Samples with very low concentrations (i.e., peak area < 3000 mV and/or peak height < 3x background signal) of any GDGT included in $TEX_{86}$ were excluded from analysis. Based on relative abundances of GDGTs, the $TEX_{86}$ and Branched versus Isoprenoid Tetraether (BIT) index values were calculated following Schouten et al. (2002) and Hopmans et al. (2004), respectively. The BIT index is used as an indicator for the contribution of terrestrially-derived organic material to the marine realm, relative to influence of marine production. High BIT index values indicate a primarily terrestrial origin of GDGTs and/or low marine production of GDGTs, whereas low BIT values indicate dominance of marine-produced GDGTs over a smaller contribution

of terrestrial GDGTs. BIT index values >0.3 imply $TEX_{86}$ might not correctly reflect SST due to an overprint by a terrestrial-derived signal (Weijers et al., 2006). In addition, several other ratios were calculated to evaluate GDGT sourcing and thus the reliability of $TEX_{86}$-based SST estimates. In short, the Methane Index (MI) (Zhang et al., 2011) and GDGT-2/crenarchaeol (Weijers et al., 2011), GDGT-0/crenarchaeol (Blaga et al., 2009), and GDGT-2/GDGT-3 (Taylor et al., 2013)

indices are calculated to investigate potential contributions by methanotrophic, methanogenic, and deep-dwelling GDGT producers to the GDGT pool in the sediments. The analytical precision for $TEX_{86}$ is ±0.3°C based on long-term analysis of in-house standards. $TEX_{86}$-to-SST calibrations include those based on mesocosm experiments and core-top datasets. We prefer the latter for paleoreconstructions, as these integrate ecological, water-column and diagenetic effects that are not incorporated in mesocosm experiments. Since our measured $TEX_{86}$ values are within the range of the modern core-top

dataset (≤0.73), no extrapolation of the modern $TEX_{86}$-to-SST relationship is necessary, and differences between linearly and exponentially fitted calibrations are small (see for example extended data figure 2 in Cramwinckel et al. (2018)). Here we calculate SST from $TEX_{86}$ values using both the exponential calibration of Kim et al. (2010) and the linear calibration of O'Brien et al. (2017) (**Supplementary Data**). Since the resulting values are highly similar, we present only the values from a single calibration, the $TEX_{86}^{H}$ calibration, in our figures. We note that however, the interest of this study primarily lies in

comparing geographic differences in SST and not absolute temperature values.

### 3.3     Statistical analyses

To assess the main patterns within the changing dinocyst assemblages at the studied sites, unconstrained ordination was applied on the proportional abundances. Both Nonmetric MultiDimensional Scaling (NMDS) and Detrended

Correspondence Analysis (DCA) were performed, using the R Package Vegan (Oksanen et al., 2015). Whereas DCA assumes a unimodal species response to the environment, NMDS is a distance-based method that does not assume any relationship, which can be considered more neutral because it introduces less assumptions (Prentice, 1977). For NMDS, the Bray-Curtis measure was used as an appropriate dissimilarity index for (paleo-) ecological community data (e.g., Faith et al., 1987), and recommendations by Clarke (1993) were followed to set the number (two or three) of dimensions used in the

ordination. Unconstrained ordination was performed on the full dinocyst assemblages from Site 1170 and Hampden Beach (this study) and Site 1172 (Bijl et al., 2010, 2011, 2013a). Furthermore, unconstrained ordination was applied to the combined dinocyst assemblages of Site 1170, Site 1172, Otway Basin and Hampden Beach. We note that caution should be taken when performing statistical analyses on microfossil assemblage counts of less than 150–200 palynomorphs (minimum 50 for Latrobe-1, minimum 90 for Hampden Beach), as diversity will likely be underrepresented. While this introduces

biases into measures of diversity and variability of the assemblage, ordination-type analyses that establish the dominant patterns within the data should be more robust for low count data.

To investigate whether dinocyst assemblage change at Site 1170 correlates with environmental change, constrained ordination using Canonical Correspondence Analysis (CCA) was performed with the R Package Vegan. We assess different

sets of environmental proxy data, including SST (based on $TEX_{86}$; this study), input of terrestrial material (BIT; this study), shipboard-generated clay contents from smear slide analysis, uranium contents, magnetic susceptibility, and colour reflectance data (Mascle et al., 1996). Higher-resolution environmental data were interpolated to the sampling resolution used here for palynology. As with DCA, CCA assumes a unimodal species response to the input environmental variables.

## 4    Results

### 4.1    Site 1170

#### 4.1.1    Palynology

Middle Eocene palynomorphs at Site 1170 are generally well preserved and assemblages are dominated (>95%) by marine forms, mainly dinocysts. Terrestrial palynomorphs occur consistently, but in low relative abundances (<5% of palynomorphs). The presence of *Impagidinium* spp. in all samples indicates an open marine setting (Dale, 1996), suggesting that palynomorphs characteristic of inshore environments have been transported off-shelf, possibly from the north. Absolute concentrations of dinocysts are extremely high, averaging ~175,000 dinocysts per gram of dry sediment over the studied section, with maxima of over 400,000 cysts per gram. The dinocyst assemblages are generally of low diversity and consist of three dominant groups that typically comprise over 90% of the total assemblage. These groups are: *Enneadocysta dictyostila*, *Deflandrea* spp. and spiny peridinioids *sensu* Sluijs et al. (2009). High abundances of *Enneadocysta* spp. and peridinioid dinocysts in combination with low diversity indicate a somewhat restricted, eutrophic assemblage with possible low-salinity influences (Sluijs et al., 2005). Endemic taxa dominate the record, typically accounting for more than half of the assemblage (**Figure 3**). The most abundant endemic species is *E. dictyostila,* particularly from 570–690 mbsf. Endemic *Vozzhennikovia apertura* also has a high average relative abundance (~20%). Other, rarer endemics include *Arachnodinium antarcticum*, *Deflandrea antarctica*, *Enneadocysta brevistila*, *Octodinium askiniae*, *Spinidinium macmurdoense*, *S. schellenbergii*, and *Vozzhennikovia netrona*. Cosmopolitan dinocyst species on average make up about 10% of the assemblage, consisting among others of *Cerebrocysta* spp., *Cordosphaeridium* spp., *Enneadocysta multicornuta*, *Operculodinium centrocarpum*, and *Thalassiphora pelagica*. Mid-/low-latitude taxa are rare. Selected taxa are illustrated in **Supplementary Figure 1**.

#### 4.1.2    Organic geochemistry and sea-surface temperatures

Out of 52 samples from Hole 1170D, five were disregarded for $TEX_{86}$ analysis due to low GDGT concentrations, particularly in the lower part of the section. The remaining 47 samples have isoprenoid GDGT concentrations of on average $18 \pm 10$ ng per g sediment. BIT index values (Hopmans et al., 2004) are consistently below 0.25, indicating a dominant marine source of the isoprenoid GDGTs (Weijers et al., 2006). Furthermore, MI values (Zhang et al., 2011) and GDGT-2/Cren ratios (Weijers et al., 2011) are below 0.3 and 0.2, respectively, indicating no substantial GDGT contributions by methanotrophic archaea. Finally, GDGT-0/Cren ratios (Blaga et al., 2009) are never above 1.2, indicating normal marine conditions, without substantial contributions by methanogenic archaea. Based on the $TEX_{86}^{H}$ calibration, $TEX_{86}$-derived SSTs

are mostly between 20–28°C, similar to time-equivalent temperatures at the East Tasman Plateau (Bijl et al., 2010) (**Figure 3**). Maximum temperatures of ~28°C are reached around 670 mbsf, and temperatures decline gradually towards the top of the studied section. Large temperature variability of several degrees between consecutive samples is recorded particularly in the interval from 600 to 550 mbsf (**Figure 3**).

### 4.1.3 Biochronostratigraphic framework

Some biostratigraphically informative dinocyst species are present. *Selenopemphix* spp. and *Impagidinium parvireticulatum* occur sparsely throughout the investigated samples from Site 1170, with their oldest occurrence at ~766 mbsf (second-to-lowermost sample). Their regional first occurrences are at 48.6 Ma and 44.0 Ma (GTS2012), respectively (Bijl et al., 2013a).

Presence of *Impagidinium parvireticulatum* thus constrains the studied sediments to an age younger than 44 Ma. The single occurrence of *Lophocysta* spp. at 569 mbsf provides a narrow age range around the MECO for this part of the investigated core, from 41.39 to 39.66 Ma (Bijl et al., 2013a). Additional age constraints from magnetostratigraphy are not possible, as inclination data suffered from a persistent large overprint (Stickley et al., 2004a). The few available shipboard nannofossil datums do not add further constraints, but confirm sediments of MECO age should lie within the studied interval (Stickley et

al., 2004a). Based on the above constraints, we consider the recorded $TEX_{86}$-based temperature maximum at ~670 mbsf to reflect the peak of the MECO and the subsequent surface ocean cooling trend to represent the MECO recovery phase ("Option 1" in **Figure 3a**). An alternative interpretation would be to consider the warming interval from ~610 to ~580 mbsf as MECO warming ("Option 2" in **Figure 3a**), which would suggest peak MECO temperatures at ~580 mbsf. However, this would imply a pre-MECO peak in temperature at ~670 mbsf. This would strongly conflict with temperature evolution across

the middle Eocene and MECO as recorded at numerous sites across the global ocean, including the nearby Site 1172 (e.g., Bijl et al., 2010; Boscolo-Galazzo et al., 2014; Cramwinckel et al., 2018). We therefore prefer the first interpretation, even though it implies (very) high sedimentation rates of 10s of centimetres per thousand years. Such rates are consistent with the middle Eocene locality of Site 1170 in a depocenter on the northeast-southwest rifting South Tasman Rise (**Figure 2b-c**). While these constraints are valuable in delimiting our study interval to the MECO, stratigraphic correlation based on

temperature records is precarious and the lack of precise and consistent age-depth tie-points impedes the construction of a solid age–depth model. We therefore present the data for Site 1170 in the depth domain.

## 4.2 Otway Basin

### 4.2.1 Marine palynology

The palynomorph assemblages from the Latrobe-1 borehole consist predominantly of sporomorphs. Absolute concentrations of dinocysts are in the order of 100–1,000 cysts per gram of dry sediment, while sporomorphs total 2,000–5,000 grains per gram of dry sediment. Sufficient dinocysts were encountered for counts of ~50–100 identified dinocysts to be undertaken. Other marine palynomoprhs such as prasinophytes and acritarchs were rare. The *Spiniferites* complex is dominant (averaging ~40 %), and *Enneadocysta* spp. (mostly consisting of *E. multicornuta*) are common (averaging ~20 %). Other minor

constituents include *Cleistosphaeridium* spp., *Cordosphaeridium* spp., *Deflandrea* spp*., *Elytrocysta* spp., *Hystrichosphaeridium* spp., and *Phthanoperidinium* spp. Notably, the dinocyst assemblages do not yield Antarctic endemic taxa; instead, they are composed solely of cosmopolitan and low-/mid-latitude taxa. Combined, the marine palynology indicates a proximal marine setting.

### 4.2.2    Terrestrial palynology

The middle Eocene sporomorph assemblage from the Latrobe-1 borehole consists of abundant gymnosperm (30–50 %) and angiosperm (30–50 %) pollen, with pteridophyte spores as a minor component of the assemblage (10–15 %). Saccate pollen are mainly represented by *Podocarpidites* spp. (*Podocarpus*), *Lygistepollenites* (*Dacrydium*) and *Phyllocladites* spp. (*Lagarostrobus*); other gymnosperms are Araucariaceae (10–20 %), which consist mainly of *Dilwynites* spp. (*Agathis/Wollemia*) and, to a lesser extent, of *Araucariacites* spp. (*Araucaria*). Angiosperm pollen are dominated by *Myricipites* spp. (Casuarinaceae; *Gymnostoma*), *Nothofagidites* (including *Nothofagus* sg. *Brassospora*) and *Malvacipollis* spp. (*Austrobuxus/Dissilaria*), with *Proteacidites* spp. and *Rhoipites* spp. as minor elements. Pteridophyte spores are mainly represented by *Cyathidites* spp. and *Laevigatosporites* spp. Furthermore, *Cycadopites* spp. (Cycadophyta), *Arecipites* spp. (Arecaeae), and *Santalumidites* spp. (*Santalum*) are also present but rare. Selected taxa are illustrated in **Supplementary Figure 1**. A stratigraphic log of the Latrobe-1 borehole and a pollen diagram are presented in **Supplementary Figure 2.**

### 4.2.3    Organic geochemistry

The analysed sample from the Latrobe-1 borehole contains predominantly terrestrial-derived branched GDGTs, resulting in a BIT index of 0.79, making the sample unsuitable for $TEX_{86}$ analysis.

### 4.2.4    Stratigraphy

Our new palynological data further constrain the position of the early-middle Eocene hiatus that was recognised in the Latrobe-1 borehole between 67.35 and 97.84 mbs (Frieling et al., 2018a) to a depth between 78.98 and 70.32 mbs. The hiatus therefore likely corresponds to the transition between the Dilwyn Formation (Wangerrip Group) and the Narrawaturk Marl (Nirranda Group) at ~70.5 mbs. Dinocyst species with biostratigraphic utility in strata above the unconformity include *Phthanoperidinium comatum* (FO 45.70 ± 0.20 Ma) and *Phthanoperidinium stockmansii* (FO 57.20 ± 0.20 Ma), *Achilleodinium biformoides* (recorded in ODP Site 1171 South Pacific Dinocyst Zone (SPDZ) 13)*, *and *Dracodinium rhomboideum* (**Supplementary Figure 1c**) (Bijl et al., 2013a). Occurrence of this last species is especially informative, as the stratigraphic range of *Dracodinium rhomboideum* in the South Pacific Dinocyst Zonation of Bijl et al. (2013a) is very restricted. In fact, it was only present in one sample at Site 1172, with an age of 40.00 ± 0.10 Ma, within Chron 18n.2n. This corresponds to peak MECO in the compilation of deep sea stable isotope records (Bohaty et al. 2009) as well as coinciding with peak MECO SSTs based on $TEX_{86}$ at Site 1172. Notably, the range of *D. rhomboideum* in the North Atlantic Ocean (Eldrett et al., 2004) is similarly restricted to the MECO interval (from C18n.2n 0% to C18n.1r 50%, corresponding to 40.14

Ma – 39.66 Ma), indicating this species to be a useful biostratigraphic marker for the MECO. The interval from 61.46 to 70.32 mbs in the Latrobe-1 borehole is therefore assigned to SPDZ13 (40.0–35.95 Ma) based on the regional dinocyst zonation of Bijl et al. (2013a). Moreover, the presence of *Dracodinium rhomboideum* in samples at 63.82 and 67.35 mbs indicate coverage of the MECO.

## 4.3 Hampden Beach

### 4.3.1 Palynology

Middle Eocene palynological assemblages at Hampden Beach are dominated by dinocysts (~65 %), with abundant sporomorphs (~30 %) and some acritarchs and prasinophytes. Sediments yield several thousand dinocysts per gram of dry sediment. The consistent presence of *Impagidinium* spp. (mean: ~7 %) indicates an open-ocean setting. The dinocyst assemblages comprise predominantly cosmopolitan and low-/mid-latitude taxa. Similar to the assemblages from the Latrobe-1 borehole, the outer neritic *Spiniferites* cpx. is dominant (averaging ~40 %). Other common cosmopolitan and low-/mid-latitude taxa include *Cordosphaeridium fibrospinosum, Dapsilidinium* spp.*, Elytrocysta brevis, Hystrichokolpoma rigaudiae, Hystrichosphaeridium tubiferum*, and *Senegalinium* spp. (together averaging ~35 %). Antarctic endemic species occur sparsely (averaging ~6 %) and consist of *Deflandrea antarctica*, *Enneadocysta dictyostila* and *Pyxidinopsis delicata*. This dinocyst assemblage is in agreement with the age of c. 41.7 Ma as previously assigned to this 4 m-thick interval within the section (Burgess et al., 2008).

## 5 Discussion

### 5.1 Surface-ocean circulation in the Southwest Pacific during the MECO

Our new dinocyst biogeographic data are generally consistent with previous interpretations of Tasmanian Gateway surface-ocean circulation based on plankton biogeography and model simulations (Huber et al., 2004; Bijl et al., 2011; Sijp et al., 2016) (**Figure 1b**). By the middle Eocene, the Antarctic endemic dinocyst assemblage associated with the proto-ACC and TC had become firmly established, while the northern bound of the AAG was primarily influenced by the low-latitude-derived PLC. Records from southern New Zealand yield a predominantly warm EAC signal, with a minor, yet constant influx of Antarctic endemics indicating limited TC influence (this study and Bijl et al., 2011).

Throughout the studied middle Eocene interval, dinocyst assemblages at Site 1170 are dominated by Antarctic-endemic taxa. This implies that the Tasmanian Gateway was influenced by westward atmospheric and surface-oceanic circulation (i.e., the polar easterlies) around 40 Ma, with the polar front thus located to the north of the gateway and the proto-ACC flowing westward through the Tasmanian Gateway (**Figure 1b**). This is supported by the similar range of $TEX_{86}$ SSTs of 20–28ºC within (Site 1170) and east of (Site 1172) the Tasmanian Gateway (**Figure 3**). In terms of paleolatitude reconstructions,

placing Site 1170 within the Tasmanian Gateway south of 60ºS at this time is within the uncertainty limits of current generation mantle (e.g., Matthews et al., 2016) as well as paleomagnetic reference frames (e.g., Torsvik et al., 2012). Notably, however, the shift in dominance from endemic to cosmopolitan dinocysts that occurs at the zenith of MECO warmth on the East Tasman Plateau (Site 1172) has no equivalent on the South Tasman Rise (Site 1170) (**Figure 3**). The dominance of cosmopolitan dinocysts at Site 1172 therefore cannot be explained by the warming TC and Ross Sea gyre alone, as this effect would have resulted in a dinocyst assemblage similar to Site 1170.

Two possible oceanographic features could have resulted in a dominantly cosmopolitan dinocyst assemblage at Site 1172 and not at Site 1170. First, weak eastward flow could have occurred through Bass Strait and/or the northern portion of the Tasmanian Gateway from the AAG (**Figure 1c**). The uncertainty on paleolatitude in principle allows for weak continuous eastward flow (or discontinuous eddy transport) under influence of the westerlies through the northern part of the TG. While this remains a possible scenario, we consider it unlikely that such a nearby current would not be reflected in the plankton assemblages at the depocenter of Site 1170, particularly since the widest opening in the TG would be located south of the South Tasman Rise (Bijl et al., 2013b), close to Site 1170. In addition, the Bass Strait, or Bass Basin, to the north of Tasmania was likely too restricted at its eastern end for throughflow (Cande and Stock, 2004). As the second option, southward extension and/or intensification of the EAC could have sustained cosmopolitan assemblages at Site 1172 (**Figure 1c**). Increased southward reach of the relatively warm EAC has been suggested before as a mechanism to warm the SWP throughout the hot early Eocene (Hollis et al., 2012; Hines et al., 2017). Model simulations (using modern boundary conditions) indicate that a wind-driven strengthening and further southward extent of the EAC is expected under conditions of enhanced global warmth, as part of intensification of the southern midlatitude circulation (Cai et al., 2005). Indeed, observational data indicate a strengthening of the South Pacific Gyre over the past six decades, including a southward extent of the EAC at the expense of the Tasman Front (Hill et al., 2008, 2011). Similarly, SST anomaly reconstructions over the peak interglacial Marine Isotope Stage 5e (~125 ka) indicate intensification of the EAC to offshore Tasmania (Cortese et al., 2013). Possibly a similar atmospheric and oceanographic response to global warming occurred during MECO.

## 5.2 Drivers of dinocyst assemblage change in the Tasmanian Gateway

Unconstrained ordination using a unimodal (DCA) or non-metric (NMDS) model shows that the primary variability in the dinocyst assemblage at Site 1170 is governed by *E. dictyostila* and follows SST quite closely (**Figure 4a, Supplementary Figure 3**), suggesting that the abundance of *E. dictyostila* responds to temperature. The first NMDS and DCA axes are virtually identical, with DCA1 accounting for 33 % of the variance in the dataset. Both DCA2 (accounting for 17 %) and MDS2 contrast *D. antarctica* and *T. pelagica* at one end of the axis with *Vozzhennikovia* spp. at the other end. Ordination results of the MECO and the surrounding interval at Site 1172 are closely comparable with those of Site 1170 (**Figure 4b, Supplementary Figure 3**). At Site 1172, the abundance of *E. dictyostila* also controls the first axis (DCA1 accounting for 42 % of the variance), and the second axis (accounting for 17 %) places *D. antarctica* and *T. pelagica vs. Vozzhennikovia* spp.

No clear patterns in biogeographic or coastal proximity grouping emerge from the ordination results of Site 1170 and Site 1172. However, unconstrained ordination of the combined dinocyst assemblages from Site 1170, Site 1172, the Otway Basin, and Hampden Beach results in a biogeographic separation on the first axis (DCA1 accounting for 77 % of the variance, DCA2 accounting for 38 %) (**Figure 5**). DCA1 and MDS1 separate the Site 1170 and Site 1172 assemblages from the Otway Basin and Hampden Beach assemblages, as these axes separate endemic (and some cosmopolitan) taxa on the left *vs.* mid-/low-latitude (and some cosmopolitan) taxa on the right. The second axis further separates Site 1170 from Site 1172.

The role of temperature in determining assemblage variability at Site 1170 is further supported by constrained ordination (CCA), in which the first axis has high explanatory power (~67 % of the total accounted variance by the environmental variables), and has $TEX_{86}$ as the dominant component (**Figure 4c**; environmental variables as time series in **Supplementary Figure 4**). Therefore, although no peak of low-latitude species characterizes the MECO at Site 1170, the ordination analyses suggest that the dinocyst assemblage as a whole, and in particular *E. dictyostila*, responded to temperature change during MECO.

Taken together, these results confirm previous evidence that once a surface-oceanography-tracking plankton community has become established, relative abundance changes within the community correspond closely with changes in SST (Bijl et al., 2011). In the modern ocean, phytoplankton distribution patterns are driven by the interplay of passive transport by surface currents and temperature selection (Thomas et al., 2012; Hellweger et al., 2016). A similar dual selection mechanism seems to have affected the middle Eocene dinocyst assemblages in the region. Regional surface-ocean circulation determined which assemblage was established and where. This spatial pattern (**Figure 5**) could change over tectonic timescales as paleogeography changed (Bijl et al. 2011). Dominance shifts and variability within these assemblages were then driven by superimposed surface-ocean changes (such as in temperature), which typically occur on shorter timescales.

**5.3 Massive middle Eocene dinocyst productivity on the South Tasman Rise**

At the South Tasman Rise, MECO sediments are not only characterised by rapid sedimentation rates (in the order of 10s of cms per kyr according to our age models; compare Section 4.1.3), but also by high concentrations of dinocysts (**Figure 6**). High sedimentation rates are readily explained by the location of Site 1170 as a middle Eocene depocenter affected by rifting between Australia and Antarctica and associated subsidence (Exon et al., 2004). However, the extraordinarily high dinocyst concentrations are more difficult to explain. They are 100–1,000 times higher than in the studied strata from the Otway Basin and Hampden Beach. They also stand out when compared to other time intervals and settings where high dinocyst concentrations are expected and found. Specifically, they are about an order of magnitude higher than those typically found in Mediterranean sapropels (e.g., Sangiorgi et al., 2006; van Helmond et al., 2015; Zwiep et al., 2018), Cretaceous Oceanic Anoxic Event 2 shelf sediments (van Helmond et al., 2014) and the Holocene Adélie drift underlying a highly productive polynya system (Hartman et al., 2018).

The high sedimentation rates and silty claystone facies make it unlikely that high dinocyst content was the result of sediment starvation and/or winnowing, respectively. Furthermore, such conditions would also have facilitated oxidation and degradation of organic-walled palynomorphs, while they are instead well-preserved and abundant. Therefore, these high

concentrations seem to represent extreme dinocyst productivity and/or preservation. Enhanced sediment accumulation rate by itself facilitates burial of organic matter, in particular through adsorption of organics to clay minerals (Berner, 2006; Hedges and Keil, 1995), so preservation could have played a role. However, total organic carbon (TOC) contents are not extremely high (mean: ~1 % over the studied interval), the sediment is well bioturbated, and there is no significant correlation between dinocysts/gram and shipboard TOC contents, uranium contents or magnetic susceptibility

(**Supplementary Figure 5**), which suggests preservation was not the driving factor leading to high dinocyst concentrations. Rather, surface ocean productivity may have been elevated. The relatively low diversity of the dinocyst assemblages in combination with the high dominance of a single taxon (*Enneadocysta dictyostila* in the MECO interval) suggests a generally eutrophic setting that could have been characterised by seasonal plankton blooms. Notably, in several records from the Paleocene-Eocene Thermal Maximum (Harding et al., 2011; Sluijs et al., 2011; Frieling et al., 2018b), and a record from

Oceanic Anoxic Event 2 at Bass River (van Helmond et al., 2014), highest concentrations of dinocysts reach 10,000–100,000 cysts per gram sediment, and also correspond to low diversity - high dominance assemblages, suggestive of dinoflagellate blooms. Dinocysts deriving from heterotrophic dinoflagellates are present at Site 1170, but not in high abundance (**Supplementary data**). This indicates that primary production of dinoflagellate prey species such as diatoms (Jeong, 1999) was not necessarily high during the studied interval. Combined, the above suggests that high surface-ocean dinoflagellate-

based productivity combined with increased production of resting cysts, was the most likely cause of rapid accumulation of dinocysts at Site 1170, with possible secondary roles for sediment transport and organic matter preservation. Indications why conditions in the middle Eocene Tasmanian Gateway would have been extremely favourable for dinoflagellate or dinocyst production are, however, yet lacking.

**5.4 Southeast Australian vegetation during the MECO**

The middle Eocene sporomorph assemblages from the Latrobe-1 borehole are generally similar to those identified in previous studies (Macphail et al., 1994; Greenwood et al., 2003; Hill, 2017), but also include a small proportion of meso–megathermal components. Although the small amount of analysed samples prohibits a description of pre-, syn-, and post-MECO vegetation, the assemblages from Latrobe-1 reveal that this middle Eocene vegetation of coastal southeast Australia consisted of a mosaic of mesothermal rainforest flora. These forests were dominated by warm temperate angiosperms

Casuarinaceae (*Gymnostoma*), *Austrobuxus/Dissilaria* and Proteaceae as shrubs and trees, with rare (paratropical) tree palms (Arecaceae) and cycads (Cycadophyta). Overstorey elements included *Nothofagus* sg. *Brassospora* and gymnosperms of the Araucariaceae and Podocarpaceae (*Podocarpus, Dacrydium* and *Lagarostrobos*). The low abundance of saccate

Podocarpaceae pollen, *i.e.*, pollen with high transport capability that are often overrepresented in pollen assemblages, suggests that these taxa were not a major part of the coastal vegetation in the lower interval. Together with small trees and shrubs, ground ferns (Gleicheniaceae and Osmundaceae) and tree ferns (Cyatheaceae) occupied the understorey in these rainforests. While the MECO marker dinocyst species *Dracodinium rhomboideum* was recorded in two of four studied samples, further stratigraphic constraints are lacking. Future regional pollen studies focussing on the Nirranda Group might therefore elucidate whether the relatively warm-loving flora described here was restricted to the MECO interval, or to a broader interval of middle-late Eocene "background" conditions.

**5.5 Sea-level rise during the MECO?**

Glacial eustacy might have played a minor role in middle Eocene sea level changes (Dawber et al., 2011), but Earth's polar regions are generally thought to have been largely ice-free during that time. Thus, accommodation space on the continental shelves (on time scales of $10^6$–$10^7$ years) was primarily determined by the interplay of thermal expansion of seawater, sediment supply and basin subsidence. In general, warm and wet early Eocene conditions are expected to have saturated passive continental shelves, resulting in relatively flat and shallow shelf platforms (Sømme et al., 2009). In the Otway Basin, sediments of middle Eocene age (basal Nirranda Group) overlie a large unconformity at the top of early Eocene sediments of the Wangerrip Group (e.g., Krassay et al., 2004). These middle Eocene sediments were deposited during the Wilson Bluff transgression, which is recognised throughout southeast Australia (Holdgate et al., 2003; McGowran et al., 2004) and has been linked to a major transgressive phase in the Indo-Pacific (the Khirthar transgression) (Jauhri and Agarwal, 2001; McGowran et al., 2004). While there is seismostratigraphic evidence for regional tectonic rifting, normal faulting and subsidence during the Paleocene and early Eocene in southeast Australia (Krassay et al., 2004; Close et al., 2009), it is unknown when subsidence terminated, and renewed. Additionally, a progressive decrease in terrigenous sediment supply as the Australian hinterland aridified throughout the Eocene might have affected accommodation space (Sauermilch et al., 2019). Whatever the relative contributions of these mechanisms, the hiatus between the Wangerrip Group and the Nirranda Group suggests no or negative accommodation space by the end of the early Eocene (51 Ma) or later. The renewed drowning of the continental shelf, as reflected in the Wilson Bluff transgression, seems unlikely to be related to slow and continuous basin subsidence. Instead, ocean warming during the MECO may have raised global average sea level by several meters by thermal expansion, while warmer and wetter regional climate could have increased sediment supply. The resumption of sedimentation accumulation above the top Latrobe unconformity has been previously dated to between 44 and 40 Ma (Holdgate et al., 2003; McGowran et al., 2004). Based on our new dinocyst-based age constraints, it is likely that the sediments overlying the Wangerrip group are close to the MECO in age, suggestive of a causal link between the Wilson Bluff transgression and MECO warming. A similar timing of renewed sedimentation occurred in the Schöningen section in the North German Basin, where the transgressive, fully marine Annenberg Formation unconformably overlies the Lutetian coal-bearing Helmstedt Formation (Riegel et al., 2012). The Annenberg Formation has been assigned an age around the

MECO (Gürs, 2005), possibly ~41 Ma (Brandes et al., 2012). Based on a compilation of New Jersey coastal plain sections, a highstand (sequence E8) is also interpreted at ~41–40 Ma (Browning et al., 2008).

Sea-level rise and warming during the MECO may have accommodated increased burial of biogenic carbonate on continental shelves, explaining a reduction in carbonate burial in the deep sea (Sluijs et al., 2013), along with a diminished silicate weathering feedback (Van der Ploeg et al., 2018). However, it should be noted that the above inferences regarding global sea-level rise during the MECO are tentative. Although these transgressive surfaces all have an age around the MECO, current age control is not nearly sufficient to correlate them to MECO with certainty. A dating accuracy of ≤100,000 years would be required for these transgressive surfaces to indicate their relationship to MECO warming, which is presently not available. It is therefore crucial to improve these constraints in order to assess the potential influence of sea-level change on the carbon cycle during the MECO.

## 6        Conclusions

Comparison of plankton and sea-surface temperature patterns during the MECO above the South Tasman Rise indicate that while dinocyst assemblages responded to surface-water warming, the acme in cosmopolitan taxa above the East Tasman Plateau at peak MECO is not mirrored at the STR. This implies either eastward throughflow through the northern portion of the Tasmanian Gateway, or a southward extension of the EAC during the zenith of MECO warmth. This illustrates how profoundly surface-ocean currents can respond to external climate forcing in these regions of the Southern Ocean. Terrestrial palynomorph assemblages indicate a warm temperate rainforest with some paratropical elements grew along the southeast Australian margin during the MECO. Finally, we suggest that the southeast Australian Wilson Bluff Transgression may be related to sea-level rise during the MECO, but improvement of the available age constraints is necessary to establish a possible causal link.

**Acknowledgements**

This research used samples and data provided by the International Ocean Discovery Program (IODP) and its predecessors. This work was carried out under the program of the Netherlands Earth System Science Centre (NESSC), financially supported by the Dutch Ministry of Education, Culture and Science. This study was made possible by the Netherlands Organisation for Scientific Research (NWO) grant number 834.11.006, which enabled the purchase of the UHPLC-MS system used for GDGT analyses. Funding was provided by the Australian IODP office and the ARC Basins Genesis Hub (IH130200012) to SJG. We thank Natasja Welters, Jan van Tongeren and Arnold van Dijk (Utrecht University Geolab) for analytical support. We thank the reviewers Severine Fauquette, Chris Hollis and G. Raquel Guerstein for their constructive reviews of the initial version of the manuscript.

**Main figures**

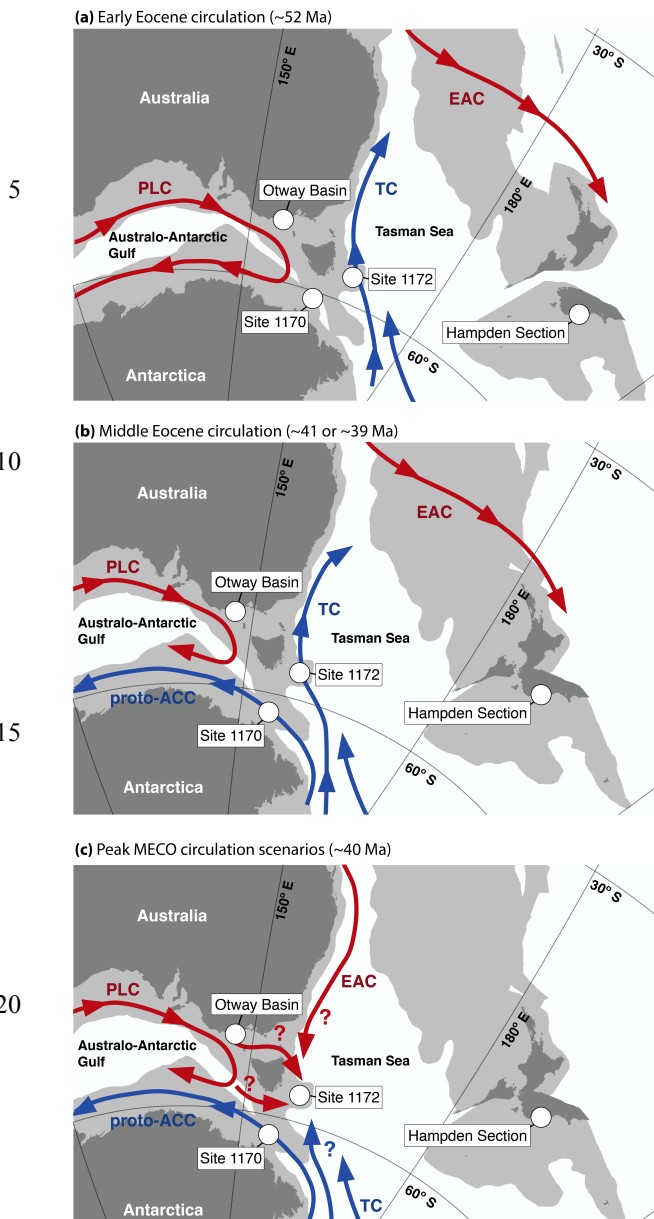

**(a)** Early Eocene circulation (~52 Ma)

**(b)** Middle Eocene circulation (~41 or ~39 Ma)

**(c)** Peak MECO circulation scenarios (~40 Ma)

**Figure 1. Generalised Eocene surface ocean circulation patterns in the southwest Pacific Ocean. (a)** Generalised early Eocene (~52 Ma) circulation. **(b)** Generalised middle Eocene circulation pre-MECO (~41 Ma) and post-MECO (~39 Ma). **(c)** Generalised peak MECO (~40 Ma) circulation. Maps constructed with GPlates, using Torsvik et al. (2012) paleomagnetic rotation frame and Matthews et al. (2016) continental polygons and coastlines for 52 Ma (a) and 40 Ma (b and c). Note that, within this rotation frame, there is uncertainty on the drawn paleolatitudes. For example, Site 1170 is drawn at 61.6 °S at 40 Ma, but the uncertainty margins on this are between 58.76 °S and 64.55 °S (van Hinsbergen et al., 2015). Currents drawn after reconstructions by Bijl et al. (2011, 2013b, 2013a) and this study. EAC = East-Australian Current; PLC = Proto-Leeuwin Current; TC = Tasman Current; proto-ACC = proto-Antarctic Counter Current.

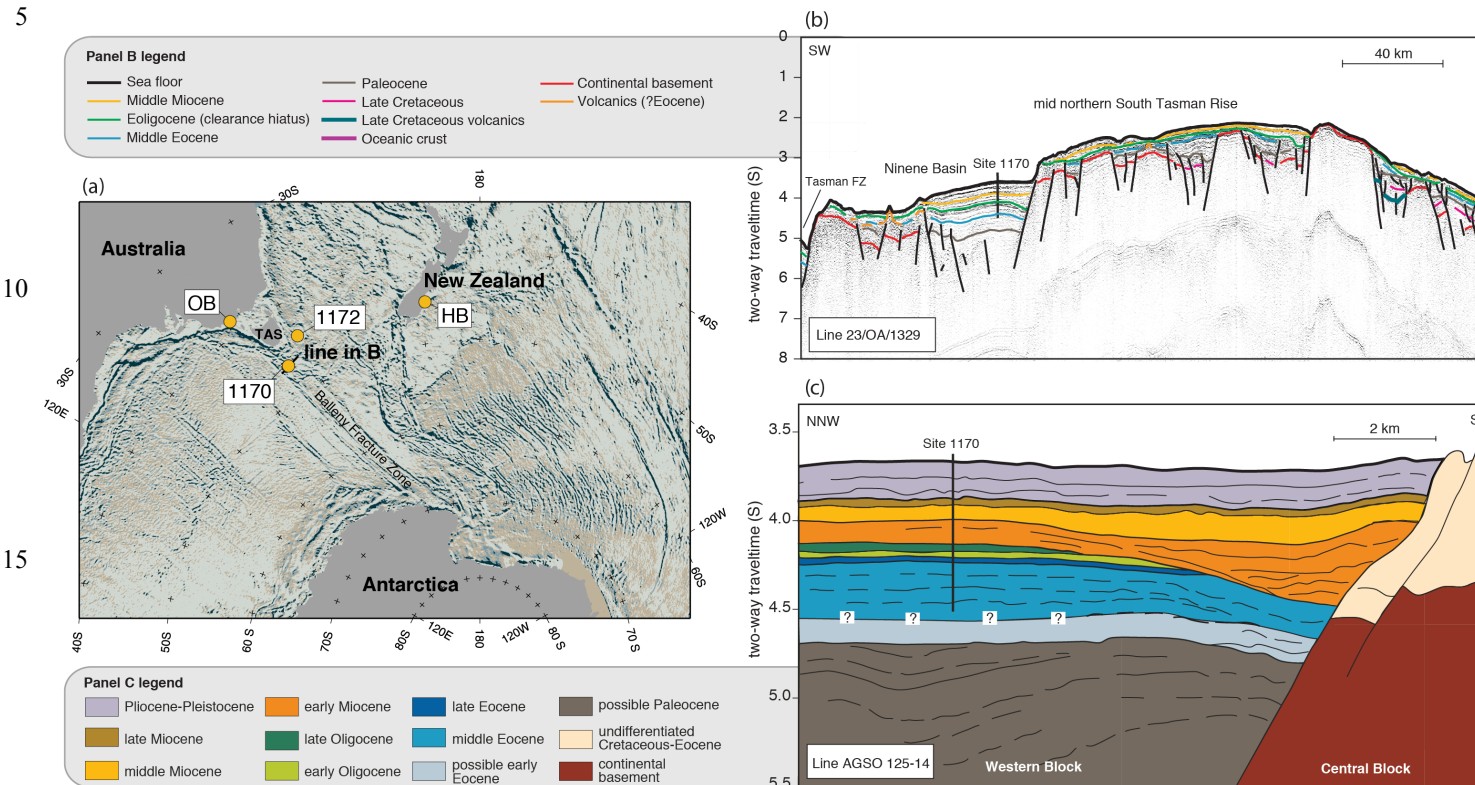

**Figure 2. Tectonic setting of ODP Site 1170 and other studied sites (a)** Present-day map of the Australo-Antarctic sector of the Southern Ocean, with present-day locations of sites and sections used in this study as yellow circles (ODP Site 1170; ODP Site 1172; OB, Otway Basin; HB, Hampden Beach). NW-SE structural trends mark the direction of rifting between Australia and Antarctica, clearly visible in the (labelled) Balleny Fracture Zone. Seismic profile line 23/OA/1329, as shown in panel b, drawn as thick black line. Seismic profile line AGSO125-14 not drawn due to its small scale. Adapted from Bijl et al. (2013b) and Cande and Stock (2004). **(b)** Interpreted SW-NE seismic profile (line SO36-58) across the South Tasman Rise, illustrating the Site 1170 location in a graben structure. Profile and interpretation adapted from Hill and Moore (2001). **(c)** Interpreted NNW-SSE seismic profile (line AGSO125-14) across the South Tasman Rise, including Site 1170, illustrating laterally thinning seismic layers of interpreted middle Eocene age. Profile and interpretation adapted from Exon et al. (2001).

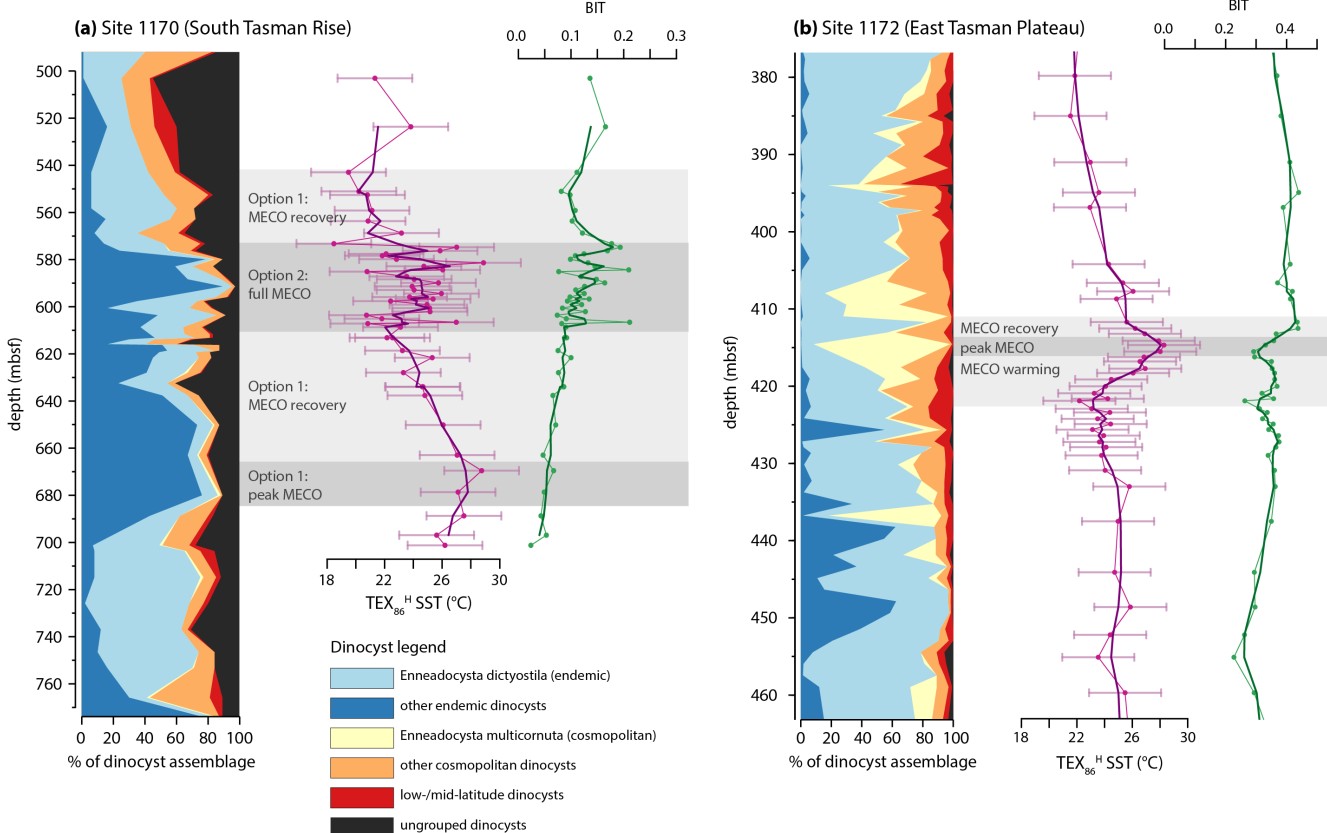

**Figure 3. Dinocyst and temperature data from ODP Site 1170 and Site 1172. (a)** Left: cumulative silhouette plot of relative abundances of dinocyst biogeographic groups at Site 1170. Especially for the younger part of the 1170 record, a high proportion of specimens of the genus *Deflandrea* could not be identified to the species level, causing the high abundance of the "others" group. Middle: $TEX_{86}^H$-based SST (in degrees celsius) in pink, with 5 point moving average in purple. Error bars are combined calibration and analytical error (1 s.d.) (± 2.6 °C). Right: BIT in green, with 5 point moving average in dark green. Plotted against depth in metres below seafloor on the vertical axis. Gray horizontal bars visualize the two different options for extent of the MECO, as presented in paragraph 4.1.3. **(b)** Same as a, but for Site 1172. Data from (Bijl et al., 2010, 2011).

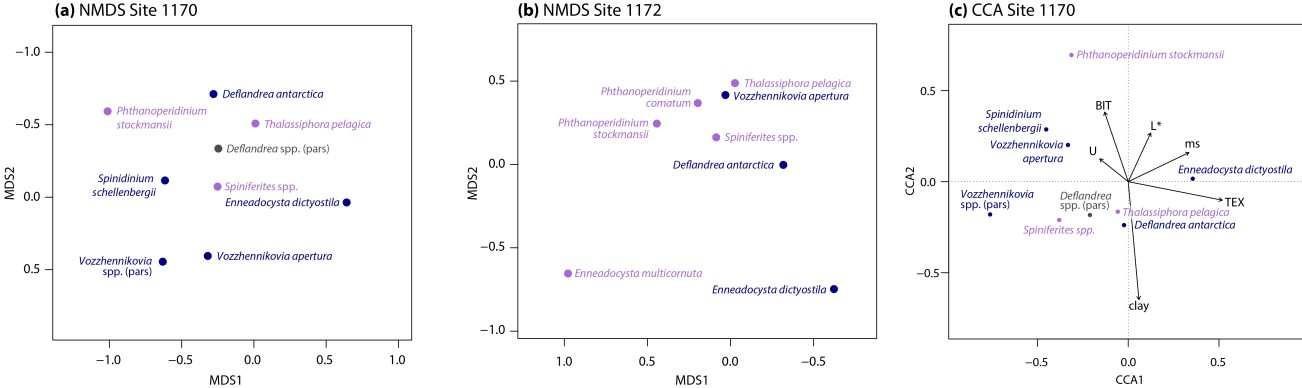

**Figure 4. Ordination results. (a)** Nonmetric multidimensional scaling ordination diagram for the dinocyst assemblage data of Site 1170. Species scores as circles, colour-coded by biogeographic affinity (purple, cosmopolitan; blue, endemic; grey, not assigned). **(b)** Nonmetric multidimensional scaling ordination diagram for the dinocyst assemblage data of Site 1172. Species colour-coding as in panel a. **(c)** Canonical correspondence analysis ordination diagram for the dinocyst assemblage data of Site 1170. Species colour-coding as in panel a. Abbreviations are as follows: BIT, BIT index; clay, clay fraction (%); L*, CIELAB lightness variable; ms, magnetic susceptibility; TEX, TEX$_{86}$; U, uranium content. Total amount of inertia in species data explained by environmental variables is 34%. For visual clarity, only the most abundant taxa (taxa that occur in >10% of the samples, have a mean relative abundance >1%, and have a maximum relative abundance of >5%) are shown in all three panels. Ordination plots showing all taxa are provided as Supplementary Figure 3.

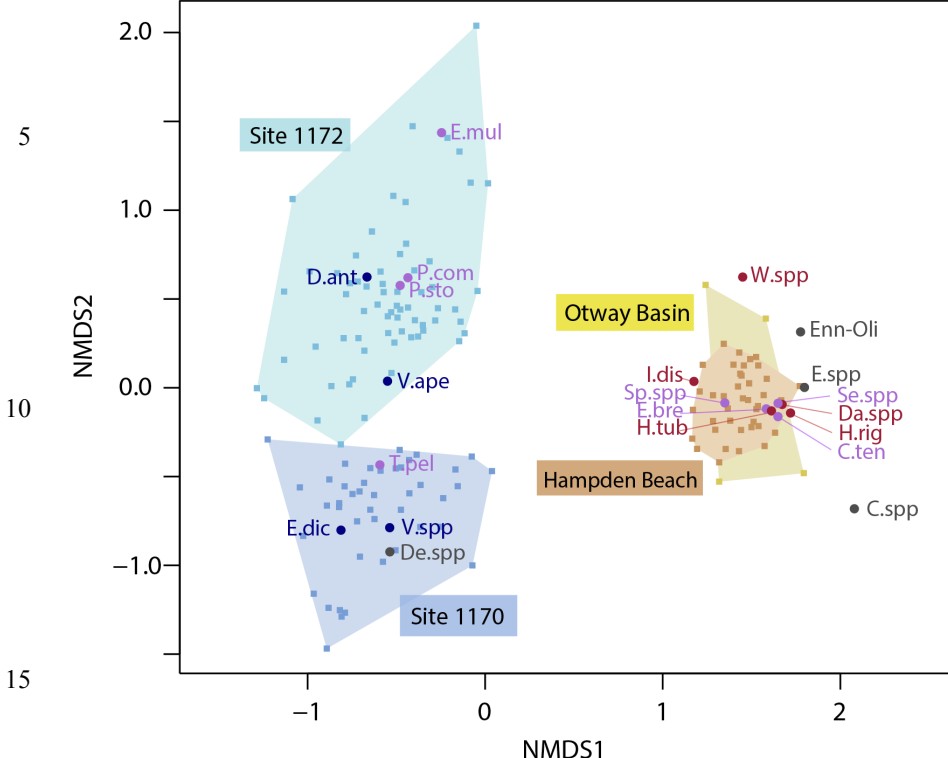

**Figure 5. Nonmetric multidimensional scaling ordination diagram for the combined dinocyst assemblage data of Site 1170, Site 1172, Hampden Beach and Otway Basin.** Species scores as circles, colour-coded by biogeographic affinity (red, mid-/low-latitude; purple, cosmopolitan; blue, endemic; grey, not assigned). Samples scores as squares, colour-coded by location (light blue, Site 1170; dark blue, Site 1172; orange, Hampden Beach; yellow, Otway Basin), with shading connecting same-location samples. Abbreviations are as follows: C.spp, *Corrudinium* spp. (pars); C.ten, *Cribroperidinium tenuitabulatum;* Da.spp, *Dapsilidinium* spp.; D.ant, *Deflandrea antarctica*; De.spp, *Deflandrea* spp.; E.bre, *Elytrocysta brevis*; E.dic, *Enneadocysta dictyostila;* E.mul, *Enneadocysta multicornuta;* Enn-Oli, *Enneadocysta-Oligosphaeridium* intermediate; E. spp, *Enneadocysta* spp. (pars); H.rig, *Hystrichokolpoma rigaudiae;* H.tub, *Hystrichosphaeridium tubiferum;* I.dis, *Impagidinium dispertitum;* P.com, *Phthanoperidinium comatum;* P.sto, *Phthanoperidinium stockmansii;* Se.spp, *Senegalinium* spp. (pars); Sp.spp, *Spiniferites* spp. (pars); T.pel, *Thalassiphora pelagica*; V.ape, *Vozzhennikovia apertura;* V.spp, *Vozzhennikovia* spp. (pars); W.spp, Wetzellioids. For visual clarity, only the most abundant taxa are shown.

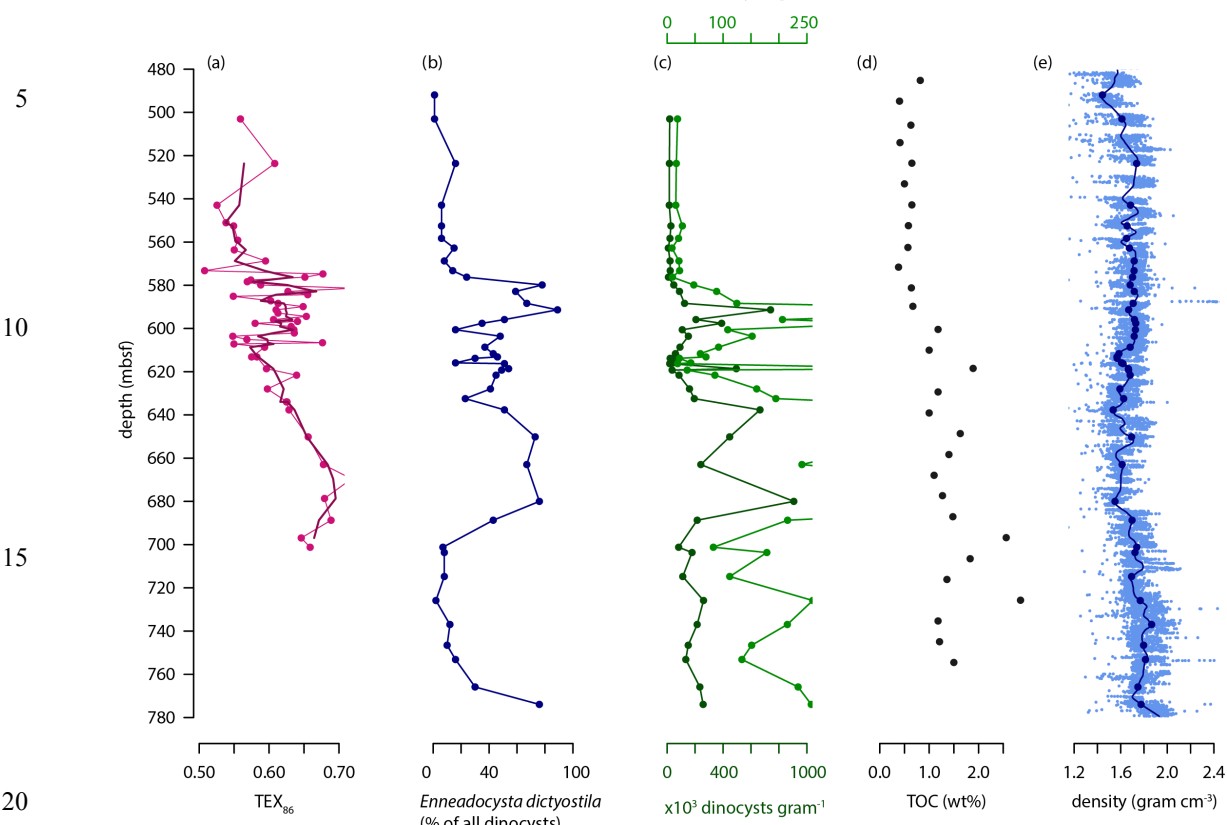

**Figure 6. Selected proxy records over the MECO interval of Site 1170**, plotted against depth in metres below sea level. **(a)** TEX$_{86}$ (pink dots and line), with three-point moving average (purple lines). **(b)** Relative abundance of *Enneadocysta dictyostila* (percentage of total dinocyst assemblage; dark blue dots and line). **(c)** Dinoflagellate cyst content (cysts per gram of dry sediment; two different scales shown for visual clarity in dark green and light green). **(d)** Total organic carbon (weight percentage; black dots) (from Exon et al., 2001). **(e)** GRA sediment density in (gram per cubic centimetre; light blue dots original data; dark blue line LOESS fit; dark blue dots interpolated LOESS fit to depth of dinocyst samples) (from Exon et al., 2001).

**Supplementary figures**

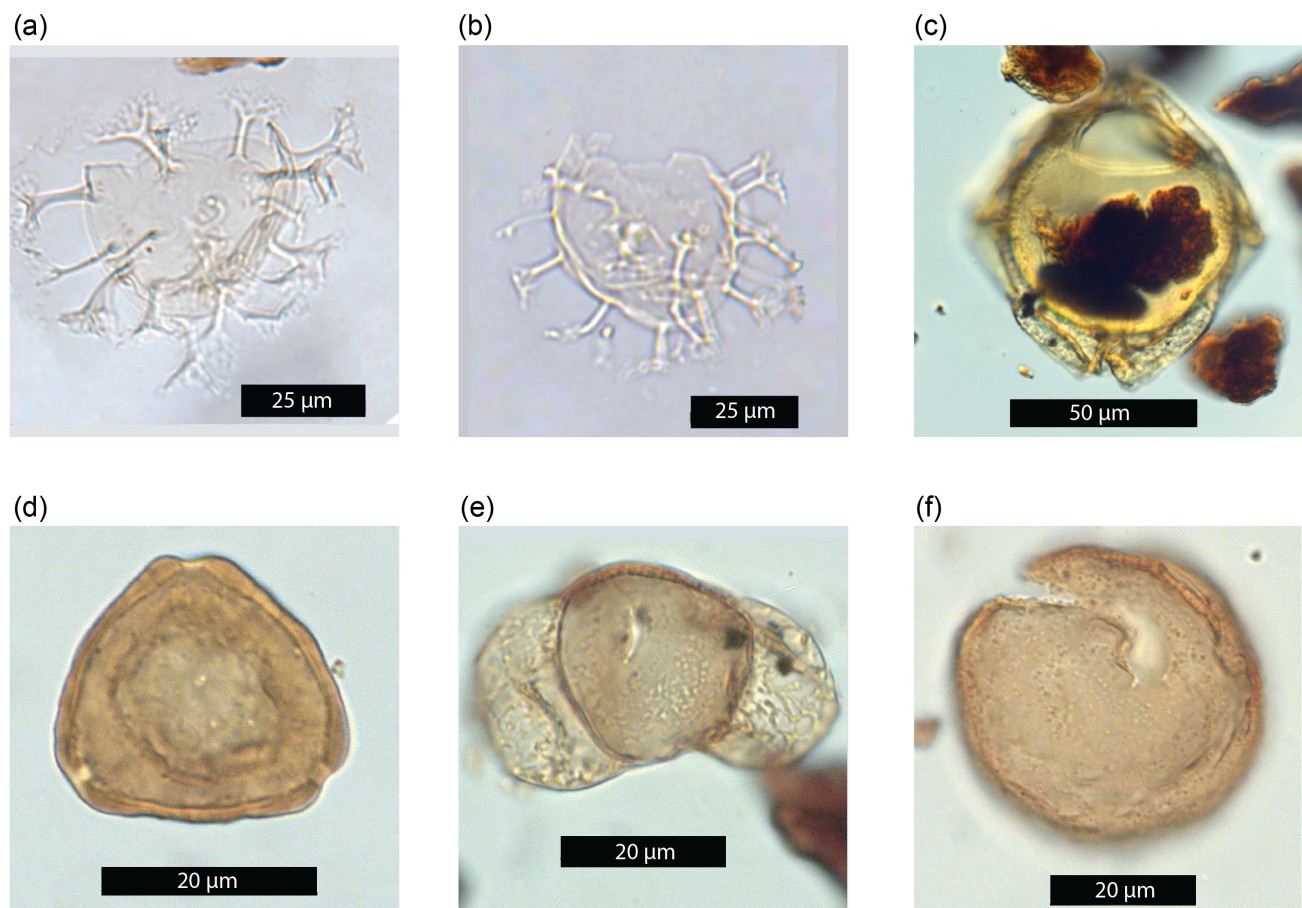

**Supplementary Figure 1.** Plate with light microscope images of relevant palynomorphs encountered in this study. (a) Dinocyst *Enneadocysta dictyostila* from sample 1170D 21R 4W 85-87 cm (EFC unavailable), scale bar 25 μm. (b) Dinocyst *Enneadocysta multicornuta* from sample 1170D 23R 2W 85-87 cm (EFC unavailable), scale bar 25 μm. (c) Dinocyst *Dracodinium rhomboideum* from sample L86 slide 2 (EFC E48.1), scale bar 50 μm. (d) Pollen *Myricipites harrisii* from sample L85 slide 1 (EFC J15.4), scale bar 20 μm. (e) Pollen *Podocarpidites ellipticus* from sample L84 slide 1 (EFC G19.2), scale bar 20 μm. (f) Pollen *Dilwynites granulatus* from sample L87 slide 1 (EFC J16.1), scale bar 20 μm.

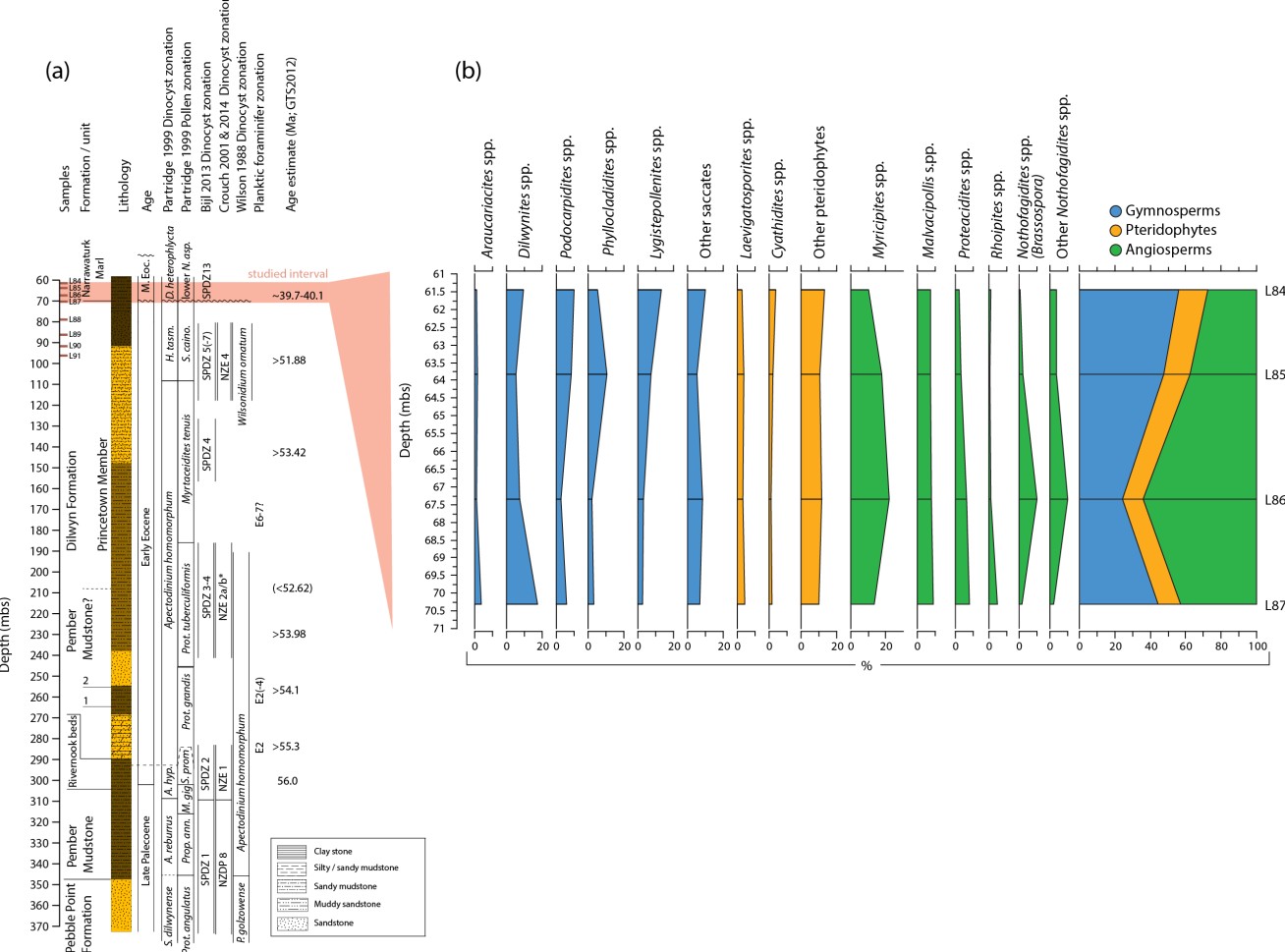

**Supplementary Figure 2.** (a) Stratigraphy of the Latrobe-1 borehole, including studied samples, lithologic units, lithology and age constraints based on microfossil biostratigraphy. The interval analysed in this study is highlighted in pink. Figure adapted from (Frieling et al., 2018a). (b) Relative abundances of representative sporomorph taxa in samples L84-L87 from the Latrobe-1 borehole, in percentage of total sporomorph assemblage.

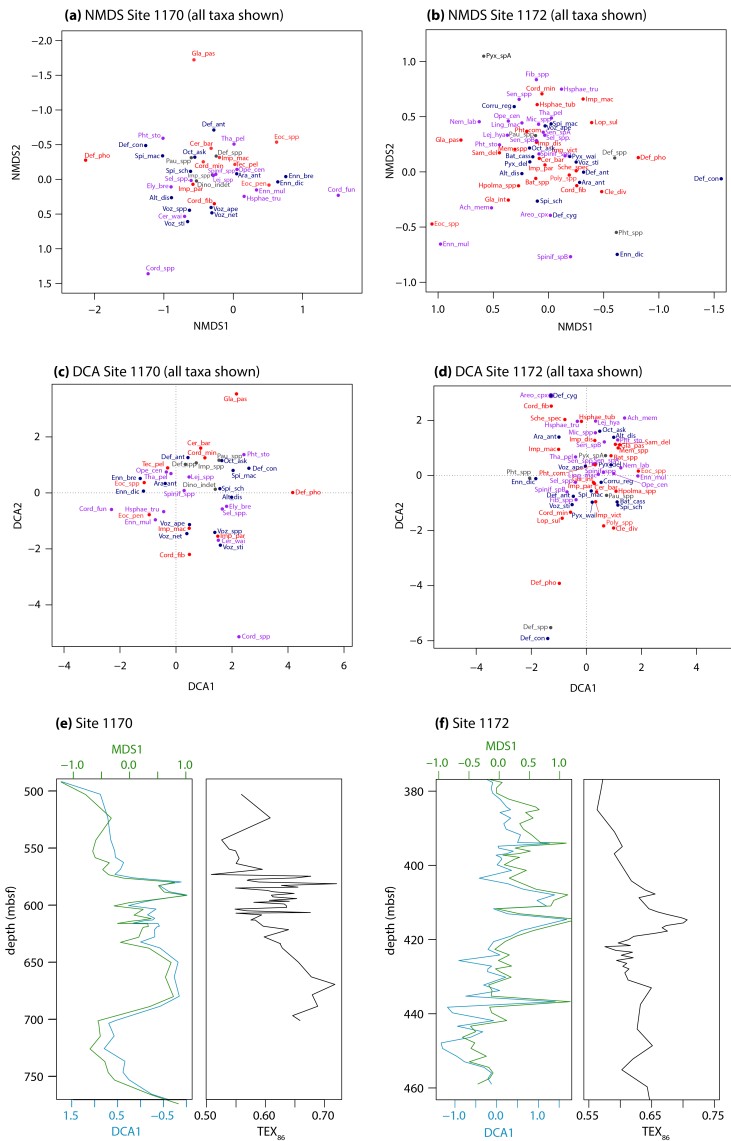

**Supplementary Figure 3.** Additional ordination results. Nonmetric multidimensional scaling (NMDS) ordination diagram for the dinocyst assemblage data of Site 1170 **(a)** and Site 1172 **(b)**. Detrended correspondence analysis (DCA) ordination diagram for the dinocyst assemblage data of Site 1170 **(c)** and Site 1172 **(d)**. Species scores in a-d as circles, colour-coded by biogeographic affinity (red, mid-low latitude; purple, cosmopolitan; blue, endemic; grey, not assigned). Full names for dinocyst abbreviations can be found in the Supplementary Datafile. First axis of DCA (blue) and NMDS (green) analysis of Site 1170 **(e)** and Site 1172 **(f)**, together with the respective TEX$_{86}$ records (black).

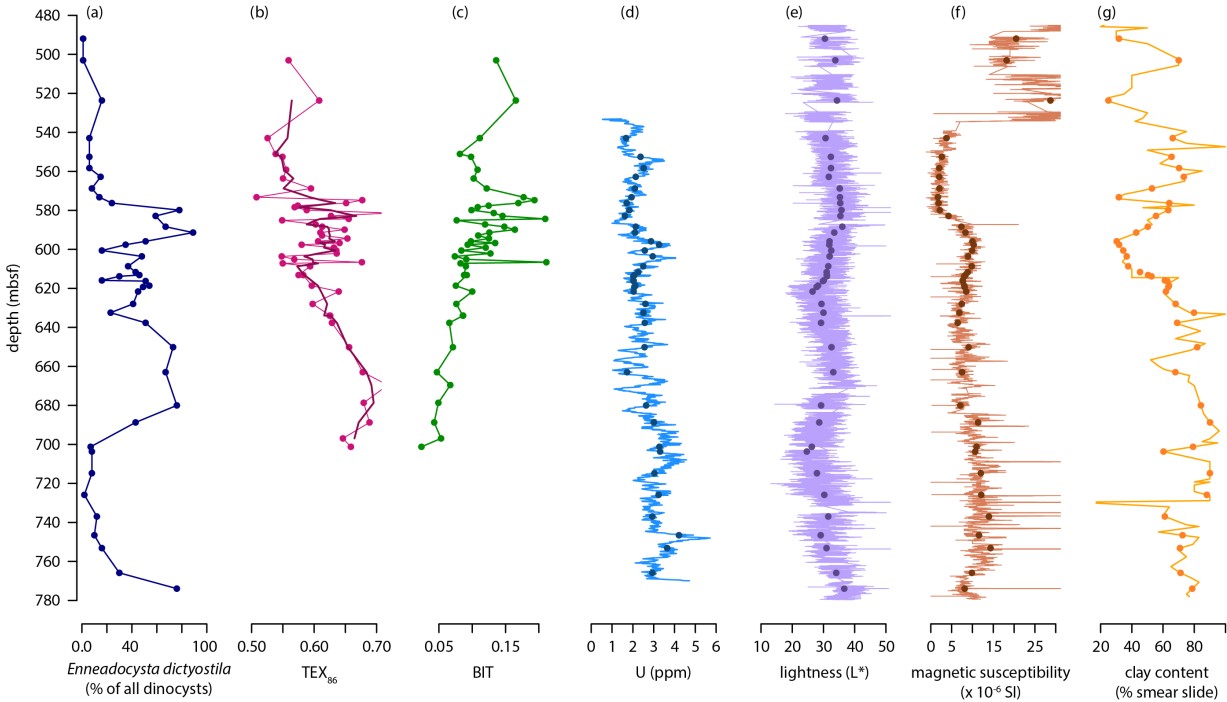

**Supplementary Figure 4.** Environmental proxy records over the MECO interval of Site 1170, as used in CCA analysis (b-g). Original data plotted as line, data interpolated to depth of dinocyst samples plotted as dots. Plotted against depth in metres below sea level. **(a)** Relative abundance of *Enneadocysta dictyostila* (percentage of total dinocyst assemblage; dark blue dots and line). **(b)** $TEX_{86}$ (pink dots and line), with three-point moving average (purple lines). **(c)** BIT (green dots and line). **(d)** Sedimentary uranium content (ppm; blue dots and line). **(e)** Spectrophotometric lightness (CIELAB L*; purple dots and line). **(f).** Core-measured magnetic susceptibility (x $10^{-6}$ SI). **(g).** Clay content (% of smear slide). Data in panels d-g from Exon et al. (2001).

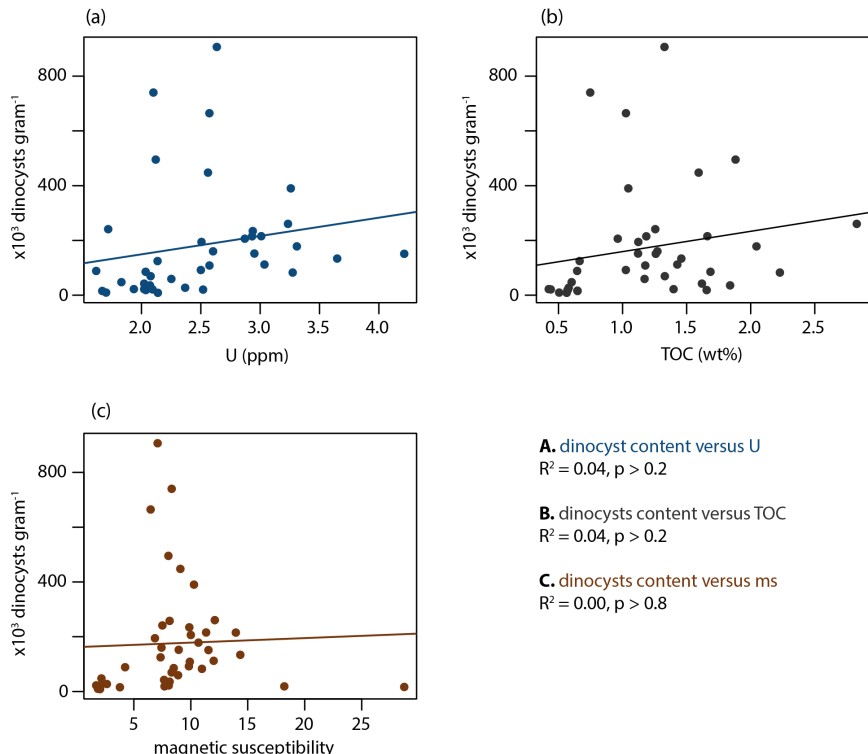

**A.** dinocyst content versus U
$R^2 = 0.04$, $p > 0.2$

**B.** dinocysts content versus TOC
$R^2 = 0.04$, $p > 0.2$

**C.** dinocysts content versus ms
$R^2 = 0.00$, $p > 0.8$

**Supplementary Figure 5.** Scatter plots and regression analysis of sedimentary dinocyst content as a function of selected proxy records of Site 1170, indicating no significant correlation. **(a)** Dinoflagellate cyst content (cysts per gram of dry sediment) against uranium content (ppm). **(b)** Dinoflagellate cyst content (cysts per gram of dry sediment) against total organic carbon content (weight percentage). **(c)** Dinoflagellate cyst content (cysts per gram of dry sediment) against magnetic susceptibility (x $10^{-6}$ SI). U, TOC and ms data from Exon et al. (2001).

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
