# Peer review of "Surface-circulation change in the Southern Ocean across the Middle Eocene Climatic Optimum: inferences from dinoflagellate cysts and biomarker paleothermometry"

_Climate of the Past, 2019_

## Referee Comment (RC1) · Severine Fauquette (Referee) · 10 May 2019

Cramwinckel et al. present a study of the surface-circulation change in the Southern Ocean during the Middle Eocene Climatic Optimum based on dinoflagellate cysts and biomarker paleothermometry. The manuscript submitted by Cramwinckel et al. is of good quality. This manuscript is within the scope of 'Climate of the Past' and is well written and structured. This will be a very useful paper on the MECO period in the

[Figure]

Southern Ocean that is not well known. This study will certainly help climate modelers who can introduce consistent boundary conditions into the models for this part of the globe. I recommend publication of their paper in Climate of the Past with however some revisions.

Main comments: - Authors should give, in supplementary data, a detailed description of the pollen morphology (apertures, ornamentation of the exine surface...) and some photos of the main palynomorphs (dinocysts and pollen grains). This period in this region is not well known and it could help for further studies. - The fossil pollen and spores should be identify, by comparing them to modern pollen grains, following current taxonomy of recent taxa, instead of using morphotaxa names. By applying such approach, pollen and spores may be assigned to family, genus, and sometimes, but rarely, even to species levels. Once they are botanically identified, their paleoecological requirements may be defined based on the modern taxa. This botanical approach allows reliable paleoenvironmental reconstructions, as described and done by Suan et al. (Geology, 2017) for the Early Ecocene of the Arctic Siberia. - Biostratigraphy: A table with the regional occurrences of the dinocysts could be interesting. - A simplified diagram with the stratigraphic log and the percentages of the main terrestrial palynological data of Latrobe-1 borehole is lacking.

Minor (technical) comments: - p5, L1: add a S to metre; L10: call figure 2a; L11: remove one "was not". - p6, L8: How many samples have been studied for this site?; L25: add "concentration" for the dinocyst content. - p10, L32: add a reference for the ages given by Lophocysta spp . - p14, L19: remove the "cf" in front of the reference Bijl et al.. - Figure 5, L17: add a "c" in the word dinocyst. - Figure 2, L21: in (a), it is not the bathymetry that is illustrated as there is no mention of the depth of the ocean.

---

## Referee Comment (RC2) · Chris Hollis (Referee) · 15 May 2019

**Chris Hollis (Referee)**

c.hollis@gns.cri.nz

Received and published: 15 May 2019

General comments This is an interesting and important study, comparing and contrasting dinocyst assemblage changes between ODP sites 1170 and 1172, one within the Australo-Antarctic Gulf and one in the SW Tasman Sea, during a time of major climate change in the middle Eocene. The study uses evidence from the assemblages to unravel the interplay of changes in ocean circulation due to tectonics and climate

changes. Additional sites and data are used to build the case for a significant regional response to the middle Eocene climatic optimum (MECO) – in terms of changes in plankton communities, terrestrial vegetation and sea level. The interpretations are reasonable in most cases but there are a few areas where the argument is weakened by over-interpretation of what the authors admit are ambiguous data. The key areas are: the definition of the MECO at Site 1170 based on the TEX86 record, which is clearly open to interpretation; the lumping together of cosmopolitan and low/mid latitude taxa, when the latter group is the one that is best able to signal the influence of the EAC and PLC; the lack of convincing evidence for the presence of the MECO in the Latrobe-1 borehole; and the very tenuous correlation of middle Eocene transgression to a purported MECO-related glacioeustatic event.

I have made numerous comments on these and other issues at the places they occur in the text.

However, there is a hidden gem in this dataset that I'm disappointed the authors appear to have overlooked. In our warming world, we are increasingly concerned about the ways ecosystems will be adversely affected by warmer oceans and changes in ocean circulation. For dinoflagellates there is the further concern of how toxic blooms may impact coastal fisheries. The authors provide a dataset that clearly shows the MECO in this region is linked to dramatic increases in the abundance of single species, analogous to present day blooms. And intriguingly, a species of one genus dominates at Site 1170 whereas another species of the same genus dominates at 1172. Even more intriguing, both species have short-lived blooms leading up to the MECO at 1172. Much of the paper simply combines the data for these two species with their respective biogeographic groups (cosmopolitan and endemic) but these two taxa clearly dominate these groups (as shown by DCA and NMDS) and it is certainly worth considering that the rise and fall of these two species is more directly related to local watermass conditions than to current transport. I'd like to know if there is any indication of EAC or PLC influence with E. multicornuta removed. And I'd like to see more discussion on the CPD
watermass conditions that might lead to monospecific blooms of these two species.

Specific comments/Corrections by page, line:

1, 20: I see the term "Tasman Gateway" or "Tasman Seaway" has been used in the literature but it's incorrect. The proper term is "Tasmanian Gateway", being the gateway between Tasmania and Antarctica (see any Leg 189 publication).

1, 22: ", including the organic walled cysts of dinoflagellates (dinocysts). I'd like to see a distinction made between dinoflagellates (plankton) and dinoflagellate cysts or dinocysts (fossil remains of the plankton)

1, 23: prefer "geographic" to "spatiotemporal" (here and elsewhere)

1, 24: "geographic" here is superfluous. And is it primarily controlled by tectonism? What about the rotation of the Earth? I wonder if this simplistic separation of tectonic and climatic controls is warranted or needed in an abstract? Sentence is awkward, so how about rephrasing: "The extent to which the climatic and tectonic controls on the distribution and composition of surface currents have influence the composition of fossil assemblages ...".

1, 26: This sentence is also a little awkward. "Indeed, the extent to which climate change affects oceanographic processes is still poorly understood"?

1, 29: Also, an awkward sentence. "trend, the Middle Eocene Climatic Optimum (MECO,  ${\sim}40$  Ma). This 500 kyr-long episode of global warming is unrelated to  $\ldots$ "

1, 31: "ocean's"; replace "only" with "alone"

- 2, 1: "our new results...", no hyphen between surface and ocean
- 2, 2: replace "southward" with "south"

2, 3: Explain how "warm temperate with paratropical elements" MECO assemblage differs from the general middle Eocene pollen assemblage?
2, 8: change "into" to "to"

2, 13: does "intermediate-deep" mean somewhere between upper and lower deep water or is it shorthand for "intermediate and deep", in which case this formulation is less ambiguous.

2, 15: None of these sites are close enough to the Antarctic margin to be sources of deep water and are all north of the 60S demarcation for the SO, using pmag reference frame (although noting the uncertainty).

2, 18: change "marine-based" to "sea" and, no, they are not supported by estimates for land temperatures from NLR approaches, which are in general closer to the modelled temperatures (add Pancost et al. 2013), so SST estimates are 5-10C warmer than models and LAT estimates.

2, 21: add comma after processes

2, 22: remove parentheses around global

2, 31: plural "changes". Lord Howe Rise is part of Zealandia so rephrase: "submerged parts of NW Zealandia..."

3, 1: that's a lot of potential effects but rather speculative. Suggest you keep it simple. "... should have affected ocean circulation in the region with likely impacts for global heat transport and climate."

3, 4: change "of" to "from"

3, 5: Change "Southern Ocean" to "SO".

3, 6: Rephrase: "... endemism are characteristic of a diverse range of fossil groups ..." (circum-Antarctic is tautological when you've already said Southern Ocean)

3, 9: here is where I'd prefer you to use "dinoflagellates". If you use cysts here, you really also need to use frustules for diatoms and tests for forams and rads. Personally,
I don't think you need to use "dinocyst" at all, but certainly should not be used when you are talking about plankton as opposed to assemblages in sediment.

3, 12: Query use of "cosmopolitan". This is unconventional usage. Cosmopolitan means found everywhere, so hard to see why this group signals the influence of the PLC or EAC.

3, 13: NZ is not in the Tasman Sea. It is east of it.

3, 26: change "biogeographical patterns" to "biogeography"

3, 27: why the "cf."?

3, 28: Why is "orbital scale" mentioned? Is it relevant? Why the "cf."?

3, 32: Why is deep ocean warming described as "transient" and surface-water warming described as "widespread"

- 3, 34: be a little more specific than "global perturbations"
- 4, 2: low-latitude and cosmopolitan are not the same thing.
- 4, 3: change "outstanding" to "unresolved"

4,5: Sentences in this paragraph from "In addition . . ." to end of paragraph should come before the description of the dinocyst assemblages. These sentences are part of the general description of the MECO.

4, 8: The two factors mentioned do not "imply" a volcanic explanation. Revise this sentence and provide a reference for the volcanic carbon hypothesis.

- 4, 11: Last sentence of paragraph is poorly worded. Revise.
- 4, 25: Revise: "in the 2-3 km-deep and 50 km-wide Ninene Basin".
- 5, 18: Delete "interval"; no hyphen between shallow and marine, as for 5, 21.
- 5, 31: Sentence doesn't make sense. What covers the unconformity and overlies basal
Nirranda Group?

5, 32: "Latrobe-1 borehole"

6, 2: change "overlying" to "underlying"; What's the age of the Dilwyn Fm?

6, 11: Elsewhere in text it is referred to as Hampden section. Be consistent. Why no mention of the work on the rest of the Eocene / Paleogene section (e.g. Morgans, 2009; Hollis, et al., 2012; Inglis et al., 2015)

6, 12: missing comma after "...E)"

6, 13: "end-member" is not the right word. How about "analysed to identify influences from the TC or EAC in the middle Eocene prior to the MECO".

6, 28: lower case "s" for section.

7, 2: 50 and 90 are normally seen as too few for robust statistical analysis.

7, 5: and identified to what taxonomic level?

7, 14, 16: Key problem issue for this paper. Definition of "cosmopolitan" is ambiguous and not in line with convention: cosmopolitan = found everywhere. I recommend you use only low and mid-latitude taxa as your guide to PLC and EAC influence.

7, 27: Again, ambiguous terminology. Your example is not of a taxon with unknown biogeographic affinities, but with conflicting biogeographic affinities.

9, 7: What is meant by "spatial"? Lateral? Geographic might be a better term.

9, 24: U is not a direct proxy for TOC.

9, 26: Change "like" to "As with".

- 10, 2: Change "for" to "of".
- 10, 5: Change "dinocysts" to "assemblage"

CPD
- 10, 7: Can low salinity be consistent with low BIT?
- 10, 9: Change "most dominant" to "most abundant".
- 10, 12: Differentiate cosmopolitan from low/mid latitude.
- 10, 13: What does "a.o." mean?
- 10, 20: delete "at this site"; redundant.
- 10, 23: Provide error values for SST estimates and show on Fig. 3.

11, 8: "Precarious" is the wrong word, but a good choice nevertheless, because the whole interpretation of this section is precarious due to the subjective way the SST record has been interpreted. This is only one possible interpretation. Another is that the warming at 670 m precedes the MECO and perhaps can be correlated with the broad peak around 440 m at 1172. Thus, the MECO is the interval between 5570 and 600 m at 1170. This shorter duration is consistent with the biostrat and would mean that the cyst accumulation rate is not so untenably high. Both options should be considered.

11, 16: Poorly worded. "sufficient numbers of dinocysts were encountered for counts of 50-100 specimens to be undertaken. Other marine palynomorphs such as prasino-phytes and acritarchs, were rare/common(?)"

11, 31: Revise sentence beginning "Furthermore..." to "Cycadopites ... are also present but rare.

12, 1. Simultaneously is the wrong word. Delete. The abundance of Dilwynites, Protea... also decrease towards the top of the borehole.

12, 17. Very poorly worded but crucial sentence. The FO of this species is said to be at 40 Ma. When is the LO? It can only be used to define the MECO if it's restricted to the MECO. I conclude from the biostrat presented that the interval may include the MECO but equally may be younger (anywhere between 40 to 35.95 Ma).
12, 24: Differentiate cosmopolitan from low/mid latitude taxa.

12, 29: Which species help to constrain the age? And revise to "this 4 m-thick interval within the section".

13, 6: Use of "records" implies plural, meaning more than just the Hampden section. Are there data from other NZ sections?

13, 10: What is meant by "60degS front"? Do you mean the polar front? What evidence is presented for it lying north of the gateway?

13, 12: This SST range excludes the high SSTs in the MECO and possible MECO intervals. Why?

13,14: Surely we are not interested in mantle-based paleolatitudes, which are not linked to the Earth's spin axis. Restrict discussion to the uncertainty on the pmag reconstruction.

13, 19. This is a key part of the argument, so needs a stronger word than "may". How about "is more likely to"

13, 20. This is an observation, so replace "suggest" with "find", but I suggest you drop the word "transported", which is interpretation.

13, 21: "transported" is similarly redundant here - "southward reach of the warm EAC..."

13, 24: "Additionally" is not needed.

14, 1: This is an interesting finding, and should be investigated further (see general comments)

14, 7: This statement further serves to highlight why it would be helpful to differentiate cosmopolitan from low/mid latitude taxa

14, 15: You don't explain how this species responded and consequently miss the op-
portunity of expanding on a major discovery: mono-specific blooms of different species of Enneadocysta during the MECO at Sites 1170 and 1172 warrants more discussion.

14, 26: This section is based on the so-called "precarious" use of the SST record to define the EECO at 1170. The alternative correlation noted above also needs to be considered. Note too that the MECO has not been identified for sure on the Otway Basin and is not described at Hampden.

15, 18: Again, a stronger word than "might" is needed here: "most likely"?

15, 25: "production OF dinoflagellate prey ...."

16, 3: Again "seem" is too weak a word. If there is evidence, specify it.

16, 4: Repetition. Replace "sporomorph record at" with "assemblages in"

16, 10: Numerous terms introduced here, either for the first time or with limited context: Wilson Bluff, Latrobe unconformity, Lutetian gap, Khirthar transgression. Consider which ones are actually needed for the argument and explain them more fully.

16, 28. Highly tenuous to suggest a short-lived event like the MECO could be linked to such a large- scale change in base level, accommodation space. A more fruitful approach may be to consider the longer-term climate shift from EECO to MECO, where significant cooling is inferred for early middle Eocene and the MECO is seen in the context of generally warmer conditions in the later middle Eocene (e.g. Pekar et al. 2005)

17, 15 and 18: STR and ETP are areas of ocean floor not localities, so the plankton communities are found "on" them not "at" them.

17, 20: Difficult to reconcile, but you suggest it may be related to the nature of preexisting assemblages. Something on this idea needs to be added to the conclusions.

17, 21: This conclusion is contingent upon age model assumptions.
17, 25: Correlation with the MECO is uncertain.

17, 26: SLR link to MECO is too speculative. Is there evidence for SLF after the MECO?

References:

Morgans, H. E. G., 2009, Late Paleocene to middle Eocene foraminiferal biostratigraphy of the Hampden Beach section, eastern South Island, New Zealand: New Zealand Journal of Geology and Geophysics, v. 52, no. 4, p. 273-320.

Pancost, R. D., Taylor, K. W. R., Inglis, G. N., Kennedy, E. M., Handley, L., Hollis, C. J., Crouch, E. M., Pross, J., Huber, M., Schouten, S., Pearson, P. N., Morgans, H. E. G., and Raine, J. I., 2013, Early Paleogene evolution of terrestrial climate in the SW Pacific, Southern New Zealand: Geochemistry, Geophysics, Geosystems, p. doi: 10.1002/2013gc004935.

Pekar, S. F., Hucks, A., Fuller, M., and Li, S., 2005, Glacioeustatic changes in the early and middle Eocene (51-42 Ma); shallow-water stratigraphy from ODP Leg 189 Site 1171 (South Tasman Rise) and deep-sea d18O records: Geological Society of America Bulletin, v. 117, no. 7-8, p. 1081-1093.

---

## Referee Comment (RC3) · G. Raquel Guerstein (Referee) · 28 May 2019

The Middle Eocene Climatic Optimum (MECO) is a global warming event at about 40 Ma that interrupted the long-term Cenozoic cooling trend. Up to now only a few studies have focused with enough resolution to evaluate the paleoenvironmental and paleobiotic consequences of this hyperthermal event. In this work Cramwinckel and co-authors have investigated the paleoecological and paleoceanographic repercussions of

the MECO in the Southweast Pacific Ocean (SWPO) primarily based on organic walled dinoflagellate cysts (dinocysts) and TEX86 palaeothermometry. The most important site analysed in this study is the ODP Site 1170 located on the western side of the South Tasman Rise (STR). The area where this site was drilled is characterised by a notably high sedimentation rate, especially the stratigraphical interval here interpreted as part of the middle Eocene including the MECO.

Despite the absence of key biostratigraphic markers to validate a robust age-depth frame, the results from this study, togeteher with the information from the Site 1172 (Bijl et., 2010, 2011 and 2013a), conform a dataset of very good quality and high potential to respond the questions posed by the authors. However, I have identified several unsubstantiated interpretations and important methodological shortcomings that reduce the relevance of the paper. In the following I list some points that may be of assistance to make the contribution stronger. I am positive that the authors can carry out the proposed modifications, and I recommend publication of the manuscript after major revisions.

My primary concern is related to the lack of physical arguments to explain the proposed change in the Southern Ocean's surface circulation through the MECO. According to the authors (page 13, lines 8 to 11): Throughout the studied middle Eocene interval, dinocyst assemblages at Site 1170 are dominated by Antarctic-endemic taxa. This implies that the Tasman Gateway was influenced by westward atmospheric and surface-oceanic circulation (i.e., the polar easterlies) around 40 Ma, with the 60° S front thus located to the north of the gateway and the proto-ACC flowing through the Tasman Gateway (Figure 1b).

Then (page 13, line 19), the authors suggest that during the MECO the East-Australian Current (EAC) waters would reach paleolatitudes somewhat less than 60°S, represented by the dinocyst assemblages at Site 1172 on the East Tasman Plateau (ETP) (Fig 1C). Such changes in the path of a Western Boundary Current (WBC) have to be driven by a substantial modification of the global wind pattern. a. Add a squematic

wind distribution in Fig. 1 A, B and C indicating the latitude of zero wind stress curl.

b. Explain the physical mechanisms conducting to the intensification and southward displacement of the the EAC shown in Fig. 1C.

c. If the changes in the EAC are wind driven, then explain the physical mechanisms by which the MECO was able to change the present distribution of wind stress.

d. According of Fig. 1C (representing the MECO situation) the latitude of zero wind stress curl should be about 10-15° to the south of its present location. In that case the southern portion of the Australo-Antarctic Gulf (AAG) would have been under the influence of the westerlies instead of the polar easterlies. Explain how a proto-Antarctic Counter Current (proto-ACC) would flow through a shallow, partially open Tasman Gateway (TG) as proposed by Bijl et al (2013a and b) under such conditions. I suggest to consider another hypothesis to explain the observed dynocysts distribution. Bearing in mind a TG area located at $\sim$ 60°S during the middle Eocene, the cosmopolitan taxa could actually have been transported eastward through the northern portion of an incipient TG from a PLC source, very much like similar interpetations for an early incipient opening of the Drake Passage (see Scher and Martin, 2006; Livermore et al., 2007, Lagabrielle et al., 2009, González Estebenet et al., 2014). This weak flow would reach the ETP (Site 1172) but not the STR (Site 1170), dominated by the TC and a proto-ACC (Fig 1B with slight modifications). Then it would be easy to explain why the surface temperature rise during the MECO would have resulted in increased production of the cosmopolitan Enneadocysta multicornuta on the ETP but not on the STR, where the dominant species is Enneadocysta dictyostila. This species is the member of the Antarctic endemic assemblage most tolerant to warm surface waters (Fig 4C). The data matrix included in the SI reinforces this hypothesis: E. multicornuta is present in Latrobe-1 borehole but has not been recorded in Hampden Section. This interpretation doesn't need Figure 1C but implies changes in the title and a reorganization of some of the sections accordingly.

There are also some methodological weaknesses that are important to take into consideration:

Data and Statistical analyses

a. According to the supplementary information it seems that the statistical analyses are based on proportions (not on counts) and this should be indicated. If they are actually based on proportions the total number of dinocyts counted in each sample should be included in the data tables.

b. Figure 3 illustrates the relative abundances of selected dinocyst biogeographic groups using 4 categories. In the Figure 3B (site 1172) the sum of the 4 categories is not 100% but is not far from it. However, in Fig. 3A (site 1170) it appears that some important information is not taken into account. Indicate which species or groups have not been considered in these cumulative plots and why.

c. In view of the high number of species included in the data tables and that many of them are underrepresented is reasonable that only some of the species were plotted in Figures 4A and 4B. Indicate which criteria were followed for the selection of species.

d. Only 4 samples from the Latrobe-1 borehole were studied and the number of of cyts counted in each sample is very small (based on a minimimum of 50 cyst in each sample). The data available from this site is not of good quality for statistical analyses nor are some of the Hampden Beach samples (based on a minimimum of 90 cyst in each sample). I hardly recommned not to include these samples in the unconstrained NMDS analysis, unless additional counts can make these dinocyst assemblages part of a reliable dataset.

e. Figure 5. Explain the meaning of Enneadocysta – Oligosphaeridium. What is Enneadocysta spp besides Enne-Oli, E.dic and E.mul? Indicate the criteria followed for the selections of species or groups to be plotted in this figure.

Illustration of key markers, taxonomy and dinocyst paleogegraphic affinity

a. The middle Eocene dinocysts assemblages are mainly composed of cysts of extint dinoflagellates. Thus, the illustration of key biostratigraphic and palaeoenvironmental markers is a matter of major relevance and should be part of the main paper or included as Supplementary Information.

b. The taxonomy of the Subfamily Wetzelielloideae is an issue of discussion, which is still open (Williams et al., 2015; Iakovleva, 2016; Bijl et al., 2016; Williams et al., 2017). In this context the ilustration of the key biomarkers is essential. As things are stand now different research groups can use the same name for different morphotypes and the same morphotype can be named in different ways. One of the key biostratigraphic markers for the MECO, here called Dracodinium rhomboideum, has previously found only at Site 1172 and has not been illustrated by Bijl et al. (2013a). Every research group can call this taxa with different names, but a good illustration allows the dinocyt specialist to know if they are talking about the same thing or not. Unquestionably, the authors have the right to follow the taxonomy they consider better and more useful. However, if they reference a "Comment on a paper", they cannot ignore that there is a "Response to that comment" and it should be mentioned (Williams et al., 2017). The authors are free to follow Fensome et al., 2004 for the wetzelielloid taxonomy, of course, but they have to do it for all the members of the subfamily. For example, Rhombodinium rhomboideum had already been transfered to Dradodinium rhomboideum 15 years ago. A taxonomic appendix should be included to avoid these mistakes.

c. Which is the difference between "endemic SO" and the "so called TF"? I suggest to consider all these taxa as "Antarctic endemics" in order to leave the old name "Transantactic Flora" behind. d. Dinolist (Excell file of SI): Indicate the meaning of "biogeo alt" and "g" and "p" Add a column indicating the source of the biogeo (Bijl et al., 2011, Bijl et al., 2013b, Frieling, Appy Sluijs, 2018... or others).

Terrestrial palynomorphs from the Latrobe-1 borehole

This section is the weakest part of the manuscript. The authors overinterpreted a poor

set of data coming from the Latrobe-1 borehole based on only 4 samples within the interval representing the MECO. The section 4.2.2 Terrestrial Palynology (pages 11-12) is merely descriptive using an open taxonomy with broad links to the modern types and no references to their present-day distribution. The section is closed with the following report: "Within the sporomorph assemblages, there is a slight dominance shift between the major pollen groups towards the top of the interval: the percentages of saccate pollen increase from ~15–20 % to ~40 % upsection, while angiosperms decrease from ~40–60 % to ~25 %".... Actualy, it is not consistent to describe a palaeoenvironmental trend based on four samples. Moreover, an avaluation of the vegetational modifications as a consequence of the climatic change during the MECO with no records of the pre and post MECO intervals does not have any sense. Furthermore, the authors concluded (page 17, lines 23-25): "Terrestrial palynomorph assemblages suggest a warm temperate rainforest with some paratropical elements that grew along the southeast Australian margin during the MECO", which can be possible, but the statement clearly does not arise from this unsupported analysis.

I suggest to remove this section unless it can be substantially improved.

Other comments

When different sources are used to reference a concept the references have to follow a chronological order, from the oldest to the youngest. (not in alfabetical order). Example: Page 2, line 22: (Kennett et al., 1974; Cande and Stock, 2004) instead of (Cande and Stock, 2004; Kennett et al., 1974). Check this aspect thoughout the manuscript since there are many of these mistakes.

Page 2, line 28: (Scher and Martin, 2004; Lagabrielle et al., 2009; González Estebenet et al., 2014) instead of (Lagabrielle et al., 2009; Scher and Martin, 2004)

Page 3. Lines 8 and 9: organic walled dinoflagellate cyst assemblage instead of organic dinoflagellate cyst assemblage

Page 3 line 9: (Wrenn and Beckman, 1982; Wrenn and Hart, 1988; Mao and Mor, 1995; Guerstein et al., 2008; Bijl et al., 2011, 2013a) instead of (Wrenn and Beckman, 1982; Wrenn and Hart, 1988; Bijl et al., 2011, 2013a)

Page 3, line 18: dinocyst assemblages instead of dinocyts assemblages

Page 5, line11: delete a repeted "was not"

Page 5, line: The overlying Wilson Bluff transgressive deposits have an age.... instead of "The overlying Wilson Bluff transgression has an age"

Page 5, line 28: Narrawaturk Formation instead of Narrawaturk formation

Page 6, line 7: Narrawaturk Formation (or Fm) instead of Narrawaturk formation

Page 6, line 13: The Hampden section at Hampden Beach, New Zealand (Figure 2a).... which could have recorded influences of both TC and/or EAC. Explain.

Page 7. Line 6: wetzielloids or Subfamily Wetzelielloideae insted of "Wetzellioid family"

Page 7. Lines 16-18: "We label taxa without a clear temperature affinity as cosmopolitan, such as those taxa with a distribution that is primarily controlled by other parameters like salinity (e.g., Senegalinium cpx.) or nutrient availability (e.g., protoperidinioids) Add references

Page 7, line 31: where the only species of Deflandrea recorded was D. antarctica insted of: where only the Deflandrea species D. antarctica is present

Page 9 lines 31 -32: Middle Eocene palynomorphs at Site 1170 are generally well preserved and assemblages are dominated (>95%) by marine forms, mainly dinocysts. Terrestrial palynomorphs occur consistently, but in low relative abundances (<2% of palynomorphs). 95 or 97%? vs. 2 or 5%?

Page 10, lines 2: "possibly from the north". Why?

Page 10, lines 6-8: "High abundances of Enneadocysta spp. and peridinioid dinocysts in combination with low diversity indicate a somewhat restricted, eutrophic assemblage with possible low-salinity influences." Add references

Page 11, line 3: MECO cooling ?

Page 17, lines 2 and 3: Annenberg Formation.... Helmstedt Formation .... Annemberg Formation instead Annenberg formation.... Helmstedt formation .... Annemberg formation

Illustrations

Be consistent using upper or lower case for the figures. Figure 1 shows A, B and C and the figure caption explains the Figure 1 a, b and c. See also Figs 2, 3, 4, 6 and supplementary figures.

References mentioned in this comments and not included in the reference list of the manuscript González Estebenet, M. S., Guerstein, G. R., and Alperin, M. I., 2014. Dinoflagellate cyst distribution during the Middle Eocene in the Drake Passage area: paleoceanographic implications. Ameghiniana, 51(6):500-510. DOI: 10.5710/AMGH.06.08.2014.2727.

Guerstein, G.R., Guler, M.V., Williams, G.L., Fensome, R.A., Chiesa, J.O., 2008. Mid Palaeogene dinoflagellate cysts from Tierra del Fuego, Argentina: biostratigraphy and palaeoenvironments. Journal of Micropalaeontology 27: 75-94.

Iakovleva, A. I., 2016. Did the PETM trigger the first important radiation of wetzelielloideans? Evidence from France and northern Kazakhstan, Palynology, DOI: 10.1080/01916122.2016.1173121 Livermore, R., Hillenbrand, C. D., Meredith, M. and Eagles, G. (2007). Drake Passage and Cenozoic climate: An open and shut case?. Geochemistry, Geophysics, Geosystems, 8 (1) Q01005.

Mao, S., Mohr, B.A.R., 1995. Middle Eocene dinocysts from Bruce Bank (Scotia Sea, Antarctica) and their paleoenvironmental and paleogeographic implications. Review of

Palaeobotany and Palynoly 86: 235-263.

Williams G.L, et al., 2015. Wetzeliella and its allies - the 'hole' story: a taxonomic revision of the Paleogene dinoflagellate subfamily Wetzelielloideae. Palynology 3:1-41.

Williams, G.L., et al., 2017. A response to 'Comment to Wetzeliella and its allies–the 'hole'story: a taxonomic revision of the Paleogene dinoflagellate subfamily Wetzelielloideae by Williams et al.(2015)'. Palynology 41 (3): 430-437. DOI: 10.1080/01916122.2017.1283367

---

## Author Comment (AC2) · 15 Aug 2019

Please see the attached supplementary pdf for our author response.

Please also note the supplement to this comment:
https://cp.copernicus.org/preprints/cp-2019-35/cp-2019-35-AC2-supplement.pdf

---

## Author Response (AR1)

Author: Below, we have copied the review by the referee, and have added our responses in blue and between square brackets.

**R1 - Severine Fauquette (Referee)**

Cramwinckel et al. present a study of the surface-circulation change in the Southern Ocean during the Middle Eocene Climatic Optimum based on dinoflagellate cysts and biomarker paleothermometry. The manuscript submitted by Cramwinckel et al. is of good quality. This manuscript is within the scope of 'Climate of the Past' and is well written and structured. This will be a very useful paper on the MECO period in the Southern Ocean that is not well known. This study will certainly help climate modelers who can introduce consistent boundary conditions into the models for this part of the globe. I recommend publication of their paper in Climate of the Past with however some revisions.

[AR: We thank the Referee, Severine Fauquette, for her positive evaluation of our manuscript and the constructive suggestions. We hope to adequately address these below.]

Main comments:

- Authors should give, in supplementary data, a detailed description of the pollen morphology (apertures, ornamentation of the exine surface…) and some photos of the main palynomorphs (dinocysts and pollen grains). This period in this region is not well known and it could help for further studies.

[AR: Referee 3 also suggested adding plates of the most important palynomorphs. We propose to add a plate containing light microscope photos of the main marine and terrestrial palynomorphs from the studied sites to the revised manuscript. We feel however, that a detailed description of pollen morphology is beyond  the scope of this paper, which focuses on paleoceanography and paleoclimate rather than taxonomy, systematics and morphology of palynomorphs. Regarding the latter topic, future pollen-based studies for this region are being worked on, which will contain the requested morphological and taxonomic data.]

- The fossil pollen and spores should be identify, by comparing them to modern pollen grains, following current taxonomy of recent taxa, instead of using morphotaxa names. By applying such approach, pollen and spores may be assigned to family, genus, and sometimes, but rarely, even to species levels. Once they are botanically identified, their paleoecological requirements may be defined based on the modern taxa. This botanical approach allows reliable paleoenvironmental reconstructions, as described and done by Suan et al. (Geology, 2017) for the Early Ecocene of the Arctic Siberia.

[AR: We agree regarding the utility of correlating fossil pollen to modern botanical taxa with known environmental preferences, to facilitate paleoenvironmental reconstruction. In the current version of the manuscript, we therefore provide the names of the pollen taxa followed by the known botanical affinity in brackets. As these are Eocene taxa, indeed, this is often at the level of family or genus instead of a modern species-level designation.]

- Biostratigraphy: A table with the regional occurrences of the dinocysts could be interesting.

[AR: Bijl et al. 2013 ESR and Bijl et al. 2011 Paleoceanography already document the regional biogeographic occurrence of dinocysts. We cite these papers in our manuscript. We think that publication of this same dataset again is not necessary, and do not include the data in this work. We will more clearly state in the revised manuscript where these tables can be found.]

- A simplified diagram with the stratigraphic log and the percentages of the main terrestrial palynological data of Latrobe-1 borehole is lacking.

[AR: In the revised manuscript, we will add a supplementary figure including a stratigraphic log and palynological data from the Latrobe-1 borehole.]

Minor (technical) comments:
- p5, L1: add a S to metre; L10: call figure 2a; L11: remove one "was not".
- p6, L8: How many samples have been studied for this site?; L25: add "concentration" for the dinocyst content.
- p10, L32: add a reference for the ages given by Lophocysta spp .
- p14, L19: remove the "cf" in front of the reference Bijl et al..
- Figure 5, L17: add a "c" in the word dinocyst.
- Figure 2, L21: in (a), it is not the bathymetry that is illustrated as there is no mention of the depth of the ocean.

[AR: We will adapt all of these technical comments. Instead of adding "concentration" to p6, L25 we propose to adapt wording to "content in specimens per gram".]

Author: Below, we have copied the review by the referee, and have added our responses in blue and between square brackets.

**R2 - Chris Hollis (Referee)**

**General comments**

This is an interesting and important study, comparing and contrasting dinocyst assemblage changes between ODP sites 1170 and 1172, one within the Australo-Antarctic Gulf and one in the SW Tasman Sea, during a time of major climate
10  change in the middle Eocene. The study uses evidence from the assemblages to unravel the interplay of changes in ocean circulation due to tectonics and climate changes. Additional sites and data are used to build the case for a significant regional response to the middle Eocene climatic optimum (MECO) – in terms of changes in plankton communities, terrestrial vegetation and sea level. The interpretations are reasonable in most cases but there are a few areas where the argument is weakened by over-interpretation of what the authors admit are ambiguous data.

[AR: We thank the referee, Chris Hollis, for his positive evaluation of our manuscript, and constructive criticism and comments. We hope we adequately respond to these below.]

The key areas are: the definition of the MECO at Site 1170 based on the TEX86 record, which is clearly open to
20  interpretation;

[AR: While the definition of the MECO at Site 1170 is indeed open to interpretation, we prefer the correlation presented in the current manuscript. We are however open to additionally presenting an alternative interpretation in the revised manuscript. We elaborate on this in response to the comment below regarding p11, line 8.]

the lumping together of cosmopolitan and low/mid latitude taxa, when the latter group is the one that is best able to signal the influence of the EAC and PLC;

[AR: We agree this will be a good addition and will separate these groups in a revised version of Figure 3. This further
30  distinction will however not change our main results or conclusions.]

the lack of convincing evidence for the presence of the MECO in the Latrobe-1 borehole;

[AR: As noted below in the AR to the comment on p12, line 17, presence of the dinocyst species *Dracodinium*
35  *rhomboideum* in two samples from the Latrobe-1 borehole tightly constrains this interval to the MECO. However, also in light of comments by Referee #3 on this topic, in the revised manuscript, we will refrain from separating the four studied Latrobe-1 samples into pre-/post-MECO and MECO samples, and will present these data together without describing trends through time.]

40  and the very tenuous correlation of middle Eocene transgression to a purported MECO-related glacioeustatic event.

[AR: In the current version of the manuscript, we tried to convey that this correlation is tentative. We propose to elaborate on this in the revised version of the manuscript, as outlined in the AR to the comment on p16, line 28.]

45  I have made numerous comments on these and other issues at the places they occur in the text.

However, there is a hidden gem in this dataset that I'm disappointed the authors appear to have overlooked. In our warming world, we are increasingly concerned about the ways ecosystems will be adversely affected by warmer oceans

and changes in ocean circulation. For dinoflagellates there is the further concern of how toxic blooms may impact coastal fisheries. The authors provide a dataset that clearly shows the MECO in this region is linked to dramatic increases in the abundance of single species, analogous to present day blooms. And intriguingly, a species of one genus dominates at Site 1170 whereas another species of the same genus dominates at 1172. Even more intriguing, both species have short-lived blooms leading up to the MECO at 1172. Much of the paper simply combines the data for these two species with their respective biogeographic groups (cosmopolitan and endemic) but these two taxa clearly dominate these groups (as shown by DCA and NMDS) and it is certainly worth considering that the rise and fall of these two species is more directly related to local watermass conditions than to current transport. I'd like to know if there is any indication of EAC or PLC influence with E. multicornuta removed. And I'd like to see more discussion on the watermass conditions that might lead to monospecific blooms of these two species.

[AR: We thank the referee for his interest in, and suggestions on, this specific part of our results. Although it would be very useful to be able to reconstruct harmful dinoflagellate blooms in the past, we unfortunately do not possess enough information to be able to make such assertions here. While we record acmes of fossils in the sediment, we cannot know what kind of paleo-concentrations of plankton in seawater on what timescales (short seasonal blooms? dominance of species throughout the year?) are actually represented by the data. Furthermore, it is not known if dinocyst species within the genus *Enneadocysta* have a blooming-type ecology, as they are not represented by extant dinoflagellate species with a known ecology. From the fossil record, they typically seem to be mid-shelfal species rather than near-coastal. We are therefore hesitant to claim more than a possibility of paleo-blooms, which we suggest in the present manuscript paragraph 5.3, e.g. p15 line 18–20: "The relatively low diversity of the dinocyst assemblages in combination with the high dominance of a single taxon (*Enneadocysta dictyostila* in the MECO interval) suggests a generally eutrophic setting that could have been characterised by seasonal plankton blooms." Especially at Site 1172, the dinocyst assemblage as a whole is characterized by alternating dominance of different taxa (*Enneadocysta*, *Deflandrea*, *Spinidinium/Vozzhennikovia* and *Phthanoperidinium*). We interpret this succession of dominance of different species as changing conditions rather than as a succession of blooms. Therefore, while we certainly agree with the referee on the relevance and appeal of this topic, we are hesitant to include more speculation on the possibility of (harmful) plankton blooms.]

**Specific comments/Corrections by page, line:**

1, 20: I see the term "Tasman Gateway" or "Tasman Seaway" has been used in the literature but it's incorrect. The proper term is "Tasmanian Gateway", being the gateway between Tasmania and Antarctica (see any Leg 189 publication).
[AR: we will correct to "Tasmanian Gateway" throughout the paper]

1, 22: ", including the organic walled cysts of dinoflagellates (dinocysts). I'd like to see a distinction made between dinoflagellates (plankton) and dinoflagellate cysts or dinocysts (fossil remains of the plankton)
[AR: we will clarify this distinction where appropriate in the manuscript]

1, 23: prefer "geographic" to "spatiotemporal" (here and elsewhere)
[AR: we respectfully feel this is a matter of preference and prefer to retain "spatiotemporal", also because the "temporal" aspect is less clearly represented in the word "geographic"]

1, 24: "geographic" here is superfluous. And is it primarily controlled by tectonism? What about the rotation of the Earth? I wonder if this simplistic separation of tectonic and climatic controls is warranted or needed in an abstract? Sentence is awkward, so how about rephrasing: "The extent to which the climatic and tectonic controls on the distribution and composition of surface currents have influence the composition of fossil assemblages …".
1, 26: This sentence is also a little awkward. "Indeed, the extent to which climate change affects oceanographic processes is still poorly understood"?

1, 29: Also, an awkward sentence. "trend, the Middle Eocene Climatic Optimum (MECO, 40 Ma). This 500 kyr-long episode of global warming is unrelated to …"

1, 31: "ocean's"; replace "only" with "alone"

[AR: we will change wording in the abstract according to above suggestions]

2, 1: "our new results…", no hyphen between surface and ocean

[AR: we will change the text accordingly]

2, 2: replace "southward" with "south"

[AR: we will change the text accordingly]

2, 3: Explain how "warm temperate with paratropical elements" MECO assemblage differs from the general middle Eocene pollen assemblage?

[AR: We elaborate on this in the relevant discussion section on the terrestrial palynology of the Latrobe-1 core (5.4). Here we note that this warm flora overlaps with the MECO interval, based on the dinocyst species that are present in the samples, but, p 16 line 15–17: "Future regional pollen studies focussing on the Nirranda group might therefore elucidate whether the relatively warm-loving flora described here was restricted to the MECO interval, or to a broader interval of middle-late Eocene "background" conditions."]

2, 8: change "into" to "to"

[AR: we will change the text accordingly]

2, 13: does "intermediate-deep" mean somewhere between upper and lower deep water or is it shorthand for "intermediate and deep", in which case this formulation is less ambiguous.

[AR: we will change the text accordingly]

2, 15: None of these sites are close enough to the Antarctic margin to be sources of deep water and are all north of the 60S demarcation for the SO, using pmag reference frame (although noting the uncertainty).

[AR: In these lines, we did not have the intention of suggesting that these sites precisely represent the locations of Eocene deep water formation, but instead, that model simulations suggest that these sites lie close to the region of intermediate-deep water formation. In this we mean to distinguish and refer to intermediate-deep water formation. According to model simulations, while bottom waters formed on the Antarctic continental shelf, intermediate-deep waters formed at southern high latitudes, not necessarily only on the Antarctic margin. We will better clarify the above in a revised version of the text.]

2, 18: change "marine-based" to "sea" and, no, they are not supported by estimates for land temperatures from NLR approaches, which are in general closer to the modelled temperatures (add Pancost et al. 2013), so SST estimates are 5-10C warmer than models and LAT estimates.

[AR: we will add the land temperatures from NLR approaches to this section of the introduction, citing Pancost et al. 2013]

2, 21: add comma after processes

[AR: we will change the text accordingly]

2, 22: remove parentheses around global

[AR: we will change the text to "regional and global"]

2, 31: plural "changes". Lord Howe Rise is part of Zealandia so rephrase: "submerged parts of NW Zealandia…"

[AR: we will change the text accordingly]

3, 1: that's a lot of potential effects but rather speculative. Suggest you keep it simple. "... should have affected ocean circulation in the region with likely impacts for global heat transport and climate."
[AR: we will change the text accordingly]

3, 4: change "of" to "from"
[AR: we will change the text accordingly]

3, 5: Change "Southern Ocean" to "SO".
[AR: we will change the text accordingly, and verify use of abbreviations after first definition is consistent throughout]

3, 6: Rephrase: "... endemism are characteristic of a diverse range of fossil groups ..." (circum-Antarctic is tautological when you've already said Southern Ocean)
[AR: we will change the text accordingly]

3, 9: here is where I'd prefer you to use "dinoflagellates". If you use cysts here, you really also need to use frustules for diatoms and tests for forams and rads. Personally, I don't think you need to use "dinocyst" at all, but certainly should not be used when you are talking about plankton as opposed to assemblages in sediment.
[AR: we respectfully disagree on this point, and prefer to restrict the discussion to dinocysts, not dinoflagellates. In contrast to diatom frustules and foraminifera and radiolaria tests, dinoflagellate cysts, being vegetative resting cysts, do not have a 1:1 relationship to the living organism. (The body of the motile dinoflagellate is composed of labile organic material and in general does not preserve in the sediment.) While the abovementioned frustules and tests of other microfossils are truly body fossils of the living organism, the dinocyst is a resting cyst produced during the life cycle of the dinoflagellate. Since we know that not all dinoflagellates produce resting cysts (Head 1996), a single species of cyst-producing dinoflagellate can produce multiple species of dinocyst (Rochon et al. 2009) and dinoflagellate taxonomy is distinct from dinocyst taxonomy, there is a big discrepancy between the dinocyst assemblage and the dinoflagellate assemblage. For the Paleogene, relationships between this plankton group and its biogeography and environmental preferences are all based on dinocyst species and groups, typically without knowledge of which dinoflagellate produced them, certainly as most Paleogene cyst types are extinct. Therefore, we should refrain from extrapolating to dinoflagellates and restrict the discussion to dinocysts. Since, in p3 line 9, we are enumerating types of fossil assemblages, we are referring to dinocysts, not dinoflagellates.]

[Author comment: below, we have grouped a few of the referees comments on the topic of biogeographic terminology, to answer these collectively.]

3, 12: Query use of "cosmopolitan". This is unconventional usage. Cosmopolitan means found everywhere, so hard to see why this group signals the influence of the PLC or EAC.

4, 2: low-latitude and cosmopolitan are not the same thing.

7, 14, 16: Key problem issue for this paper. Definition of "cosmopolitan" is ambiguous and not in line with convention: cosmopolitan = found everywhere. I recommend you use only low and mid-latitude taxa as your guide to PLC and EAC influence.

10, 12: Differentiate cosmopolitan from low/mid latitude.

12, 24: Differentiate cosmopolitan from low/mid latitude taxa.

14, 7: This statement further serves to highlight why it would be helpful to differentiate cosmopolitan from low/mid latitude taxa

[AR: In our dinocyst grouping, we consider species that occur everywhere with respect to latitude, and do not have a specific latitudinal affinity, as "cosmopolitan" species. However, it is definitely likely that within this group of cosmopolitan species there are different habitat preferences, even though these species are principally able to occur at all latitudes (as seems indicated by the ordination analysis and statement on p 14, line 7). Regarding surface currents, the EAC and PLC are expected to bring an assemblage consisting of both cosmopolitan and low/mid latitude taxa, and no Southern Ocean endemic taxa, whereas the TC primarily transports an SO endemic assemblage. During MECO dominance of *Enneadocysta multicornuta* (cosmopolitan ecogroup) is recorded at Site 1172, at the expense of Antarctic endemic species. Although presence of a cosmopolitan assemblage by itself might not be very informative, this change from dominantly Antarctic endemic species to cosmopolitan species during MECO provides information, signalling a change in surface currents. We propose to better explain the above reasoning in paragraph 3.1.2 on "Dinocyst biostratigraphy and palaeogeographic affinity". Furthermore, we propose to better differentiate the cosmopolitan from low/mid latitude group where the referee is asking for a distinction.]

3, 13: NZ is not in the Tasman Sea. It is east of it.
[AR: we will change the text accordingly]

3, 26: change "biogeographical patterns" to "biogeography"
[AR: we will change the text accordingly]

3, 27: why the "cf."?
[AR: upon reflection, we think the "cf." in front of the reference is redundant and will remove it]

3, 28: Why is "orbital scale" mentioned? Is it relevant? Why the "cf."?
[AR: upon reflection, we think the "cf." in front of the reference is redundant and will remove it. We would like to mention "orbital scale" variability here as an indication for the timescale on which these assemblage changes can occur.]

3, 32: Why is deep ocean warming described as "transient" and surface-water warming described as "widespread"
[AR: "transient" is meant to describe both deep- and surface-water warming, whereas "widespread" is meant to describe surface-water warming. We will change wording to clarify.]

3, 34: be a little more specific than "global perturbations"
[AR: we will change the text to more specifically describe oceanographic and environmental changes during the MECO at the sites studied in the cited papers]

4, 3: change "outstanding" to "unresolved"
[AR: we will change the text accordingly]

4,5: Sentences in this paragraph from "In addition ..." to end of paragraph should come before the description of the dinocyst assemblages. These sentences are part of the general description of the MECO.
[AR: agree and we will change the text accordingly]

4, 8: The two factors mentioned do not "imply" a volcanic explanation. Revise this sentence and provide a reference for the volcanic carbon hypothesis.
[AR: we omitted to mention carbon isotope trends over the MECO here. As d$^{13}$C of DIC does not show a negative trend over the MECO, this rules out a depleted source of carbon. Together with the cited reconstructions of carbon cycling

during MECO, a more heavy source of carbon, such as volcanic carbon, thus was the more likely cause. We will add these additional constraints in the revised text to clarify this sentence.]

4, 11: Last sentence of paragraph is poorly worded. Revise.
[AR: we propose to simplify this sentence to ", constraints on global sea level change during the MECO are lacking"]

4, 25: Revise: "in the 2–3 km-deep and 50 km-wide Ninene Basin".
[AR: here we meant to describe that Site 1170 is located in one of the grabens within the Ninene Basin, and that this certain graben is 2-3 km deep and 50 km wide. We will revise the text to clarify.]

5, 18: Delete "interval"; no hyphen between shallow and marine, as for 5, 21.
[AR: we will change the text accordingly]

5, 31: Sentence doesn't make sense. What covers the unconformity and overlies basal Nirranda Group?
5, 32: "Latrobe-1 borehole"
[AR: we will combine this sentence with the next to "The Latrobe-1 borehole (38.693009° S, 143.149995° E) was drilled in 1963–1964 near the Port Campbell Embayment depocenter, reaching a total depth of 620 metres." The information on stratigraphy then follows in the following lines.]

6, 2: change "overlying" to "underlying"; What's the age of the Dilwyn Fm?
[AR: we will change wording here to clarify that the middle Eocene Narrawaturk Fm overlies the early Eocene Dilwyn Fm.]

6, 11: Elsewhere in text it is referred to as Hampden section. Be consistent. Why no mention of the work on the rest of the Eocene / Paleogene section (e.g. Morgans, 2009; Hollis, et al., 2012; Inglis et al., 2015)
[AR: to conform to the other location descriptions, we will add more background information on this section, including these appropriate references for which we thank the reviewer.]

6, 12: missing comma after "...E)"
[AR: we will change the text accordingly]

6, 13: "end-member" is not the right word. How about "analysed to identify influences from the TC or EAC in the middle Eocene prior to the MECO".
[AR: we will change the text accordingly]

6, 28: lower case "s" for section.
[AR: we will change the text accordingly]

7, 2: 50 and 90 are normally seen as too few for robust statistical analysis.
[AR: we agree and will add discussion on this to paragraph 3.3 on statistical analyses]

7, 5: and identified to what taxonomic level?
[AR: typically to the level of genus - we will add this to the text]

7, 27: Again, ambiguous terminology. Your example is not of a taxon with unknown biogeographic affinities, but with conflicting biogeographic affinities.
[AR: We do mean to use the term "unknown" here. Regarding the specific *Deflandrea* example - different species within the genus *Deflandrea* have different geographic ranges. For example, *Deflandrea antarctica* is endemic to the Southern Ocean, whereas *Deflandrea phosphoritica* occurs globally. In our samples, we encountered specimens of *Deflandrea* that

we could only bring to the genus level, because of poor preservation. For those specimens of *Deflandrea* spp., no biogeographic grouping could therefore be made, and they are categorized as "unknown".]

9, 7: What is meant by "spatial"? Lateral? Geographic might be a better term.
[AR: we will change the text accordingly]

9, 24: U is not a direct proxy for TOC.
[AR: we agree and will remove the part in between brackets here]

9, 26: Change "like" to "As with".
[AR: we will change the text accordingly]

10, 2: Change "for" to "of".
[AR: we will change the text accordingly]

10, 5: Change "dinocysts" to "assemblage"
[AR: we will change the text accordingly]

10, 7: Can low salinity be consistent with low BIT?
[AR: since the BIT index is the relative proportion between a (chiefly) terrestrially- and a (chiefly) marine-produced set of components, changing either terrestrial input or marine production can change the BIT index. In terms of BIT index, an increase in influx of terrestrially produced components can thus be offset by an increase in the accumulation of marine components. Therefore, in some settings, low salinity can indeed be consistent with low BIT indices, if marine GDGT production is relatively high.]

10, 9: Change "most dominant" to "most abundant".
[AR: we will change the text accordingly]

10, 13: What does "a.o." mean?
[AR: we will change this to "i.a.", inter alia]

10, 20: delete "at this site"; redundant.
[AR: we will change the text accordingly]

10, 23: Provide error values for SST estimates and show on Fig. 3.
[AR: we will add error bars to Fig. 3, incorporating calibration and analytical uncertainty]

11, 8: "Precarious" is the wrong word, but a good choice nevertheless, because the whole interpretation of this section is precarious due to the subjective way the SST record has been interpreted. This is only one possible interpretation. Another is that the warming at 670 m precedes the MECO and perhaps can be correlated with the broad peak around 440 m at 1172. Thus, the MECO is the interval between 5570 and 600 m at 1170. This shorter duration is consistent with the biostrat and would mean that the cyst accumulation rate is not so untenably high. Both options should be considered.
[AR: As we agree the age constraints for Site 1170 are not conclusive, we present the data for Site 1170 in the depth domain. Our dinocyst age constraints indicate the oldest layers studied (around 770 mbsf) contain *Impagidinium parvireticulatum* (FO 44 Ma), implying they are younger than 44 Ma. Therefore, it is highly unlikely that the temperature optimum around 675 mbsf represents the EECO. While we interpret this temperature optimum to represent peak MECO conditions, we cannot exclude the possibility that a "pre-MECO warming phase" occurred at Site 1170. Although we note

that such a warming, to temperatures above peak MECO, would likely be regional in nature, as it does not seem to occur in MECO SST records from other sites (e.g. Bijl et al. 2010; Boscolo-Galazzo et al. 2014; Cramwinckel et al. 2018) or the global deep ocean (e.g. Zachos et al. 2008). Furthermore, we do note that high sedimentation rates are plausible given the seismic interpretation, as shown in Figure 2B. However, we agree that both temperature correlations cannot be excluded and propose to present the alternative explanation in the revised manuscript. We will then revise our (very rough) estimate of cyst accumulation rates to consider both options. We note however, that tenfold lower dinocyst accumulation rates would still be very high, and dinocyst concentrations are very high regardless. All of our other analyses and conclusions are neutral to which interpretation is chosen, as the dinocyst assemblages are highly similar over the interval ~575-680 mbsf at Site 1170.]

11, 16: Poorly worded. "sufficient numbers of dinocysts were encountered for counts of 50-100 specimens to be undertaken. Other marine palynomorphs such as prasinophytes and acritarchs, were rare/common(?)"
[AR: we will change the text accordingly (using "rare")]

11, 31: Revise sentence beginning "Furthermore…" to "Cycadopites … are also present but rare.
[AR: we will change the text accordingly]

12, 1. Simultaneously is the wrong word. Delete. The abundance of Dilwynites, Protea… also decrease towards the top of the borehole.
[AR: in light of comments by Referee 3, we will remove the description of trends in these four samples.]

12, 17. Very poorly worded but crucial sentence. The FO of this species is said to be at 40 Ma. When is the LO? It can only be used to define the MECO if it's restricted to the MECO. I conclude from the biostrat presented that the interval may include the MECO but equally may be younger (anywhere between 40 to 35.95 Ma).
[AR: In fact, the stratigraphic range of *Dracodinium rhomboideum* in the South Pacific Dinocyst Zonation of Bijl et al. 2013 is very restricted, as *D. rhomboideum* was only recorded in one sample at Site 1172 with an age of 40 ± 0.1 Ma, within Chron 18n.2n. This corresponds to peak MECO in a compilation of deep sea stable isotope records (Bohaty et al. 2009) as well as coinciding with peak SST based on TEX$_{86}$ at Site 1172 itself. Notably, the range of *D. rhomboideum* in the North Atlantic Ocean (Eldrett et al. 2004 Marine Geology) is also restricted to the MECO interval (from C18n.2n 0% to C18n.1r 50%, or from 40.14 Ma to 39.66 Ma in GTS2012).
Therefore, even a few specimens (1 in sample L85, 3 in sample L86 in this case - we will add the counts in addition to the relative proportions in the supplementary datafile) of this dinocyst species firmly correlate this interval to the MECO.]

12, 29: Which species help to constrain the age? And revise to "this 4 m-thick interval within the section".
[AR: we will change to: "This dinocyst assemblage is in agreement with the age of c. 41.7 Ma as previously assigned to this 4 m-thick interval within the section]

13, 6: Use of "records" implies plural, meaning more than just the Hampden section. Are there data from other NZ sections?
[AR: In addition to the here presented data from the Hampden section there are also a few records from other NZ sections as presented in Bijl et al. 2011 Paleoceanography Figure 2f. We will add a citation here.]

13, 10: What is meant by "60degS front"? Do you mean the polar front? What evidence is presented for it lying north of the gateway?
[AR: Here we are indeed referring to the polar front. As further elaborated in the response to Referee 3, the fossil plankton evidence is suggestive of a Tasmanian Gateway that is influenced by a westward surface circulation, i.e. the polar front separating the polar easterlies from the westerlies to the north.]

13, 12: This SST range excludes the high SSTs in the MECO and possible MECO intervals. Why?
[AR: we could indeed expand this range to also include MECO SSTs, and not just "background" SSTs and will do so.]

13,14: Surely we are not interested in mantle-based paleolatitudes, which are not linked to the Earth's spin axis. Restrict discussion to the uncertainty on the pmag reconstruction.
[AR: Since, as far as we can judge, there is still discussion within the community over which reference frame to use, we prefer to be inclusive and shortly mention both. This is also relevant in model-proxy comparison, as several GCMs use mantle-based absolute reference frames.]

13, 19. This is a key part of the argument, so needs a stronger word than "may". How about "is more likely to"
[AR: In response to comments by Referee 3 we will adapt the section including this line in two ways. First, we will support the possibility of further southward extent of the EAC during MECO by citing literature on model simulations and modern observations illustrating wind-driven intensification of the EAC under conditions of enhanced global warmth. Next to this, we will also discuss the suggestion of weak eastward surface transport through the northern part of the Tasmanian Gateway (see response to Referee 3 for more details).]

13, 20. This is an observation, so replace "suggest" with "find", but I suggest you drop the word "transported", which is interpretation.
[AR: we will change the text accordingly]

13, 21: "transported" is similarly redundant here - "southward reach of the warm EAC..."
[AR: we will change the text accordingly]

13, 24: "Additionally" is not needed.
[AR: we will change the text accordingly]

14, 1: This is an interesting finding, and should be investigated further (see general comments)
[AR: see AR above under the general comments]

14, 15: You don't explain how this species responded and consequently miss the opportunity of expanding on a major discovery: mono-specific blooms of different species of Enneadocysta during the MECO at Sites 1170 and 1172 warrants more discussion.
[AR: as outlined above, correlating fossil acmes to ecological blooms warrants caution and we are therefore hesitant to call these acmes "blooms".]

14, 26: This section is based on the so-called "precarious" use of the SST record to define the EECO at 1170. The alternative correlation noted above also needs to be considered.
[AR: In this section we primarily discuss dinocyst concentrations, not accumulation rates. These concentrations are very high, regardless of age model. With our preferred MECO correlation, cyst accumulation rates, although with large error, would also have been extremely high. We will add the alternative correlation to generate a low-end estimate on accumulation rates.]

Note too that the MECO has not been identified for sure on the Otway Basin and is not described at Hampden.

15, 18: Again, a stronger word than "might" is needed here: "most likely"?
[AR: we prefer the word "might" here, since more positive evidence would be necessary to improve certainty]

15, 25: "production OF dinoflagellate prey …"
[AR: we will change the text accordingly]

16, 3: Again "seem" is too weak a word. If there is evidence, specify it.
[AR: we will change to "but also include a small proportion of meso- and megathermal components"]

16, 4: Repetition. Replace "sporomorph record at" with "assemblages in"
[AR: we will change the text accordingly]

16, 10: Numerous terms introduced here, either for the first time or with limited context: Wilson Bluff, Latrobe unconformity, Lutetian gap, Khirthar transgression. Consider which ones are actually needed for the argument and explain them more fully.
[AR: we agree and will make this more concise, focussing on the sequence of early Eocene sedimentation followed by erosion (Latrobe unconformity, Lutetian gap) followed by middle Eocene sedimentation (Wilson Bluff transgression, Khirthar transgression).]

16, 28. Highly tenuous to suggest a short-lived event like the MECO could be linked to such a large- scale change in base level, accommodation space. A more fruitful approach may be to consider the longer-term climate shift from EECO to MECO, where significant cooling is inferred for early middle Eocene and the MECO is seen in the context of generally warmer conditions in the later middle Eocene (e.g. Pekar et al. 2005)
[AR: In a largely ice-free world such as the middle Eocene, accommodation space on the continental shelf (on time scales of $10^6$-$10^7$ years) would have been determined indeed not only by thermal expansion, but also by sediment supply and basin subsidence. The renewed drowning of the continental shelf, as reflected in the Wilson Bluff transgression, seems unlikely to be related to slow and continuous basin subsidence. Instead, ocean warming during the MECO would have caused thermal expansion of seawater, and climate and environmental change could have altered sediment supply. As we note in the Conclusion section, the current age control on these sections is not nearly sufficient to be able to correlate these transgressive surfaces to the MECO with certainty. We are merely noting the curious coincidence in timing, which we feel is worthy of further investigation. We shall express this point more clearly in the revised text.]

17, 15 and 18: STR and ETP are areas of ocean floor not localities, so the plankton communities are found "on" them not "at" them.
[AR: we will change the text accordingly]

17, 20: Difficult to reconcile, but you suggest it may be related to the nature of preexisting assemblages. Something on this idea needs to be added to the conclusions.
[AR: With this statement, we meant to indicate that similar sea surface temperatures above the STR and ETP are not expected, if indeed an extension of the (warmer) EAC reaches the ETP, while the (colder) proto-ACC influences the STR. We will rephrase to clarify.]

17, 21: This conclusion is contingent upon age model assumptions.
[AR: We will add the accumulation rate estimates based on alternative age constraints, but note that absolute concentrations of dinocyst are very high given the setting, independent of accumulation rates.]

17, 25: Correlation with the MECO is uncertain.
[AR: As noted above, we will better illustrate the stratigraphic usability of *Dracodinium rhomboideum* to strengthen this correlation.]

17, 26: SLR link to MECO is too speculative. Is there evidence for SLF after the MECO?
[AR: We agree the SLR link to the MECO is speculative. Higher resolution age control combined with a more detailed paleoenvironmental and/or sedimentological study could better resolve the timing of SLR and SLF. However, as noted above, the stratigraphic range of *Dracodinium rhomboideum* is very short and strongly tied to the MECO. We would like to include this curious timing of the regional transgression in a final discussion paragraph, to be able to present this as a promising direction for further investigation to the paleoceanography community.]

]

Author: Below, we have copied the review by the referee, and have added our responses in blue and between square brackets.

**R3 - G. Raquel Guerstein (Referee)**

The Middle Eocene Climatic Optimum (MECO) is a global warming event at about 40 Ma that interrupted the long-term Cenozoic cooling trend. Up to now only a few studies have focused with enough resolution to evaluate the paleoenvironmental and paleobiotic consequences of this hyperthermal event. In this work Cramwinckel and coauthors have investigated the paleoecological and paleoceanographic repercussions of the MECO in the Southwest Pacific

10 Ocean (SWPO) primarily based on organic walled dinoflagellate cysts (dinocysts) and TEX86 palaeothermometry. The most important site analysed in this study is the ODP Site 1170 located on the western side of the South Tasman Rise (STR). The area where this site was drilled is characterised by a notably high sedimentation rate, especially the stratigraphical interval here interpreted as part of the middle Eocene including the MECO. Despite the absence of key biostratigraphic markers to validate a robust age-depth frame, the results from this study, togeteher with the

15 information from the Site 1172 (Bijl et., 2010, 2011 and 2013a), conform a dataset of very good quality and high potential to respond the questions posed by the authors. However, I have identified several unsubstantiated interpretations and important methodological shortcomings that reduce the relevance of the paper. In the following I list some points that may be of assistance to make the contribution stronger. I am positive that the authors can carry out the proposed modifications, and I recommend publication of the manuscript after major revisions.

[AR: We thank the referee, G. Raquel Guerstein, for her positive evaluation of our dataset and manuscript, and critical but constructive concerns and comments. We hope to adequately respond to these below and in a revised version of the manuscript.]

25 My primary concern is related to the lack of physical arguments to explain the proposed change in the Southern Ocean's surface circulation through the MECO. According to the authors (page 13, lines 8 to 11): Throughout the studied middle Eocene interval, dinocyst assemblages at Site 1170 are dominated by Antarctic-endemic taxa. This implies that the Tasman Gateway was influenced by westward atmospheric and surfaceoceanic circulation (i.e., the polar easterlies) around 40 Ma, with the 60 S front thus located to the north of the gateway and the proto-ACC flowing through the

30 Tasman Gateway (Figure 1b). Then (page 13, line 19), the authors suggest that during the MECO the East-Australian Current (EAC) waters would reach paleolatitudes somewhat less than 60 S, represented by the dinocyst assemblages at Site 1172 on the East Tasman Plateau (ETP) (Fig 1C). Such changes in the path of a Western Boundary Current (WBC) have to be driven by a substantial modification of the global wind pattern.

35 [AR: In this study, we use our fossil dinocyst data as a tool to reconstruct surface ocean currents. In the MECO interval we find cosmopolitan dinocysts at Site 1172 but not Site 1170, and consequently explore ocean circulation changes that can account for this biogeographic distribution. We explore several mechanisms and identify the one we consider most likely (southward extent of the EAC). Indeed, such changes in surface ocean circulation would follow changes in the wind pattern - given bathymetric and geographic constraints. We would like to emphasize that the

40 bathymetric/paleogeographic constraints are just as important as the wind patterns, and both are much less well-constrained than the existing model simulations seem to suggest. As discussed below, we thank the referee for bringing another potential mechanism to our attention and will add this to the discussion.

We respond to the specific comments a-d below.]

a. Add a squematic wind distribution in Fig. 1 A, B and C indicating the latitude of zero wind stress curl.

[AR: While we agree that it would be insightful to draw in the prevailing wind directions in the Eocene, unfortunately these reconstructions do not reliably exist, so we respectfully refrain from drawing them. The middle Eocene ocean circulation patterns that we draw are based on fossil plankton biogeography, but we prefer to not infer wind circulation patterns from this, as this additional step would introduce a lot of uncertainty. Alternatively, drawing wind circulation patterns as derived from model simulations does not provide a solution either. Atmospheric simulations as derived from fully-coupled coarse resolution GCMs (that are tuned to reproduce modern conditions), are still limited by the poorly-constrained Eocene boundary conditions. Detailed model output is too dependent on these poorly resolved boundary conditions in order to be leading in drawing atmospheric reconstructions.]

b. Explain the physical mechanisms conducting to the intensification and southward displacement of the the EAC shown in Fig. 1C.

c. If the changes in the EAC are wind driven, then explain the physical mechanisms by which the MECO was able to change the present distribution of wind stress.

[AR to b and c: We thank the referee for noticing we did not elaborate on this mechanism. Given the present constraints on MECO temperature (Bohaty et al. 2009 Paleoceanography; Bijl et al. 2010 Science; Boscolo-Galazzo et al. 2014 Paleoceanography; Cramwinckel et al. 2018 Nature; Giorgioni et al. 2019 Scientific Reports), the MECO was likely a global warming event, possibly driven by atmospheric $CO_2$ increase (Bijl et al. 2010 Science; Steinthorsdottir et al. 2019 Geology). For the modern ocean, climate model simulations using modern boundary conditions indicate that increased $CO_2$ forcing (with associated global warming) causes changes in zonal wind stress (maximum change around 60 °S) and large increases in positive wind stress curl south of the Tasman Sea and New Zealand (Cai et al. 2005 GRL). In these simulations, the changes in wind stress curl drive changes in ocean surface circulation characterized by intensification of the southern midlatitude circulation, including strengthening and further southward extent of the EAC. Indeed, observational data indicate a strengthening of the South Pacific Gyre over the past six decades, including a southward extent of the EAC at the expense of the Tasman Front (Hill et al. 2008 GRL; Hill et al. 2011 GRL). SST anomaly reconstructions over the peak interglacial Marine Isotope Stage 5e (~125 ka) similarly indicate strengthening and further southward extent of the EAC to offshore Tasmania (Cortese et al. 2013 Paleoceanography). We propose a similar atmospheric and oceanographic response to global warming occurred during MECO and will add the above discussion to our discussion paragraph on ocean circulation change during MECO.]

d. According of Fig. 1C (representing the MECO situation) the latitude of zero wind stress curl should be about 10-15 to the south of its present location. In that case the southern portion of the Australo-Antarctic Gulf (AAG) would have been under the influence of the westerlies instead of the polar easterlies. Explain how a proto-Antarctic Counter Current (proto-ACC) would flow through a shallow, partially open Tasman Gateway (TG) as proposed by Bijl et al (2013a and b) under such conditions.

[AR: Notably, the 60 °S line we draw in Figure 1C has quite some uncertainty. First, there is the choice of (and discussion on) which reference frame to use in order to reconstruct paleolatitude, with the first-order choice being between mantle- and paleomagnetic-based absolute reference frames. Second, there is an intrinsic error or uncertainty associated with the paleolatitude reconstructions of every chosen reference frame. For example, in Figure 1C, Site 1170 is drawn at 61.6 °S at 40 Ma, according to the Torsvik et al. (2012) paleomagnetic reference frame, but the uncertainty margins on this are between 58.76 °S and 64.55 °S (www.paleolatitude.org; Hinsbergen et al. 2015). Using the Besse and Courtillot (2002) reference frame gives a range of 57.52 °S – 64.12 °S. Given these uncertainties on the precise location of the 60 °S paleolatitude that approximately separates the westerlies from polar easterlies, we prefer to instead follow the paleobiogeographical data in order to infer circulation. These data suggest westward flow through the southern portion of the Tasmanian Gateway, which is within the uncertainty limits of the paleolatitude reconstructions (pointing more

towards the more southerly latitudes within the uncertainty). To clarify the above, we propose to add uncertainty to the lines of paleolatitude in Figure 1.]

I suggest to consider another hypothesis to explain the observed dynocysts distribution. Bearing in mind a TG area located at 60 S during the middle Eocene, the cosmopolitan taxa could actually have been transported eastward through the northern portion of an incipient TG from a PLC source, very much like similar interpetations for an early incipient opening of the Drake Passage (see Scher and Martin, 2006; Livermore et al., 2007, Lagabrielle et al., 2009, González Estebenet et al., 2014). This weak flow would reach the ETP (Site 1172) but not the STR (Site 1170), dominated by the TC and a proto-ACC (Fig 1B with slight modifications). Then it would be easy to explain why the surface temperature rise during the MECO would have resulted in increased production of the cosmopolitan Enneadocysta multicornuta on the ETP but not on the STR, where the dominant species is Enneadocysta dictyostila. This species is the member of the Antarctic endemic assemblage most tolerant to warm surface waters (Fig 4C). The data matrix included in the SI reinforces this hypothesis: E. multicornuta is present in Latrobe-1 borehole but has not been recorded in Hampden Section.

This interpretation doesn0t need Figure 1C but implies changes in the title and a reorganization of some of the sections accordingly.

[AR: We thank the referee for this suggestion. We agree that weak continuous eastward flow through the northern portion of the Tasmanian Gateway, or discontinuous eddy transport, could have been a mechanism that brought cosmopolitan dinocysts to Site 1172, but not Site 1170. We will add this potential mechanism to our discussion section in the revised version of the manuscript. We note that this explanation, similar to the EAC extending further south, raises the question why this process would only occur during MECO warmth. We propose that eastward eddy or weak continuous transport could principally occur throughout the middle Eocene, but transported species were only able to dominate the assemblage under sufficiently warm temperatures during MECO. We will add the above considerations to our revised text.]

There are also some methodological weaknesses that are important to take into consideration:

**Data and Statistical analyses**

a. According to the supplementary information it seems that the statistical analyses are based on proportions (not on counts) and this should be indicated. If they are actually based on proportions the total number of dinocyts counted in each sample should be included in the data tables.

[AR: Indeed, the ordination analyses are based on proportions, or relative abundances. In the revised version, we will clearly state this in the methods section. Furthermore, we will add the total number of dinocysts counted per sample to the data tables, as we agree this is important information.]

b. Figure 3 illustrates the relative abundances of selected dinocyst biogeographic groups using 4 categories. In the Figure 3B (site 1172) the sum of the 4 categories is not 100% but is not far from it. However, in Fig. 3A (site 1170) it appears that some important information is not taken into account. Indicate which species or groups have not been considered in these cumulative plots and why.

[AR: Unfortunately, especially the younger part of the Site 1170 record contained a high proportion of poorly preserved *Deflandrea* specimens that we could only determine to the level of genus. Therefore, these could not be given a biogeographic grouping, as described on page 7, lines 26–30 of the present manuscript. We will note the relevance of this to the Site 1170 dinocyst record in the caption of Figure 3 in the revised version of the manuscript.]

c. In view of the high number of species included in the data tables and that many of them are underrepresented is reasonable that only some of the species were plotted in Figures 4A and 4B. Indicate which criteria were followed for the selection of species.

[AR: This is indeed the case. In the figure caption, the sentence "For visual clarity, only the most abundant taxa (taxa that occur in >10% of the samples, have a mean relative abundance >1%, and have a maximum relative abundance of >5%) are shown in these plots" contains our criteria. We will change "in these plots" to "in all three panels" to clarify this applies to panel A and B as well as C.]

d. Only 4 samples from the Latrobe-1 borehole were studied and the number of of cyts counted in each sample is very small (based on a minimimum of 50 cyst in each sample). The data available from this site is not of good quality for statistical analyses nor are some of the Hampden Beach samples (based on a minimimum of 90 cyst in each sample). I hardly recommnd not to include these samples in the unconstrained NMDS analysis, unless additional counts can make

15  these dinocyst assemblages part of a reliable dataset.

[AR: We agree with the referee that caution should be taken in doing statistical analyses on assemblage counts of <150–200 palynomorphs. We prefer, however, to present the results for the reader to assess, adding the cautionary note that these analyses are based on low count data.]

e. Figure 5. Explain the meaning of Enneadocysta – Oligosphaeridium. What is Enneadocysta spp besides Enne-Oli, E.dic and E.mul?

[AR: We encountered these *Enneadocysta-Oligosphaeridium* intermediates (as we have designated them) only at the

25  Latrobe-1 borehole. These specimens have a morphology in between *Enneadocysta* (*multicornuta*) and *Oligosphaeridium* spp., being dorsoventrally compressed and following the tabulation pattern of  *Enneadocysta* spp. and having several processes conform *Enneadocysta* (thin, solid, distally radiating), but also having multiple processes conform *Oligosphaeridium* (much thicker, tubiform, distally less complex). The preservation and quantity of the material is not sufficient for description of this as a new species, which is why we describe them as *"Enneadocysta-Oligosphaeridium*

30  intermediate". To clarify, we will add a short description to our datafile, in the sheet "Dinolist" column "notes". *Enneadocysta* spp. are species of *Enneadocysta* with insufficient characteristics preserved to bring their determination to the species level, but that do not fall into the category of *Enneadocysta-Oligosphaeridium* intermediates.]

Indicate the criteria followed for the selections of species or groups to be plotted in this figure.

[AR: The criteria are the same as for Figure 4, which we will add to the caption of Figure 5.]

**Illustration of key markers, taxonomy and dinocyst paleogegraphic affinity**

a. The middle Eocene dinocysts assemblages are mainly composed of cysts of extint dinoflagellates. Thus, the illustration of key biostratigraphic and palaeoenvironmental markers is a matter of major relevance and should be part of the main

45  paper or included as Supplementary Information.

[AR: Referee 1 also commented that a plate with key markers would make a useful addition to the manuscript. We propose to add a plate with key palynomorph species to the revised manuscript as a supplementary figure, including the below mentioned *Dracodinium rhomboideum*.]

5  b. The taxonomy of the Subfamily Wetzelielloideae is an issue of discussion, which is still open (Williams et al., 2015; Iakovleva, 2016; Bijl et al., 2016; Williams et al., 2017). In this context the ilustration of the key biomarkers is essential. As things are stand now different research groups can use the same name for different morphotypes and the same morphotype can be named in different ways. One of the key biostratigraphic markers for the MECO, here called Dracodinium rhomboideum, has previously found only at Site 1172 and has not been illustrated by Bijl et al. (2013a).
10  Every research group can call this taxa with different names, but a good illustration allows the dinocyt specialist to know if they are talking about the same thing or not. Unquestionably, the authors have the right to follow the taxonomy they consider better and more useful. However, if they reference a "Comment on a paper", they cannot ignore that there is a "Response to that comment" and it should be mentioned (Williams et al., 2017). The authors are free to follow Fensome et al., 2004 for the wetzelielloid taxonomy, of course, but they have to do it for all the members of the subfamily. For
15  example, Rhombodinium rhomboideum had already been transfered to Dradodinium rhomboideum 15 years ago. A taxonomic appendix should be included to avoid these mistakes.

[AR: We will add an illustration of *Dracodinium rhomboideum* to a supplementary plate. In the supplementary datafile, we will add author references to the dinocyst species, to change this into a taxonomic appendix. Furthermore, we will add a
20  citation to the response to the comment at the appropriate place in the text.]

c. Which is the difference between "endemic SO" and the "so called TF"? I suggest to consider all these taxa as "Antarctic endemics" in order to leave the old name "Transantactic Flora" behind.

25  [AR: We agree and will group the "endemic SO" and "so called TF" as "Antarctic endemics.]

d. Dinolist (Excell file of SI): Indicate the meaning of "biogeo alt" and "g" and "p" Add a column indicating the source of the biogeo (Bijl et al., 2011, Bijl et al., 2013b, Frieling, Appy Sluijs, 2018... or others).

30  [AR: We will make these additions to the datafile.]

**Terrestrial palynomorphs from the Latrobe-1 borehole**

This section is the weakest part of the manuscript. The authors overinterpreted a poor set of data coming from the
35  Latrobe-1 borehole based on only 4 samples within the interval representing the MECO. The section 4.2.2 Terrestrial Palynology (pages 11-12) is merely descriptive using an open taxonomy with broad links to the modern types and no references to their present-day distribution. The section is closed with the following report: "Within the sporomorph assemblages, there is a slight dominance shift between the major pollen groups towards the top of the interval: the percentages of saccate pollen increase from 15–20 % to 40 % upsection, while angiosperms decrease from 40–60 % to
40  25 %".... Actualy, it is not consistent to describe a palaeoenvironmental trend based on four samples. Moreover, an avaluation of the vegetational modifications as a consequence of the climatic change during the MECO with no records of the pre and post MECO intervals does not have any sense. Furthermore, the authors concluded (page 17, lines 23-25): "Terrestrial palynomorph assemblages suggest a warm temperate rainforest with some paratropical elements that grew along the southeast Australian margin during the MECO", which can be possible, but the statement clearly does not arise
45  from this unsupported analysis. I suggest to remove this section unless it can be substantially improved.

[AR: While we agree with the referee that 4 palynological samples comprise a limited set of data, we respectfully disagree that this would make the data less suitable for publication in our manuscript. While limited in number, these

palynological assemblages provide crucial additional information on middle Eocene warmth on the nearby continent, supporting the marine-based reconstructions. This is important, as land and ocean temperatures did not necessarily change synchronously in this region throughout the Eocene (e.g., Pancost et al. 2013 G³; Bijl et al. 2013 PNAS). The presence of dinocyst marker species *Dracodinium rhomboideum* strongly indicates a MECO age (see the author response to Referee 2). Nevertheless, we agree that our description of trends based on 4 samples might not be sensible, so we propose to omit this in the revised version.]

**Other comments**

When different sources are used to reference a concept the references have to follow a chronological order, from the oldest to the youngest. (not in alfabetical order). Example: Page 2, line 22: (Kennett et al., 1974; Cande and Stock, 2004) instead of (Cande and Stock, 2004; Kennett et al., 1974). Check this aspect thoughout the manuscript since there are many of these mistakes.
Page 2, line 28: (Scher and Martin, 2004; Lagabrielle et al., 2009; González Estebenet et al., 2014) instead of (Lagabrielle et al., 2009; Scher and Martin, 2004)
Page 3 line 9: (Wrenn and Beckman, 1982; Wrenn and Hart, 1988; Mao and Mor, 1995; Guerstein et al., 2008; Bijl et al., 2011, 2013a) instead of (Wrenn and Beckman, 1982; Wrenn and Hart, 1988; Bijl et al., 2011, 2013a)
[AR: Although the CP formatting guidelines leave these decisions to the authors, we will adjust to a chronological reference order.]

Page 3. Lines 8 and 9: organic walled dinoflagellate cyst assemblage instad of organic dinoflagellate cyst assemblage
[AR: we will change the text accordingly]

Page 3, line 18: dinocyst assemblages instead of dinocyts assemblages
[AR: we will change the text accordingly]

Page 5, line11: delete a repeted "was not"
[AR: we will change the text accordingly]

Page 5, line: The overlying Wilson Bluff transgressive deposits have an age.... instead of "The overlying Wilson Bluff transgression has an age"
[AR: we will change the text accordingly]

Page 5, line 28: Narrawaturk Formation instead of Narrawaturk formation
[AR: we will change the text accordingly]

Page 6, line 7: Narrawaturk Formation (or Fm) instead of Narrawaturk formation
[AR: we will change the text accordingly]

Page 6, line 13: The Hampden section at Hampden Beach, New Zealand (Figure 2a).... which could have recorded influences of both TC and/or EAC. Explain.
[AR: we will change this sentence following the suggestion by Referee 2, who also commented on it]

Page 7. Line 6: wetzellioids or Subfamily Wetzellioideae insted of "Wetzellioid family"
[AR: we will change to "wetzellioids"]

Page 7. Lines 16-18: "We label taxa without a clear temperature affinity as cosmopolitan, such as those taxa with a distribution that is primarily controlled by other parameters like salinity (e.g., Senegalinium cpx.) or nutrient availability (e.g., protoperidinioids) Add references
[AR: we will add appropriate references to Sluijs et al. 2005; Sluijs and Brinkhuis 2009; Frieling and Sluijs 2018.]

Page 7, line 31: where the only species of Deflandrea recorded was D. antarctica insted of: where only the Deflandrea species D. antarctica is present
[AR: we will change the text accordingly]

10  Page 9 lines 31 -32: Middle Eocene palynomorphs at Site 1170 are generally well preserved and assemblages are dominated (>95%) by marine forms, mainly dinocysts. Terrestrial palynomorphs occur consistently, but in low relative abundances (<2% of palynomorphs). 95 or 97%? vs. 2 or 5%?
[AR: we will change "<2%", to "<5%". This was a small inconsistency, because there is only one sample with 95% marine and 5% terrestrial palynomorphs.]

Page 10, lines 2: "possibly from the north". Why?
[AR: because this was a relatively nearby land mass for offshore transport of material]

Page 10, lines 6-8: "High abundances of Enneadocysta spp. and peridinioid dinocysts in combination with low diversity
20  indicate a somewhat restricted, eutrophic assemblage with possible low-salinity influences." Add references
[AR: we will cite Sluijs et al. 2005 for these environmental inferences]

Page 11, line 3: MECO cooling ?
[AR: we will change this to "MECO recovery"]

Page 17, lines 2 and 3: Annenberg Formation.... Helmstedt Formation .... Annemberg Formation instead Annenberg formation.... Helmstedt formation .... Annemberg formation
[AR: we will capitalize "Formation" here]

30  Illustrations Be consistent using upper or lower case for the figures. Figure 1 shows A, B and C and the figure caption explains the Figure 1 a, b and c. See also Figs 2, 3, 4, 6 and supplementary figures.
[AR: we will be consistent and change these to lower case between brackets, in accordance with the CP house style]

**References mentioned in this comments and not included in the reference list of the manuscript**

González Estebenet, M. S., Guerstein, G. R., and Alperin, M. I., 2014. Dinoflagellate cyst distribution during the Middle Eocene in the Drake Passage area: paleoceanographic implications. Ameghiniana, 51(6):500-510. DOI: 10.5710/AMGH.06.08.2014.2727.

40  Guerstein, G.R., Guler, M.V., Williams, G.L., Fensome, R.A., Chiesa, J.O., 2008. Mid Palaeogene dinoflagellate cysts from Tierra del Fuego, Argentina: biostratigraphy and palaeoenvironments. Journal of Micropalaeontology 27: 75-94.

Iakovleva, A. I., 2016. Did the PETM trigger the first important radiation of wetzelielloideans? Evidence from France and northern Kazakhstan, Palynology, DOI: 10.1080/01916122.2016.1173121

Livermore, R., Hillenbrand, C. D., Meredith, M. and Eagles, G. (2007). Drake Passage and Cenozoic climate: An open and shut case?. Geochemistry, Geophysics, Geosystems, 8 (1) Q01005.

Mao, S., Mohr, B.A.R., 1995. Middle Eocene dinocysts from Bruce Bank (Scotia Sea, Antarctica) and their paleoenvironmental and paleogeographic implications. Review of Palaeobotany and Palynoly 86: 235-263.

Williams G.L, et al., 2015. Wetzeliella and its allies - the 'hole' story: a taxonomic revision of the Paleogene dinoflagellate subfamily Wetzelielloideae. Palynology 3:1-41.

Williams, G.L., et al., 2017. A response to 'Comment to Wetzeliella and its allies–the 'hole'story: a taxonomic revision of the Paleogene dinoflagellate subfamily Wetzelielloideae by Williams et al.(2015)'. Palynology 41 (3): 430-437. DOI: 10.1080/01916122.2017.1283367

[AR references not cited in manuscript:

Besse, J. and Courtillot, V.: Apparent and true polar wander and the geometry of the geomagnetic field over the last 200 Myr, Journal of Geophysical Research: Solid Earth, 107(B11), EPM 6-1-EPM 6-31, doi:10.1029/2000JB000050, 2002.

Cai, W., Shi, G., Cowan, T., Bi, D. and Ribbe, J.: The response of the Southern Annular Mode, the East Australian Current, and the southern mid-latitude ocean circulation to global warming, Geophysical Research Letters, 32(23), doi:10.1029/2005GL024701, 2005.

Cortese, G., Dunbar, G. B., Carter, L., Scott, G., Bostock, H., Bowen, M., Crundwell, M., Hayward, B. W., Howard, W., Martínez, J. I., Moy, A., Neil, H., Sabaa, A. and Sturm, A.: Southwest Pacific Ocean response to a warmer world: Insights from Marine Isotope Stage 5e, Paleoceanography, 28(3), 585–598, doi:10.1002/palo.20052, 2013.

Giorgioni, M., Jovane, L., Rego, E. S., Rodelli, D., Frontalini, F., Coccioni, R., Catanzariti, R. and Özcan, E.: Carbon cycle instability and orbital forcing during the Middle Eocene Climatic Optimum, Scientific Reports, 9(1), 9357, doi:10.1038/s41598-019-45763-2, 2019.

[revised manuscript text omitted]

**Acknowledgements**

This research used samples and data provided by the International Ocean Discovery Program (IODP) and its predecessors. This work was carried out under the program of the Netherlands Earth System Science Centre (NESSC), financially supported by the Dutch Ministry of Education, Culture and Science. This study was made possible by the Netherlands
30  Organisation for Scientific Research (NWO) grant number 834.11.006, which enabled the purchase of the UHPLC-MS system used for GDGT analyses. Funding was provided by the Australian IODP office and the ARC Basins Genesis Hub (IH130200012) to SJG. We thank Natasja Welters, Jan van Tongeren and Arnold van Dijk (Utrecht University Geolab) for

analytical support. We thank the reviewers Severine Fauquette, Chris Hollis and G. Raquel Guerstein for their constructive reviews of the initial version of the manuscript.

Automatic citation updates are disabled. To see the bibliography, click Refresh in the Zotero toolbar.

**Author: References are available in the manuscript pdf**

Margot Cramwinckel 18/10/2019 18:00

**Main figures**

[Figure]

(a) Early Eocene circulation (~52 Ma)

[Figure]

(b) Middle Eocene circulation (~41 or ~39 Ma)

[revised manuscript text omitted]

Margot Cramwinckel 21/10/2019 16:35

---

## Referee Report (RR1)

I have carefully read the new version of the ms submitted by Cramwinckel and co-authors to Climate of the Past. The authors modified most of the points highlighted by the three referees and presented arguments to support some of the concepts and hypothesis they decided to maintain. As I mentioned in my first review, the most critical point was the lack of physical arguments to explain the proposed change in the Southern Ocean's surface circulation through the MECO, which is the crucial aspect of this work. I regret to say that despite the information added to the discussion (point 5.1), the work still lacks physical foundations for the proposed hypothesis. I will refer in detail to the authors´s answers to my concerns (AR in blue italics) and part of the revised version (RV in black italics). I also have comments and serious worries about the new Fig. 1 and Fig.3.

*AR. We explore several mechanisms and identify the one we consider most likely (southward extent of the EAC). Indeed, such changes in surface ocean circulation would follow changes in the wind pattern - given bathymetric and geographic constraints. We would like to emphasize that the bathymetric/paleogeographic constraints are just as important as the wind patterns, and both are much less well-constrained than the existing model simulations seem to suggest.*

Given the dinoflagellate cyst distribution the authors pick-up a possible explanation (a southward extention of the EAC). The problem is that to sustain this explanation the current should be more than 15º south of its present configuration. Without a proper physical explanation of how the current can attain the studied area this hypothesis is as weak as the one proposed be Kennet during the 70´s (See Huber et al., 2004).

*AR: While we agree that it would be insightful to draw in the prevailing wind directions in the Eocene, unfortunately these reconstructions do not reliably exist, so we respectfully refrain from drawing them. The middle Eocene, as this additional step would introduce a lot of uncertainty. Alternatively, drawing wind circulation patterns as derived from model simulations does not provide a solution either. Atmospheric simulations as derived from fully-coupled coarse resolution GCMs (that are tuned to reproduce modern conditions), are still limited by the poorly-constrained Eocene boundary conditions. Detailed model output is too dependent on these poorly resolved boundary conditions in order to be leading in drawing atmospheric reconstructions.*

On one side the authors have the dinocyt assemblage distribution, or the fossil plankton biogeography (THE DATA). On the other side, they are suggesting a paleoceanographic model based on the dinocyst distribution (THE INTERPRETATION). Any surface ocean circulation hypothesis at this scale has to be consistent with an atmospheric circulation pattern at the same scale. So, the authors cannot *choose to draw ocean circulation patterns based on fossil plankton biogeography, but prefer not to infer wind circulation patterns from this.* If they are not confident about the prevaling wind direction they cannot be confident on the EAC extension either. Again, the main hypothesis of this manuscript is weakly supported by ocean-atmposphere physics. If the authors cannot explain (physically) the huge southward shift of the EAC the hypothesis should be carefully revised and discussed and probably disregarded.

RV, page 14 line 15: *As the second option, southward extension and/or intensification of the EAC could have sustained cosmopolitan assemblages at Site 1172 (Figure 1c). Increased southward reach of the relatively warm EAC has been suggested before as a mechanism to warm the SWP throughout the hot early Eocene (Hollis et al., 2012; Hines et al., 2017).*

Hollis et al. (2012) show model simulations for the middle Eocene and EECO. Their Fig. 7 shows a nortward flowing western boundary current in all scenarios (presumably, because the caption does not indicate what the arrows are). I do not understand how these simulations could help the authors to sustain their hypothesis of a southward flowing EAC reaching 60°S.

In their conclusions Hines et al. (2017) state: "Intensification of a proto-East Australian Current (EAC) during the EECO provides an efficient means of oceanic heat distribution, subsequently resulting in the decreased thermal gradient from the equator to poles suggested by Southwest Pacific proxy records". This, like the one proposed in the revised manuscript, is a very surprising conclusion, because there is not a single paragraph in Hines et al (2017) explaining how and why such a proto EAC could be generated. What is the (physical) forcing mechanism that induces "an intensification of a proto-East Australian Current (EAC) and corresponding weakening of the Tasman Current (TC)" (Hines et al, 2017). It seems like the physics of climate is again underestimated.

RV, page 14 line 18: *Model simulations (using modern boundary conditions) indicate that a wind-driven strengthening and further southward extent of the EAC is expected under conditions of enhanced global warmth, as part of intensification of the southern midlatitude circulation (Cai et al., 2005). Indeed, observational data indicate a strengthening of the South Pacific Gyre over the past six decades, including a southward extent of the EAC at the expense of the Tasman Front (Hill et al., 2008, 2011). Similarly, SST anomaly reconstructions over the peak interglacial Marine Isotope Stage 5e (~125 ka) indicate intensification of the EAC to offshore Tasmania (Cortese et al., 2013). Possibly a similar atmospheric and oceanographic response to global warming occurred during MECO.*

Indeed, all the papers about the present EAC extension are related to changes of the wind stress curl (See Hill et al., 2011, 3rd Paragraph:"The pattern of wind stress curl determines the strength and spatial pattern of the gyre [Munk, 1950]. Hence variations in the basin-scale wind field will drive variability in the strength of the western boundary current"). So, how many latitudinal degrees outspreads today the southward extension of the EAC? According to Ridgway and Hill (2009) it corresponds to a poleward extension of some 350 km (~ 3 degrees). How this scenario can be used to explain the EAC extension reaching 60°S?

Cai et al., 2005 indicates an intensification of wind stress curl and of the EAC transport but it seems that the zero of the wind stress curl does not shift southward very much. Therefore it is still an open question how the EAC can flow until 60ºS as proposed by the authors. Remember that to support the proposed hypothesis we are talking of a 15 degree southward extension of the EAC. This would require a major change of the global wind stress distribution whose drivers are not explained in the revised manuscript.

Cortese et al., 2013 indicate an intensification of the EAC until ~45S. If a similar atmospheric response to global warming occured during MECO then the current should reach 45ºS and not 60ºS. Note again that the authors must change the latitude of zero wind stress curl to push the EAC farther south (page 1, 3$^{rd}$ paragraph of Hill et al, JGR, 2011). The large (proposed) southward extension of the EAC should be correlated with a corresponding change in the spatial wind pattern during the MECO. Unless there is a climatological (physical) explanation of how (and why) these changes are produced the hypothesis is flawed.

Considering an alternative hypothesis the authors indicate (page 14 line 8): *Two possible oceanographic features could have resulted in a dominantly cosmopolitan dinocyst assemblage at Site 1172 and not at Site 1170. First, weak eastward flow could have occurred through Bass Strait and/or the northern portion of the Tasmanian Gateway from the AAG (Figure 1c). The uncertainty on paleolatitude in principle allows for weak continuous eastward flow (or discontinuous eddy transport) under influence of the westerlies through the northern part of the TG. While this remains a possible scenario, we consider it unlikely that such a nearby current would not be reflected in the plankton assemblages at the depocenter of Site 1170, particularly since the widest opening in the TG would be located south of the South Tasman Rise (Bijl et al., 2013b), close to Site 1170. In addition, the Bass Strait, or Bass Basin, to the north of Tasmania was likely too restricted at its eastern end for throughflow (Cande and Stock, 2004).*

According to Fig 3b *Enneadocysta multicornuta* (cosmopolitan) along with other cosmopolitan dinocysts and low-mid latitud dinocyts were already well represented at the Site 1172 from the bottom of the core and even since 480 m upwards (see supplementary data). Those levels are considered to be about 44 Ma in age. It is clear that those species were close to the 1172 Site at ~4Ma before the MECO. Perhaps a weak eastward flow would reach the ETP (Site 1172) but not the STR (Site 1170), dominated by the TC and a proto-ACC. To understand the increase of different taxa during the MECO we can use a good explanation settle by the authors  (From 5.2, page 15, line 15)*: Taken together, these results confirm previous evidence that once a surface-oceanography-tracking plankton community has become established, relative abundance changes within the community correspond closely with changes in SST (Bijl et al., 2011). T*he surface temperature rise during the MECO would have resulted in increased production of the cosmopolitan  Enneadocysta multicornuta and other cosmopolitan taxa on the ETP but not on the STR, where the dominant species is

Enneadocysta dictyostila. This species is the member of the Antarctic endemic assemblage most tolerant to warm surface waters.

Thus, the authors should look for a source of waters bringing cosmopolitan taxa to the Site 1172 area since at least 44 Ma. Perhaps this interesting dataset needs both tectonic and climate mechanisms to find an acceptable hypothesis, which certainly in not an EAC extention reaching 60°S during the MECO.

Comments on Fig. 1. This new figure instead of adding some light to this work presents serious theoretical mistakes and is really confusing:

Fig 1 b represents both the pre and post MECO scenarios and Fig 1c the paleoceanographic situation during the MECO. I cannot understand why the shallow connections are resticted to the MECO interval (Fig 1c). Why the increase in SST produces an incipient eastward flow through the northern part of the TG from the AGG and then, when the temperatures decrease to normal conditions (at about 39Ma), the connection is blocked again? From the tectonic point of view this sequence is really strange. Fig.1 needs revision and / or a suitable explanation.

---

## Author Response (AR2)

**Message to the editor**

Dear editor Yannick Donnadieu,

Thank you for your patience with our revision, and for thinking along with us in order to work towards an acceptable compromise for the discussion paragraph on regional surface ocean circulation.

Please find our author reply and the second revision of the manuscript (with tracked changes) attached below. As we discussed in follow-up emails, we have tried to accommodate both the reviewers comments and your recommendations in a much elaborated paragraph 5.1. First, we now present the various possible mechanisms that could explain our dinocyst records (transport by Tasman Current, transport through Tasmanian Gateway, transport by East Australian Current) as alternative scenarios next to each other. In the second half of the paragraph, we focus on the uncertainties in available model simulations, and make the case that, at present, we cannot use these to fully constrain the plausibility of the different scenarios. We outline the three major obstacles: limitations in model resolution, uncertainty on geographic boundary conditions, and the prevailing regional proxy-model temperature mismatch. In this discussion, we also incorporate your comments on the interplay between deep-water formation and surface-ocean circulation. Rather than trying to choose one of the proposed scenarios with too limited means (wich did not seem to work in the revised version of our manuscript), we would like to focus a bit more on what future research might be needed to distinguish between these options. See also the AR below.

As we also noted in our previous email, forthcoming collaborations between several of the co-authors of the present manuscript and the group of Anna von der Heydt (Utrecht University) will present more high-resolution simulations that might shed more light on this issue in the future. We recognize, however, that these results are as yet preliminary and unpublished, and cannot use these here.

Finally, next to the thorough revision of discussion paragraph 5.1, we did another run through the entire text to fix minor typos and style issues.

We hope that with this, we have improved the discussion of the manuscript to now represent a comprehensive overview of the possibilities (and their uncertainties) that might explain our MECO dinocyst assemblages. Note that since all of these scenarios represent surface-ocean circulation change, we feel that the current title covers the main subject matter and conclusions of the manuscript.

On behalf of all of the co-authors,
Best regards,
Margot Cramwinckel

**Reply to review, author reply in blue**

I have carefully read the new version of the ms submitted by Cramwinckel and co-authors to Climate
of the Past. The authors modified most of the points highlighted by the three referees and presented arguments to
support some of the concepts and hypothesis they decided to maintain. As I mentioned in my first review, the
most critical point was the lack of physical arguments to explain the proposed change in the Southern Ocean's
surface circulation through the MECO, which is the crucial aspect of this work. I regret to say that despite the
information added to the discussion (point 5.1), the work still lacks physical foundations for the proposed
hypothesis. I will refer in detail to the authors´s answers to my concerns (AR in blue italics) and part of the
revised version (RV in black italics). I also have comments and serious worries about the new Fig. 1 and Fig.3.

AR: We thank the reviewer, G. Raquel Guerstein, for critically thinking along with our manuscript. Below, we
reply to her comments, integrating several comments into one answer.

Given the dinoflagellate cyst distribution the authors pick-up a possible explanation (a southward extention of the
EAC). The problem is that to sustain this explanation the current should be more than 15° south of its present
configuration. Without a proper physical explanation of how the current can attain the studied area this
hypothesis is as weak as the one proposed be Kennet during the 70´s (See Huber et al., 2004).

On one side the authors have the dinocyt assemblage distribution, or the fossil plankton biogeography (THE
DATA). On the other side, they are suggesting a paleoceanographic model based on the dinocyst distribution
(THE INTERPRETATION). Any surface ocean circulation hypothesis at this scale has to be consistent with an
atmospheric circulation pattern at the same scale. So, the authors cannot *choose to draw ocean circulation*
*patterns based on fossil plankton biogeography, but prefer not to infer wind circulation patterns from this.* If they
are not confident about the prevaling wind direction they cannot be confident on the EAC extension either.
Again, the main hypothesis of this manuscript is weakly supported by ocean-atmposhere physics. If the authors
cannot explain (physically) the huge southward shift of the EAC the hypothesis should be carefully revised and
discussed and probably disregarded.
RV, page 14 line 15: *As the second option, southward extension and/or intensification of the EAC could have*
*sustained cosmopolitan assemblages at Site 1172 (Figure 1c). Increased southward reach of the relatively warm*
*EAC has been suggested before as a mechanism to warm the SWP throughout the hot early Eocene (Hollis et al.,*
*2012; Hines et al., 2017).*

Hollis et al. (2012) show model simulations for the middle Eocene and EECO. Their Fig. 7 shows a nortward
flowing western boundary current in all scenarios (presumably, because the caption does not indicate what the
arrows are). I do not understand how these simulations could help the authors to sustain their hypothesis of a
southward flowing EAC reaching 60°S.

In their conclusions Hines et al. (2017) state: "Intensification of a proto-East Australian Current (EAC) during the
EECO provides an efficient means of oceanic heat distribution, subsequently resulting in the decreased thermal
gradient from the equator to poles suggested by Southwest Pacific proxy records". This, like the one proposed in
the revised manuscript, is a very surprising conclusion, because there is not a single paragraph in Hines et al
(2017) explaining how and why such a proto EAC could be generated. What is the (physical) forcing mechanism
that induces "an intensification of a proto-East Australian Current (EAC) and corresponding weakening of the
Tasman Current (TC)" (Hines et al, 2017). It seems like the physics of climate is again underestimated.

RV, page 14 line 18: *Model simulations (using modern boundary conditions) indicate that a wind-driven strengthening and further southward extent of the EAC is expected under conditions of enhanced global warmth, as part of intensification of the southern midlatitude circulation (Cai et al., 2005). Indeed, observational data indicate a strengthening of the South Pacific Gyre over the past six decades, including a southward extent of the EAC at the expense of the Tasman Front (Hill et al., 2008, 2011). Similarly, SST anomaly reconstructions over the peak interglacial Marine Isotope Stage 5e (~125 ka) indicate intensification of the EAC to offshore Tasmania (Cortese et al., 2013). Possibly a similar atmospheric and oceanographic response to global warming occurred during MECO.*

Indeed, all the papers about the present EAC extension are related to changes of the wind stress curl (See Hill et al., 2011, 3rd Paragraph:"The pattern of wind stress curl determines the strength and spatial pattern of the gyre [Munk, 1950]. Hence variations in the basin-scale wind field will drive variability in the strength of the western boundary current"). So, how many latitudinal degrees outspreads today the southward extension of the EAC? According to Ridgway and Hill (2009) it corresponds to a poleward extension of some 350 km (~ 3 degrees). How this scenario can be used to explain the EAC extension reaching 60°S?

Cai et al., 2005 indicates an intensification of wind stress curl and of the EAC transport but it seems that the zero of the wind stress curl does not shift southward very much. Therefore it is still an open question how the EAC can flow until 60ºS as proposed by the authors. Remember that to support the proposed hypothesis we are talking of a 15 degree southward extension of the EAC. This would require a major change of the global wind stress distribution whose drivers are not explained in the revised manuscript.

Cortese et al., 2013 indicate an intensification of the EAC until ~45S. If a similar atmospheric response to global warming occured during MECO then the current should reach 45ºS and not 60ºS. Note again that the authors must change the latitude of zero wind stress curl to push the EAC farther south (page 1, 3rd paragraph of Hill et al, JGR, 2011). The large (proposed) southward extension of the EAC should be correlated with a corresponding change in the spatial wind pattern during the MECO. Unless there is a climatological (physical) explanation of how (and why) these changes are produced the hypothesis is flawed.

Considering an alternative hypothesis the authors indicate (page 14 line 8): *Two possible oceanographic features could have resulted in a dominantly cosmopolitan dinocyst assemblage at Site 1172 and not at Site 1170. First, weak eastward flow could have occurred through Bass Strait and/or the northern portion of the Tasmanian Gateway from the AAG (Figure 1c). The uncertainty on paleolatitude in principle allows for weak continuous eastward flow (or discontinuous eddy transport) under influence of the westerlies through the northern part of the TG. While this remains a possible scenario, we consider it unlikely that such a nearby current would not be reflected in the plankton assemblages at the depocenter of Site 1170, particularly since the widest opening in the TG would be located south of the South Tasman Rise (Bijl et al., 2013b), close to Site 1170. In addition, the Bass Strait, or Bass Basin, to the north of Tasmania was likely too restricted at its eastern end for throughflow (Cande and Stock, 2004).*

AR: This discussion is about plausibility. The reviewer and editor are of the opinion that the limits of plausibility are solely defined by the current state of GCM simulations, a field of which we are well-aware and actively contribute to, also regarding the present question (e.g., Huber et al., 2004; Sijp et al. 2014, 2016; Baatsen et al. 2016, 2020; Nooteboom et al. in review). We argue, however, that the current state of simulations is incomplete to fully assess plausibility. The simulations most likely correctly reflect large-scale circulation patterns. However, model skill in representing smaller-scaled features such as regional winds and EAC extent strongly depend on bathymetric and geographic constraints as well as model resolution. It is clear that for the Eocene Tasman region only few sensitivity studies have been run and not a single high-resolution (eddy-permitting) model simulation has been published. This is also apparent from the editor's comments. In our view, this means we cannot solely assess plausibility based on the available modelling literature. It also implies that a plausible explanation from
biogeography cannot be excluded based on the available simulations for a tectonically and oceanographically complex situation such as the Eocene Tasman region.

This is important because our hypothesis concerning southward extent of the EAC to warm the SWP in already hot periods of the Eocene has been proposed in the recent literature, even in combination with model
components. As discussed, Hollis et al. (2012) specifically indicate in their data-model integration paper that "it is possible that warmer proxy SSTs for the western Tasman Sea (Fig.7) may signal the influence of a proto-East Australian Current (EAC)". They clearly do not disregard this hypothesis based on their simulations, nor do follow-up papers, and our biogeographical data are consistent with this hypothesis.

Collectively, we argue it would be scientifically incorrect to disregard this hypothesis based on our results and the existing modelling literature. However, given the strong arguments by reviewer and editor, we will adapt our manuscript and refrain from selecting a preferred hypothesis as we agree that we cannot "prove" it with available proxy data and model simulations. Therefore, taking into account the above discussion on the validity of southward EAC extension to the location of Site 1172 in the middle Eocene, combined with recommendations by
the editor, we have decided to strongly revise our discussion paragraph 5.1 - on surface ocean circulation patterns during MECO. In the revised discussion, we now present three potential scenarios (warming TC, southward extent of EAC, eastward transport through northern portion of TG), that might explain our recorded dinocyst assemblage data next to each other (5.1.1 in revised manuscript). We now further explicitly discuss the difficulties in using existing model simulations to choose between these scenarios, and clarify that uncertainties
in surface ocean and wind patterns are large due to uncertain paleogeographic/-bathymetric boundary conditions, the regional proxy-model mismatch in temperature, as well as limited model resolution (5.1.2 in revised manuscript).

According to Fig 3b *Enneadocysta multicornuta* (cosmopolitan) along with other cosmopolitan dinocysts and
low-mid latitud dinocyts were already well represented at the Site 1172 from the bottom of the core and even since 480 m upwards (see supplementary data). Those levels are considered to be about 44 Ma in age. It is clear that those species were close to the 1172 Site at ~4Ma before the MECO. Perhaps a weak eastward flow would reach the ETP (Site 1172) but not the STR (Site 1170), dominated by the TC and a proto-ACC. To understand the increase of different taxa during the MECO we can use a good explanation settle by the authors (From 5.2, page
15, line 15)*: Taken together, these results confirm previous evidence that once a surface-oceanography-tracking plankton community has become established, relative abundance changes within the community correspond closely with changes in SST (Bijl et al., 2011). T*he surface temperature rise during the MECO would have resulted in increased production of the cosmopolitan Enneadocysta multicornuta and other cosmopolitan taxa on the ETP but not on the STR, where the dominant species is Enneadocysta dictyostila. This species is the member
of the Antarctic endemic assemblage most tolerant to warm surface waters.

AR: TEX$_{86}$-based temperatures rise to similar values during the MECO. Both *E. multicornuta* and *E. dictyostila* are present at both Site 1170 and Site 1172 prior to the MECO, although in different abundances. Therefore, the different response at both sites does not seem to be a community change in response to SST. We have added this
point to the discussion (p14, lines 12-15 of the revised manuscript); "As both species do occur at both sites, and similar SSTs are reached during MECO, a purely paleoecological explanation for this disparity seems unlikely.

Therefore, the difference between the dinocyst response at the East Tasman Plateau and South Tasman Rise requires a change in surface ocean current configuration and dinocyst transport".

Thus, the authors should look for a source of waters bringing cosmopolitan taxa to the Site 1172 area since at least 44 Ma. Perhaps this interesting dataset needs both tectonic and climate mechanisms to find an acceptable hypothesis, which certainly in not an EAC extention reaching 60°S during the MECO.

AR: Indeed, there are some cosmopolitan taxa present at Site 1172 throughout the Eocene, in accordance with previous studies (Bijl et al., 2011, 2013). Essentially, this is part of their definition, as cosmopolitan taxa can principally occur at low to high latitudes. The striking observation here is the sharp increase in abundance of these taxa, even to dominating percentages, during peak MECO, not their presence itself. We refer to the above discussion regarding EAC reach and alternative explanations.

Comments on Fig. 1. This new figure instead of adding some light to this work presents serious theoretical mistakes and is really confusing:

Fig 1 b represents both the pre and post MECO scenarios and Fig 1c the paleoceanographic situation during the MECO. I cannot understand why the shallow connections are resticted to the MECO interval (Fig 1c). Why the increase in SST produces an incipient eastward flow through the northern part of the TG from the AGG and then, when the temperatures decrease to normal conditions (at about 39Ma), the connection is blocked again? From the tectonic point of view this sequence is really strange. Fig.1 needs revision and / or a suitable explanation.

AR: We attempted a revision of the figure along the lines of the reviewers' advice but clearly we did not succeed. We now clearly state in the caption that we are only presenting scenarios that might explain the peak MECO dinocyst assemblages in Figure 1c, as the other currents we draw are also primarily inferred based on dinocyst assemblages.

**Author reply references**

Baatsen, M., van Hinsbergen, D. J. J., von der Heydt, A. S., Dijkstra, H. A., Sluijs, A., Abels, H. A. and Bijl, P. K.: Reconstructing geographical boundary conditions for palaeoclimate modelling during the Cenozoic, Clim. Past, 12(8), 1635–1644, doi:10.5194/cp-12-1635-2016, 2016.

Baatsen, M., Heydt, A. S. von der, Huber, M., Kliphuis, M. A., Bijl, P. K., Sluijs, A. and Dijkstra, H. A.: The middle-to-late Eocene greenhouse climate, modelled using the CESM 1.0.5, Climate of the Past Discussions, 1–44, doi:https://doi.org/10.5194/cp-2020-29, 2020.

Bijl, P. K., Pross, J., Warnaar, J., Stickley, C. E., Huber, M., Guerstein, R., Houben, A. J. P., Sluijs, A., Visscher, H. and Brinkhuis, H.: Environmental forcings of Paleogene Southern Ocean dinoflagellate biogeography, Paleoceanography, 26(1), PA1202, doi:10.1029/2009PA001905, 2011.

Bijl, P. K., Sluijs, A. and Brinkhuis, H.: A magneto- and chemostratigraphically calibrated dinoflagellate cyst zonation of the early Palaeogene South Pacific Ocean, Earth-Science Reviews, 124, 1–31, doi:10.1016/j.earscirev.2013.04.010, 2013.

Hollis, C. J., Taylor, K. W. R., Handley, L., Pancost, R. D., Huber, M., Creech, J. B., Hines, B. R., Crouch, E. M., Morgans, H. E. G., Crampton, J. S., Gibbs, S., Pearson, P. N. and Zachos, J. C.: Early Paleogene temperature history of the Southwest

Pacific Ocean: Reconciling proxies and models, Earth and Planetary Science Letters, 349–350, 53–66, doi:10.1016/j.epsl.2012.06.024, 2012.

Huber, M., Brinkhuis, H., Stickley, C. E., Döös, K., Sluijs, A., Warnaar, J., Schellenberg, S. A. and Williams, G. L.: Eocene circulation of the Southern Ocean: Was Antarctica kept warm by subtropical waters?, Paleoceanography, 19(4), PA4026, doi:10.1029/2004PA001014, 2004.

Nooteboom, P. D., Delandmeter, P., van Sebille, E., Bijl, P. K., Dijkstra, H. A. and von der Heydt, A. S.: Resolution-dependent variations of sinking Lagrangian particles in general circulation models, arXiv:2004.07099 [physics] [online] Available from: http://arxiv.org/abs/2004.07099 (Accessed 3 May 2020), 2020.

[revised manuscript text omitted]

==NB The entire below paragraph (5.1) in blue has been adapted:==

**5.1 Surface-ocean circulation in the southwest Pacific Ocean during the MECO**

### 5.1.1 Dinocyst constraints on MECO ocean circulation
Our new dinocyst biogeographic data indicate that Antarctic endemic taxa dominated at the South Tasman Rise during the middle Eocene, while they did not occur in the Otway Basin within the Australo-Antarctic Gulf. Furthermore, a mixed assemblage with sparse endemics was found at Hampden Beach, New Zealand. This is generally consistent with previous interpretations of the surface-ocean circulation in and around the Tasmanian Gateway based on dinocyst biogeography (Bijl et al., 2011, 2013b) (**Figure 1b**). By the middle Eocene, the Antarctic endemic dinocyst assemblage associated with the proto-ACC and TC had become firmly established in the SWP and on the Antarctic margin, while the northern bound of the AAG remained primarily influenced by the low-latitude-derived PLC (Bijl et al., 2011; Houben et al., 2019). Records from southern New Zealand yield a predominantly warm EAC signal (Hines et al., 2017), with a minor, yet constant influx of Antarctic endemics indicating a limited TC influence (this study and Huber et al., 2004; Bijl et al., 2011).

Throughout the studied middle Eocene interval, dinocyst assemblages at Site 1170 are dominated by Antarctic-endemic taxa. This implies that the Tasmanian Gateway was influenced by westward atmospheric and surface-oceanic circulation (i.e., the polar easterlies) around 40 Ma, with the polar front thus being located to the north of the gateway and the proto-ACC flowing westward through the Tasmanian Gateway (**Figure 1b**). This is supported by the similar range of $TEX_{86}$-derived SSTs of 20–28ºC within (Site 1170) and east of (Site 1172) the Tasmanian Gateway (**Figure 3**). In terms of paleolatitude reconstructions, placing Site 1170 within the Tasmanian Gateway south of 60ºS at this time is within the uncertainty limits of current generation mantle (e.g., Matthews et al., 2016) as well as paleomagnetic reference frames (e.g., Torsvik et al., 2012). Notably, however, the shift in dominance from endemic to cosmopolitan dinocysts at the peak of MECO warmth on the East Tasman Plateau (Site 1172) (Bijl et al., 2010) has no equivalent on the South Tasman Rise (Site 1170) (**Figure 3**). The dominant species at Site 1172 during peak MECO is cosmopolitan *Enneadocysta multicornuta*, while at the same time endemic *Enneadocysta dictyostila* is dominant at Site 1170. While belonging to the same dinocyst genus and thus being morphologically closely related, these species have very different biogeographic affinities (e.g., Bijl et al. 2011). As both species occur at both sites and similar SSTs are reached during MECO, a purely paleoecological explanation for this disparity seems unlikely. Therefore, the difference between the dinocyst response at the East Tasman Plateau and South Tasman Rise requires a change in surface-ocean current configuration and dinocyst transport.

Given the dinocyst biogeographic patterns that are in place prior to the MECO (Bijl et al., 2011), three different surface-ocean currents might have brought dominant cosmopolitan dinocysts to Site 1172 during peak MECO: 1. flow from the southeast (with the Tasman Current), 2. flow from the north (as a southward extension of the East Australian Current), or 3. flow from the west (through the Tasmanian Gateway) (options depicted in **Figure 1c**).

In Scenario 1, a warmer TC could have resulted in a higher abundance of non-endemic species, including *E. multicornuta* that was able to dominate the assemblage under peak MECO warmth. However, in this scenario it seems likely that the TC would have supplied the same dinocyst assemblages to waters overlying Site 1170 at the STR – where we do not find them. In Scenario 2, a southward extension of the EAC could have sustained cosmopolitan assemblages at Site 1172. An increased southward reach of the relatively warm EAC has been suggested before as a mechanism to warm the SWP throughout the hot early Eocene (Hines et al., 2017; Hollis et al., 2012). Model simulations for the modern system indicate that a wind-driven strengthening and further southward extent of the EAC is expected under conditions of enhanced global warmth, as part of intensification of the southern midlatitude circulation (Cai et al., 2005). Similarly, SST anomaly reconstructions over the peak interglacial Marine Isotope Stage 5e (~125 ka) indicate intensification of the EAC to offshore Tasmania (Cortese et al., 2013).

Possibly a similar atmospheric and oceanographic response to global warming occurred during MECO. General circulation model simulations have not yielded an EAC that reaches this far south during the Eocene (Lunt et al., 2012, 2020). However, simulations using a range of plausible geographies and bathymetries with higher resolution might show different small-scale circulation patterns in this sensitive region, therefore we cannot discard this hypothesis at present. Although the Eocene EAC was likely located too far north to extend it all the way to Site 1172, we cannot, for example, rule out the possibility of eddie- diffused influence of the EAC towards the ETP. In Scenario 3, throughflow from the AAG into the SWP could have brought cosmopolitan assemblages to the ETP and not to the STR, if flow went through Bass Strait and/or the northern portion of the Tasmanian Gateway. The uncertainty on paleolatitude in principle allows for weak continuous eastward flow (or discontinuous eddy transport) under influence of the westerlies through the northern part of the TG. Although the Bass Strait, or Bass Basin, to the north of Tasmania might have been too restricted at its eastern end for throughflow, this remains poorly constrained (Cande and Stock, 2004).

**5.1.2 Comparison to simulated ocean-circulation patterns**

At present, none of the above scenarios can be excluded given the available proxy data, geographical and bathymetrical constraints, and simulations. Climate-model simulations of various complexity can provide insights into the ocean circulation patterns that are possible under specific boundary conditions, the latter of which include, e.g., geography/bathymetry and radiative forcing. Complex general-circulation models show multi-model consistency in simulating broad-scale gyral circulation in the south Pacific and south Atlantic/Indian sectors of the Eocene Southern Ocean (Lunt et al., 2012). However, the available simulations do not allow us to conclude which of the above scenarios was the most likely for the Tasman region due to three primary reasons that we briefly discuss in the following.

The first is limitations in model resolution. In available climate-model simulations for the Eocene, mesoscale eddies are parametrized and not resolved, which affects the details of surface-ocean circulation, especially in regions of high eddy activity such as the (modern) Southern Ocean (Rintoul, 2018). Nooteboom et al. (in review) demonstrate the crucial role eddie-diffusivity plays in realistically simulating ocean flow in models, particularly in assessing ocean flow from sedimentary microplankton assemblages. Eddie flow is notably important at oceanic fronts, and also western boundary currents such as the

EAC (Kirtman et al., 2012). Although low-resolution models adequately simulate general supra-regional ocean flow, the lack of proper eddie-parameterization complicates using these models for the detailed purpose of assessing the above scenarios, particularly given the uncertainties in boundary conditions.

Secondly, there are large uncertainties regarding important details of regional (middle Eocene) geographic, bathymetric and paleolatitudinal boundary conditions, and the timing and consequences of progressive changes therein (*e.g.*, Hollis et al., 2012).

These include the geometry of the TG conduit (both Tasmania-Antarctica and Tasmania-Australia), the bathymetry of relative bathymetric highs such as the STR and Lord Howe Rise, sea level and the regional shape of the coastlines around the Tasman Gateway, all of which might have a large impact on the resulting ocean flow (e.g., Olbers and Eden, 2003; Olbers et al., 2007 for the modern ACC). Although the available model simulations use best estimates on geography/bathymetry, the uncertainties remain large as a full sensitivity study is of course not feasible. This causes large uncertainty on both the plausibility of PLC eastward flow as well as the possibility for southward extent of the EAC. The impact of paleolatitude uncertainty becomes particularly evident when comparing moving hotspot reference frames with paleomagnetic reference frames (van Hinsbergen et al., 2015). Although the difference between these two models and the uncertainties within both are only a few degrees latitude, this may cause large differences in ocean-current direction through the TG, which is located close to the boundary between the Antarctic easterlies and mid-latitude westerlies (Baatsen et al., 2016; Hines et al., 2017). Although the sensitivity of the TG to paleolatitude has been recognized conceptually (e.g., Scher et al., 2015), the oceanographic consequences of these varying paleolatitudinal models have not been simulated and compared in detail. Of relevance for the above Scenarios 2 and 3 are recent middle-late Eocene model simulations using CESM 1.0.5 (Baatsen et al., 2020) – with the Seton et al. (2012) paleogeography and the paleomagnetic reference frame from Torsvik et al. (2012) (described in Baatsen et al., 2016) – simulating Southern Ocean frontal systems that were located much further south than under modern conditions, facilitating both southward EAC and eastward PLC flow.

The third reason considers the implications of the persistent regional mismatch between proxy- and model-based SST reconstructions on the interpretation of model simulations. While the match between multi proxy-based and model-simulated global mean temperature is improving for the (early) Eocene, the regional mismatch in SST in the SWP has remained large (Lunt et al., 2012, 2020), indicating gaps in our understanding of how heat was distributed regionally. Around New Zealand and south of Australia, model-based SSTs are 5–10 °C cooler than proxy estimates in simulations that show reasonable data-model consistency elsewhere. This results in much stronger local SST gradients in the model simulations than apparent in SST reconstructions. These generally lower SSTs for the SWP in model simulations will tend to increase the simulated strength and locus of deep-water formation in the South Pacific, affecting horizontal ocean flow as well through the pull from deep-water formation. This implies that underestimation of simulated absolute SSTs in the Eocene SWP possibly overestimates the simulated TG throughflow, and in turn, regional ocean current structure. Although proxy data are consistent with southern Pacific deep-water formation (Hollis et al., 2012; Huck et al., 2017), it is unknown what the strength, exact locus (Antarctic shelf *versus* deep-sea) or precise seasonality of that deep-water formation was.

In summary, while at this point we cannot select one of our proposed scenarios, our data show the presence of dinocyst assemblages derived from a lower-latitude current east of the TG, and an Antarctic-derived current in the central TG during the zenith of MECO warmth. We stress the importance of this issue given the proximity of the Tasmanian Gateway to the likely region of deep-water formation in the South Pacific, and the possible interaction between gateway throughflow and deep-water formation in the Eocene. We propose to revisit this issue with future simulations using high-resolution eddypermitting models and improved constraints on regional paleogeography and bathymetry. These would be necessary to assess which of these scenarios is realistic in response to higher global temperatures during the Eocene and peak 
[revised manuscript text omitted]
., 2011; Henehan et al., 2020), but direct, physical evidence for presence of ice prior to the MECO has remaineds absent so far. Even with small glaciers present, accommodation space on the continental shelves (on time scales of $10^6$–$10^7$ years) was primarily determined by the interplay of steric components, sediment supply and basin subsidence. In general, warm and wet early Eocene conditions are expected to have saturated passive continental shelves, resulting in relatively flat and shallow shelf platforms (Sømme et
al., 2009). In the Otway Basin, sediments of middle Eocene age (basal Nirranda Group) overlie a large unconformity at the top of early Eocene sediments of the Wangerrip Group (e.g., Krassay et al., 2004). These middle Eocene sediments were deposited during the Wilson Bluff transgression, which is recognised throughout southeast Australia (Holdgate et al., 2003; McGowran et al., 2004) and has been linked to a major transgressive phase in the Indo-Pacific (the Khirthar transgression) (Jauhri and Agarwal, 2001; McGowran et al., 2004). While there is seismostratigraphic evidence for regional tectonic rifting,
normal faulting and subsidence during the Paleocene and early Eocene in southeast Australia (Krassay et al., 2004; Close et al., 2009), it is unknown when subsidence terminated, and renewed. Additionally, a progressive decrease in terrigenous sediment supply as the Australian hinterland aridified throughout the Eocene might have affected accommodation space (Sauermilch et al., 2019). Whatever the relative contributions of these mechanisms, the hiatus between the Wangerrip Group and the Nirranda Group suggests no or negative accommodation space by the end of the early Eocene (51 Ma) or later. The renewed drowning of the continental shelf, as reflected in the Wilson Bluff transgression, seems unlikely to be related to slow and continuous basin subsidence. Instead, ocean warming during the MECO may have raised global average sea level by several meters by thermal expansion, while warmer and wetter regional climate could have increased sediment supply. The resumption of sedimentation accumulation above the top Latrobe unconformity has been previously dated to between 44 and 40 Ma (Holdgate et al., 2003; McGowran et al., 2004). Based on our new dinocyst-based age constraints, it is likely that the sediments overlying the Wangerrip group are close to the MECO in age, suggestive of a causal link between the Wilson Bluff transgression and MECO warming. A similar timing of renewed sedimentation occurred in the Schöningen section in the North German Basin, where the transgressive, fully marine Annenberg Formation unconformably overlies the Lutetian coal-bearing Helmstedt Formation (Riegel et al., 2012). The Annenberg Formation has been assigned an age around the MECO (Gürs, 2005), possibly ~41 Ma (Brandes et al., 2012). Based on a compilation of New Jersey coastal plain sections, a highstand (sequence E8) is also interpreted at ~41–40 Ma (Browning et al., 2008).

Sea-level rise and warming during the MECO may have accommodated increased burial of biogenic carbonate on continental shelves, explaining a reduction in carbonate burial in the deep sea (Sluijs et al., 2013), along with a diminished silicate weathering feedback (Van der Ploeg et al., 2018). However, it should be noted that the above inferences regarding global sea-level rise during the MECO are tentative. Although these transgressive surfaces all have an age around the MECO, current age control is not nearly sufficient to correlate them to MECO with certainty. A dating accuracy of ≤100,000 years would be required for these transgressive surfaces to indicate their relationship to MECO warming, which is presently not available. It is therefore crucial to improve these constraints in order to assess the potential influence of sea-level change on the carbon cycle during the MECO.

**6        Conclusions**

Comparison of plankton and sea-surface temperature patterns during the MECO above the South Tasman Rise indicates that while dinocyst assemblages as a whole responded to surface-water warming,. However, the acme in cosmopolitan taxa above the East Tasman Plateau at peak MECO is not mirrored at the STR. This implies either transport by a warming Tasman Current, a southward extension of the EAC during the zenith of MECO warmth, or eastward throughflow through the northern portion of the Tasmanian Gateway, or a southward extension of the EAC during the zenith of MECO warmth. While we cannot distinguish between these scenarios, Tthis seems to illustrates how profoundly surface-ocean currents can respond to external climate forcing in these regions of the Southern Ocean. Terrestrial palynomorph assemblages indicate that a warm temperate rainforest with some paratropical elements grew along the southeast Australian margin during the MECO. Finally, we suggest that the southeast Australian Wilson Bluff Transgression may be related to sea-level rise during the MECO, but improvement of the available age constraints is necessary to establish a possible causal link.

**Acknowledgements**

This research used samples and data provided by the International Ocean Discovery Program (IODP) and its predecessors.
This work was carried out under the program of the Netherlands Earth System Science Centre (NESSC), financially supported by the Dutch Ministry of Education, Culture and Science. This study was made possible by the Netherlands Organisation for Scientific Research (NWO) grant number 834.11.006, which enabled the purchase of the UHPLC-MS system used for GDGT analyses. Funding was provided by the Australian IODP office and the ARC Basins Genesis Hub (IH130200012) to SJG. We thank Natasja Welters, Jan van Tongeren and Arnold van Dijk (Utrecht University Geolab) for analytical support. We thank
the reviewers Severine Fauquette, Chris Hollis and G. Raquel Guerstein for their constructive reviews and Yannick Donnadieu for thorough editing of the manuscript.

**Main figures**

[Figure]

**(a)** Early Eocene surface ocean circulation (~52 Ma)

NB Figure 1c and the figure caption have been adapted:

**Figure 1. Inferred generalised Eocene surface ocean circulation patterns in the southwest Pacific Ocean** based on dinocyst biogeographic patterns. **(a)** Generalised early Eocene (~52 Ma) surface ocean circulation. **(b)** Generalised middle Eocene surface ocean circulation pre-MECO (~41 Ma) and post-MECO (~39 Ma). **(c)** Three potential surface ocean circulation scenarios (S1, S2 and S3) for peak MECO (~40 Ma) based on dinocyst assemblage constraints (cf. section 5.1). Maps constructed with GPlates, using Torsvik et al. (2012) paleomagnetic rotation frame and Matthews et al. (2016) continental polygons and coastlines for 52 Ma (a) and 40 Ma (b and c). Note that, within this rotation frame, there is uncertainty on the drawn paleolatitudes. For example, Site 1170 is drawn at 61.6 °S at 40 Ma, but the uncertainty margins on this are between 58.76 °S and 64.55 °S (van Hinsbergen et al., 2015). Currents drawn after reconstructions by Bijl et al. (2011, 2013b, 2013a) and this study. EAC = East-Australian Current; PLC = Proto-Leeuwin Current; TC = Tasman Current; proto-ACC = proto-Antarctic Counter Current.

[Figure]

**(b)** Middle Eocene surface ocean circulation (~41 or ~39 Ma)

[Figure]

**(c)** Peak MECO surface ocean circulation scenarios (~40 Ma)

[revised manuscript text omitted]

van der Ploeg, R., Selby, D., Cramwinckel, M. J., Li, Y., Bohaty, S. M., Middelburg, J. J. and Sluijs, A.: Middle Eocene greenhouse warming facilitated by diminished weathering feedback, Nature Communications, 9(1), 2877, doi:10.1038/s41467-018-05104-9, 2018.

Prentice, I. C.: Non-Metric Ordination Methods in Ecology, Journal of Ecology, 65(1), 85–94, doi:10.2307/2259064, 1977.

Pross, J., Contreras, L., Bijl, P. K., Greenwood, D. R., Bohaty, S. M., Schouten, S., Bendle, J. A., Röhl, U., Tauxe, L., Raine, J. I., Huck, C. E., van de Flierdt, T., Jamieson, S. S. R., Stickley, C. E., van de Schootbrugge, B., Escutia, C., Brinkhuis, H. 10  and Scientists, I. O. D. P. E. 318: Persistent near-tropical warmth on the Antarctic continent during the early Eocene epoch, Nature, 488(7409), 73–77, doi:10.1038/nature11300, 2012.

Raine, J. I., Mildenhall, D. C. and Kennedy, E. M.: New Zealand fossil spores and pollen: an illustrated catalogue. 4th edition., GNS Science miscellaneous series no. 4 [online] Available from: http://data.gns.cri.nz/sporepollen/index.htm, 2011.

Riegel, W., Wilde, V. and Lenz, O. K.: The early Eocene of Schöningen (N-Germany) - an interim report, Austrian Journal of 15  Earth Sciences, 105(1), 88–109, 2012.

Rintoul, S. R.: The global influence of localized dynamics in the Southern Ocean, Nature, 558(7709), 209–218, doi:10.1038/s41586-018-0182-3, 2018.

Röhl, U., Brinkhuis, H., Stickley, C. E., Fuller, M., Schellenberg, S. A., Wefer, G. and Williams, G. L.: Sea Level and Astronomically Induced Environmental Changes in Middle and Late Eocene Sediments from the East Tasman Plateau, in The 20  Cenozoic Southern Ocean: Tectonics, Sedimentation, and Climate Change Between Australia and Antarctica, edited by N. F. Exon, J. P. Kennett, and M. J.lone, pp. 127–151, American Geophysical Union. [online] Available from: http://onlinelibrary.wiley.com/doi/10.1029/151GM09/summary (Accessed 17 November 2015), 2004.

Sangiorgi, F., Dinelli, E., Maffioli, P., Capotondi, L., Giunta, S., Morigi, C., Principato, M. S., Negri, A., Emeis, K.-C. and Corselli, C.: Geochemical and micropaleontological characterisation of a Mediterranean sapropel S5: A case study from core 25  BAN89GC09 (south of Crete), Palaeogeography, Palaeoclimatology, Palaeoecology, 235(1–3), 192–207, doi:10.1016/j.palaeo.2005.09.029, 2006.

Sauermilch, I., Whittaker, J. M., Bijl, P. K., Totterdell, J. M. and Jokat, W.: Tectonic, Oceanographic, and Climatic Controls on the Cretaceous-Cenozoic Sedimentary Record of the Australian-Antarctic Basin, Journal of Geophysical Research: Solid Earth, 124(8), 7699–7724, doi:10.1029/2018JB016683, 2019.

Scher, H. D. and Martin, E. E.: Circulation in the Southern Ocean during the Paleogene inferred from neodymium isotopes, Earth and Planetary Science Letters, 228(3), 391–405, doi:10.1016/j.epsl.2004.10.016, 2004.

Scher, H. D., Whittaker, J. M., Williams, S. E., Latimer, J. C., Kordesch, W. E. C. and Delaney, M. L.: Onset of Antarctic Circumpolar Current 30 million years ago as Tasmanian Gateway aligned with westerlies, Nature, 523(7562), 580–583, doi:10.1038/nature14598, 2015.

[revised manuscript text omitted]